# Novel tissue mechanics-guided cellular flows drive the formation of feather follicles

Hans I-Chen Harn [ID][1], Ting-Xin Jiang [ID][1], Chih-Han Huang[1,2,3], Wen-Tau Juan [ID][4,5], Tzu-Yu Liu [ID][1,6], Tsao-Chi Chuang[4,5], Wan-Chi Liao[4,5], Yingxiao Wang[7,8], Ji Li [ID][9], Cornelis J Weijer [ID][10], Ping Wu [ID][1], Chin-Lin Guo [ID][11] & Cheng-Ming Chuong [ID][1 ✉]

## Abstract

**Complex tissue architecture is achieved through multiple rounds of morphological transitions. Here, we analyzed cellular flows and tissue mechanics during avian skin development by employing chicken and transgenic quail skin explant models. We demonstrate how novel cellular flows initiate chemo-mechanical circuits that guide epithelial protrusion, folding, invagination, and spatial cell fate specification. During initial feather bud formation, stiff dermal condensates protrude vertically from the locally softened epithelial sheet. As the bud elongates, it stretches the epithelial cells at the base, thus mechanically activating YAP, which causes the epithelial sheet to fold downward and form a stiff cylindrical wall that invaginates into the skin. This stiff epithelial tongue is essential for the compaction and formation of the tightly packed dermal papillae. These topological transformational events are mechanically interconnected, and the completion of one circuit initiates the next. In contrast, during scale development, the rigid epithelial sheet restricts dermal cell flows, preventing further topological transformation. Based on these findings, we developed a topological transformation model describing how this process enabled the evolution of feather follicles from scales.**

**Subject Categories** Development; Evolution & Ecology

## Introduction

The integument forms the boundary between an organism and its environment and evolves according to essential functions. In early chordates, such as amphioxus, the integument is smooth and lacks appendages. In fish, scales begin to form through epithelial placode development. More complex skin appendages gradually emerge, enabling vertebrates to adapt to diverse ecological niches (Dhouailly, 2023; Wu et al, 2004). Various types of scales, glands, and spines are generated through simple topological transformations such as epithelial folding, invagination, protrusion, and branching. The most complex structures are follicles, found in feathers, hair, and reptile teeth (Fig. EV1A). Although these follicles evolved independently, they share defining characteristics: stem cells, a dermal niche, and an architecture that supports cyclic renewal throughout the organism's lifetime (Fig. EV1B). Among them, feather follicles are the most complex. The filament cylinder contains progenitor cells at the proximal end (Yue et al, 2005) that undergo branching morphogenesis toward the distal end (Li et al, 2017; Yu et al, 2002). The dermal papilla not only signals epidermis but also undergoes cyclic renewal itself (Wu et al, 2021b). At different developmental stages, mature feathers exhibit distinct phenotypes (Chen et al, 2015; Chen et al, 2024). This diversity is made possible by follicles that house stem cells (Yue et al, 2005) and support cyclic regeneration.

During feather evolution, the capacity to molt and regenerate was a prerequisite for generating diverse feather forms. While the evolutionary development (evo-devo) of feather forms has been extensively studied, tracing back to feathered dinosaurs (Li et al, 2017; Prum, 2005; Xu et al, 2014), fewer studies have addressed the evolution of feather follicles. Fossil evidence suggests that the two feather forms found in the oviraptorosaur *Similicaudipteryx* (Prum, 2010; Xu et al, 2010) may represent the primary feather transition process still observed in modern birds (Chen et al, 2024). Analyses of flight feather fossils further suggest that sequential molting occurred in *Microraptor* (Kiat et al, 2020) and other Mesozoic birds. These processes require follicle-based molting and regeneration, pointing to the existence of ancient feather follicles in feathered dinosaurs. Meanwhile, diffusible morphogens such as Shh, Wnt, BMP, and Eda are present not only in reptile scales and other planar integuments but also in feather morphogenesis (Chen et al, 2015; Di-Poi and Milinkovitch, 2016). With similar molecular tools available, what drove planar epithelia to transform into a complex architecture such as the feather follicle (Fig. EV1B)?

Here, we take a biophysical approach to study how feather follicles are built and evolved, aiming to understand the key

[1]Department of Pathology, Keck School of Medicine, University of Southern California, Los Angeles, CA, USA. [2]Ostrow School of Dentistry, University of Southern California, Los Angeles, CA, USA. [3]Division of Cardiovascular Surgery, Department of Surgery, Tri-Service General Hospital, National Defense Medical University, Taipei, Taiwan. [4]Department of Biomedical Imaging and Radiological Science, China Medical University, Taichung, Taiwan. [5]Department of Medical Research, China Medical University Hospital, Taichung, Taiwan. [6]Department of Life Sciences, National Cheng Kung University, Tainan, Taiwan. [7]Department of Bioengineering & Institute of Engineering in Medicine, University of California, San Diego, La Jolla, CA, USA. [8]Alfred E. Mann Department of Biomedical Engineering, University of Southern California, Los Angeles, CA, USA. [9]Department of Dermatology, Xiangya Hospital, Central South University, Changsha, China. [10]School of Life Sciences, University of Dundee, Dundee, United Kingdom. [11]Institute of Physics, Academia Sinica, 128 Sec 2 Academia Rd, Nankang District, Taipei, Taiwan. ✉E-mail: cmchuong@med.usc.edu

biomechanical–biochemical crosstalk. The morphogenetic process can be viewed as a sequence of symmetry-breaking events (Prum, 2005) involving cell migration and tissue folding, with cellular forces and movements serving as crucial drivers that initiate and execute these transitions, leading from simple to complex architectures (Collinet and Lecuit, 2021; Tozluoglu et al, 2019). Sculpting biological structures requires dynamic changes in stiffness contrasts (Lenne and Trivedi, 2022), reflected in changes in cellular movements and mechanical properties. Local adjustments in tissue stiffness and fluidity, in turn, generate differences that allow cells to collectively migrate or "flow," shaping the developing organ. Here, we define "cellular flows" as the collective, spatially organized movement of cell populations during morphogenesis, encompassing both active migration (cell-autonomous, cytoskeleton-driven translocation) and passive displacement (movement resulting from mechanical forces, tissue deformation). For example, in zebrafish, tissue jamming and unjamming, and transitions between solid and fluid states, underlie unidirectional axis elongation (Mongera et al, 2018). During Drosophila wing disc development, multiple epithelial invaginations and buckling events generate localized, anisotropic forces that drive symmetry breaking and tissue folding (Bailles et al, 2019; Tozluoglu et al, 2019). However, the chemo-mechanical regulation of morphogenesis in bilayered composite tissues involves continual interactions between epithelia and mesenchyme, and how such dynamics produce a complex ectodermal organ, such as the feather, remains unresolved.

At different stages of avian skin development—feather bud formation, follicle invagination, and dermal papilla formation—epidermal and dermal cells migrate synergistically, transforming a simple two-dimensional placodal sheet into a three-dimensional follicle architecture that houses spatially distinct cell fates (Chuong et al, 2000). How does epithelial folding influence dermal cell flow and fate specification? How do chemo-mechanical factors control anisotropic tissue expansion, enabling transitions from simple to complex multidimensional structures? In this study, we use chicken and transgenic quail skin explant models to characterize cellular flows, tissue mechanics, chemo-mechanical coupling molecules, and to simulate these interactions during different stages of feather development. We selected exemplary key molecules and identified their roles not only in molecular signaling but also in initiating critical mechanical events that drive sequential stages of morphogenesis. Based on cellular flow, spatial mechanical, and molecular findings, we developed a mechano-chemical coupling model of follicle formation that describes the initiation and extension of epidermal invagination around the dermal condensate (DC) and the subsequent establishment of the dermal papilla (DP). This model distills the essential parameters of topological transformation. We then tested its predictions in scutate scale formation and in the induction of feather follicles from scales (Lai et al, 2018; Wu et al, 2018b). Our findings highlight follicle formation as an evolutionarily novel morphogenetic process. New cell flows, driven by mechanics, enable the construction of a stem cell–based follicle.

# Results

## Stiff dermal cell condensate vertically protrudes locally softened epithelial sheet during bud formation

To characterize cell flow dynamics of the developing skin, we recorded 4D (x,y,z,t) confocal videos using RCAS-H2B-mOrange-labeled chicken skin explants. RCAS-H2B-mOrange virus were injected into the somite of the chicken embryo and amniotic sac to transfect and label the nucleus of dermal and epidermal cells, respectively, at E3. During early feather bud formation (E7-E8), dermal condensations (DC) form via chemo-mechanically sensitive cell proliferation and migration, such as those mediated by Fgf (Song et al, 1996; Widelitz et al, 1996). DC formation is characterized by dermal cells that first migrate horizontally, which later protrude out to form the initial feather bud (Fig. 1A,B). The bud shown is in the short bud stage, which later elongates to form the long bud (Li et al, 2013; Li et al, 2017). Here, we focus on the formation of the short bud. Cell tracking analysis demonstrates the changes of cellular flow directions during different stages of feather bud formation (Fig. 1A; Movie EV1). In the first 6 h of culturing, most of the dermal cell movements occur in the horizontal (xy) plane parallel to the epidermal layer, forming the DC, which is accompanied by a decrease in the distance between neighboring cells (Fig. EV2A). Interestingly, between E7 6–18 h, the DC cells begin to move vertically (z) in a direction perpendicular to the epidermal layer, forming the protrusion of the early feather bud (Fig. 1B). This is accompanied by the increase in the average distance to the nearest cells (Fig. EV2A, late protrusion). The illustration shows dermal cells undergo a switch in movement from the horizontal to vertical cellular direction during feather bud protrusion (Figs. 1C and EV2A).

How do these dermal cells change their migration direction to move upward? Considering that the FGF-stimulated DC cells exhibit high motility and are confined by the epidermis, one likely mechanism to initiate their vertical upward motion is to have a mechanically weak point at the epidermis above the DC. We used atomic force microscopy (AFM), an image-based quantitative morphology field (QMorF) measurement (Wu et al, 2021a), and laser ablation of the epidermal cell junction to investigate the spatial mechanical dynamics and the associated cellular morphing of epidermis and dermis in the E8 feather bud in contrast to surrounding non-bud regions. The stiffness heatmaps demonstrate that the whole skin feather bud is more rigid ($273 \pm 31$ Pa) than the surrounding peribud region ($187 \pm 19$ Pa, Figs. 1D, left and Fig. EV2B, $P < 0.001$, $n = 9/5$). We then separate the epidermis and dermis to examine their contributions to tissue rigidity separately. The dermis shows a similar stiffness difference, as well as an overall decrease in stiffness (bud: $239 \pm 22$ Pa, interbud: $178 \pm 19$ Pa, Fig. EV2B, $P < 0.001$). Interestingly, the peeled epidermis shows an opposite stiffness map. The bud region is softer ($189 \pm 17$ Pa), while the inter-bud region is stiffer ($209 \pm 21$ Pa) than that of the dermis (Figs. 1D and EV2B, F $(1, 8) = 37.66$, $P = 0.0031$ $n = 9/5$). We also found that the regional stiffness of the bud region increases during dermal condensation and early feather bud protrusion (Fig. EV2C, $n = 3/3$). These findings demonstrate a spatial correlation between locally softened epidermis and vertical DC movement.

To evaluate whether cell proliferation contributes significantly to feather bud protrusion, we quantitatively analyzed dermal cell division during the E8 + 18 h time window. We found that the location of proliferation occurs mostly at the basal posterior end of the feather bud (circle), and detected less than 7% of total cells showing cell division activity (Fig. EV2D–F, averaged 48 events in 695 nuclei, $n = 3/3$), indicating that local dermal proliferation plays a minor role. This is also consistent with prior reports that proliferation is spatially restricted to the basal posterior region of

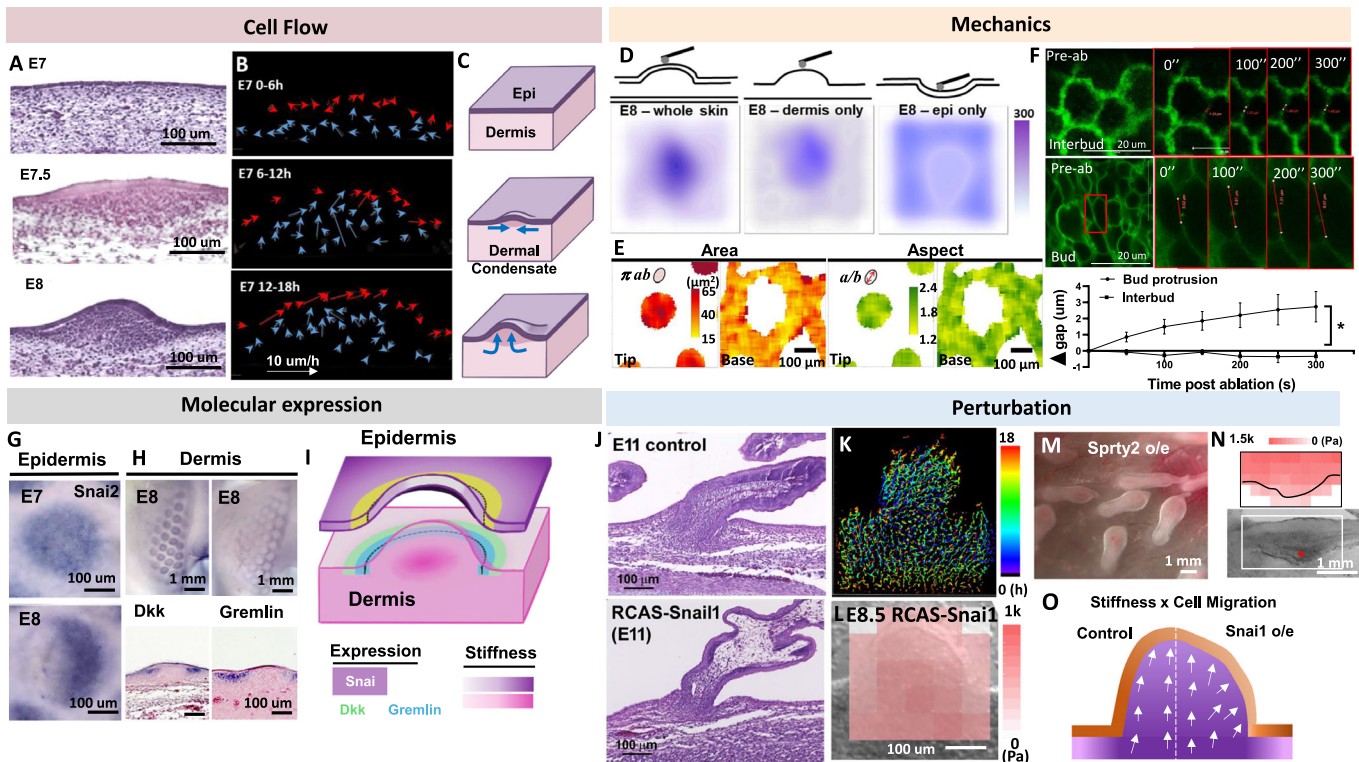

**Figure 1. Feather bud protrusion: Vertical cellular flows and the formation of bud boundary.**

(A) H&E images of chicken skin development from E7 to E8. (B) Cell tracking analysis of E7 chicken explant culture for 18 h reveals epidermal (red arrows) and dermal cell (blue arrows) migration patterns changed from horizontal to vertical during bud protrusion. The length and direction of the arrows indicate the velocity of the cells tracked. $n = 3/3$, biological replicates. (C) Illustration of cellular flows at different stages of early feather bud formation. (D) Landscape of tissue stiffness measured by AFM. Shown here are E8 whole skin, or dermis, epidermis alone. $n = 3/3$, biological replicates. (E) QMorF analysis showing the spatial patterns of cell deformation in epidermal cells in terms of cell area and aspect ratio during feather bud protrusion. The complete serial sections are shown in Fig. EV3E. $n = 3/3$, biological replicates. (F) Laser ablation analyses showing ablated cell membrane retracted faster in epidermal cells of E8 protruding feather buds than that of interbud region. The complete serial images are shown in Fig. EV4A. Two-way ANOVA, $F (1, 4) = 9.448$, $*P = 0.0372$, $n = 3/3$, biological replicates. Error bar: SD. (G, H) In situ hybridization of selective molecules. (G) Snai in E7 and E8 chicken skin placode epidermis. (H) DKK1 and Gremlin in E8 chicken skin dermis. DKK and Gremlin show a ring staining pattern of different radii in the whole mount. Section staining shows epidermal staining for Dkk (inside) and Gremlin (outside) flank bud boundary. $n = 3/3$, biological replicates. (I) Illustration of the spatial distribution of epidermal (purple) and dermal (pink) tissue stiffness and molecular expression in an early feather bud. Darker color indicates higher stiffness. (J) H&E of E11 control and RCAS-overexpressed Snai1 feather bud morphology. $n = 3/3$, biological replicates. (K) Representative E8 + 18 h RCAS-Snai1 feather bud cell tracks shown by time-scaled dragon tails. Color scale: 0–18 h. $n = 3/3$, biological replicates. (L) Stiffness heatmap of E8.5 RCAS-Snai1 feather bud in sagittal section. Measured using AFM. $n = 3/3$, biological replicates. (M) Sprouty2 overexpression in the epidermis causes buds to shorten and expand distally. Representative photo. $n = 3/3$, biological replicates. (N) Characterization of Sprouty perturbed buds. AFM stiffness heatmap and bright view photo of a bulging feather bud on one side of a locally Sprouty2-overexpressing feather filament. $n = 3/3$, biological replicates. (O) Stiffness and cell migration illustration. Epidermis softening through Snai1 overexpression on one side of a bud. On the perturbed side, dermal cell migration is directed more laterally, leading to a bulging feather bud. Source data are available online for this figure.

the protruding bud, and dermal condensate formation is driven predominantly by directed cell migration rather than local cell division (Riddell and Headon, 2025).

For force balance to occur, the upward vertical movement of dermal cells must be accompanied by reciprocal downward vertical forces exerted at the epidermal layer. We wonder what the reciprocal physical-morphogenic profile of the epidermal cells exhibits during bud protrusion. We use cell shape changes (by E-cadherin staining) and QMorF method (Chang et al, 2019; Wu et al, 2021a) to analyze the changes in cell shape dynamics at the skin surface. We observed that during early bud formation, the angle and aspect ratio of epidermal cells are converging and stretched towards the feather bud (Figs. 1E and EV3A–C). The initial analyses focused on cross-sections at different levels from the top (z = 54 μm) to the base (z = 0) of a developing feather bud at the E9 stage (Fig. 1E). The raw fluorescence image for each optical

section is displayed in (Fig. EV3C). QMorF distribution heatmaps of the coarse-grain averaged cross-sectional morphological characteristics (Fig. EV3B) including area (πab), aspect ratio (a/b), and orientation (θ) were illustrated in Fig. EV3D–F, respectively. Figure EV3D revealed that the average area decreases from the top to the base of the feather bud, reflecting the initial DC formation (Fig. 1A, E7.5) and later protrusion (Figs. 1A E8; and EV3C–I). To further examine the stretching force experienced by the epidermal cells at the protruding bud, we utilized skin explants from the membrane-bound GFP transgenic quails, performed laser ablation on the cell junction and recorded the change in gap distance over time. The results show that the ablated membrane at the protruding bud retracted significantly faster than that of the interbud region (Fig. 1F). implying epidermal cells at the bud tip experience greater passive stretching force from the actively protruding dermal cells than those at the interbud. The epidermis functions as a continuous

system, where the traction force exerted by the DC at the base is transmitted to the bud region through a force balance with the tension at the epidermal cell–cell junctions. Since the inter-bud epidermis is influenced by this traction force, it is expected that the tension in the inter-bud region is lower, whereas the bud region experiences higher tension to counterbalance the transmitted traction force, as demonstrated by membrane recoil after ablation.

What are the molecular responses following these spatiotemporally entangled physical-morphogenic events? Mechanical forces such as stretching have shown to be able to induce rapid changes in cellular behaviors such as epithelial-mesenchymal transition (EMT, a more fluidic-like behavior) by EMT-actuators such as Tgfβ, Snai1 and Slug (Leggett et al, 2021). Interestingly, we found that Snai1, an EMT-associated transcription factor that enhances epithelial migration, is expressed at the whole placode epidermis at E7, which later shifts toward the posterior bud epidermis (Fig. 1G) where the posterior bud protrusion continues on E8 (Li et al, 2013). Morphogen is also expressed with similar spatial patterns from E7 to E8 in the epidermal placode (Fig. EV4B). In the dermis, DKK1 is expressed at the periphery of DC. Gremlin, in a bigger ring configuration, is expressed outside of the DC (Fig. 1H). The observations above indicate that bud protrusion is associated with local changes in tissue mechanics (Fig. 1I).

To test whether tissue mechanics can reciprocally influence the physical property of the developing feather bud, we performed various molecular functional perturbations on the mechanical properties of the epidermis and dermis. Snai1, along with Snai2, is an important EMT-associated transcription factor that leads to various epithelial to mesenchymal state changes, including loosening of cell junctions, cell motility and shape changes (Yang et al, 2020). Overexpressing Snai1 via RCAS in the epidermis led to expanded feather bud morphology and less condensed dermal cells in the feather bud compared to control (Fig. 1J). Cell tracking analysis showed that dermal cells inside the mutant feather bud migrated laterally (xy plane) in addition to vertically (Figs. 1K and EV5A–C, $n = 3/3$), and the average distance to the nine nearest cells in the bud tip increased after 15 h, echoing the dispersal-like behavior observed (Fig. EV5C). AFM stiffness map also showed that RCAS-Snai1 overexpressed feather bud is composed of a significantly softer epidermis layer ($491 \pm 34$ Pa) than the dermis ($680 \pm 42$ Pa, Fig. 1L and Fig. EV5E, $P = 0.0007$, $n = 3/3$). Sprty2 has been shown to enhance epithelial invagination (Schneider et al, 2017; Sternlicht et al, 2006; Yue et al, 2012). By overexpressing Sprty2, we observed a visibly widened distal end and a shorter bud length in feather buds (Fig. 1M), and a decrease in epidermal stiffness (from $1325 \pm 180$ to $934 \pm 102$ Pa, Figs. 1N and EV5F,G, $P = 0.0109$, $n = 3/3$). These results suggest that epidermal stiffness helps confine the bud shape, and the softened epidermis may allow the dermal cells to migrate laterally, leading to the expansion of bud width (Fig. 1O). Mechanistically, cell flow and tissue mechanics both depend on cell contractility and adhesions.

To further test the impact of cell contractility dynamics on feather bud protrusion, we treated the E8 skin explants with an actin polymerization stabilizer, Jasplakinolide (Jasp) (Holzinger, 2001) and evaluated the ensuing changes. The changes in DC density and the average distance to nine nearest neighbor cells in DC both suggest that dermal cells were able to aggregate and form an early DC for the first 35 h, then rapidly disassembled afterward (Fig. EV6A). Laser ablation analysis showed that Jasp treatment

significantly increased cell membrane retraction distance at the bud protrusion and interbud region (Fig. EV6B,C, bud protrusion: $F (1.177, 4.707) = 12.15$, *$P = 0.0179$, interbud: $F (2.021, 8.084) = 7.817$, + $P = 0.0128$, $n = 3/3$). Jasp treatment also increased vinculin and phalloidin expression in E8 + 2 d explants, and led to mal-developed feather bud morphology (Fig. EV6D). On the other hand, we treated the explants with Blebbistatin—a known inhibitor of non-muscle myosin II and therefore cell contraction and migration (Wu et al, 2019). This led to a sac-like bud morphology. This may be due to the failure of dermal cells to actively migrate vertically upward and create reciprocal forces at the epidermis (Figs. 1M and EV4C). Similarly, treating cells with ADH-1, a peptide that blocks N-cadherin (Shintani et al, 2008), which is enriched in the DC, led to the formation of shorter feather buds (Fig. EV6F), suggesting that dermal cell–cell adhesion is an important factor contributing to the forces driving vertical movement during short bud formation. Together, these results show that feather bud protrusion is achieved through a sequence of spatial biomechanical–biochemical interactions and is characterized by dermal cells to actively migrate vertically and upward, while keeping the interbud skin in place.

## Bud elongation stretch-activates epidermal YAP and MMP in a ring around the bud base and initiates downward epidermal folding to form the stiff follicle wall

During follicle formation (E9–12), epidermal cells at the bud base invaginate downward into the dermis (Fig. 2A,B, red arrows), folding into a bilayered cylindrical wall around the presumptive dermal papilla (DP). Cell tracking analysis showed that while the feather bud elongated distally to form the putative pulp (Fig. 2A,B, yellow arrows; Movie EV2), dermal cells at the bud base redirected their migration downward to establish the putative DP (Fig. 2B, green arrows). Outside the invaginating wall, dermal cells migrated along the epithelial fold to form the putative dermal sheath (DS) (Fig. 2B, blue arrows). The schematic in Fig. 2C illustrates how epidermal invagination (red arrows) produces a cylindrical wall around the putative DP, thereby separating distinct dermal cell flows along the follicle wall (Fig. 2C, lower panel).

Mechanical measurements revealed that invaginating epidermal cells were more than twice as stiff ($1040 \pm 110$ Pa) as the surrounding dermis ($421 \pm 32$ Pa, Fig. 2D, $n = 3/3$, biological replicates). The DC ($721 \pm 74$ Pa) is also stiffer than the distal dermal cells ($322 \pm 31$ Pa, forming pulp) and basal dermal cells ($441 \pm 53$ Pa, forming DP). Together, these mechanical contrasts hint that stiffened epidermal cells, in cooperation with the DC, drive invagination and physically compartmentalize dermal cell fates during follicle formation.

Cell migration also exerts mechanical influences on surrounding tissues. To evaluate the cell shape changes associated with early chemo-mechanical remodeling at the bud base, we performed QMorF analysis of regional cellular deformation (Fig. 2E–G). As the bud elongated, the epidermal cells at the bud neck adjacent to the DC were stretched (Fig. 2F, white dotted circle), potentially activating downstream chemo-mechanical signals. Epidermal invagination represents a higher-order 3D transformation (Fig. EV3J–M) that can be analyzed using an azimuthal framework (Campas and Mahadevan, 2009), treating the DC as the central axis. This approach showed that mechanical coupling and cellular

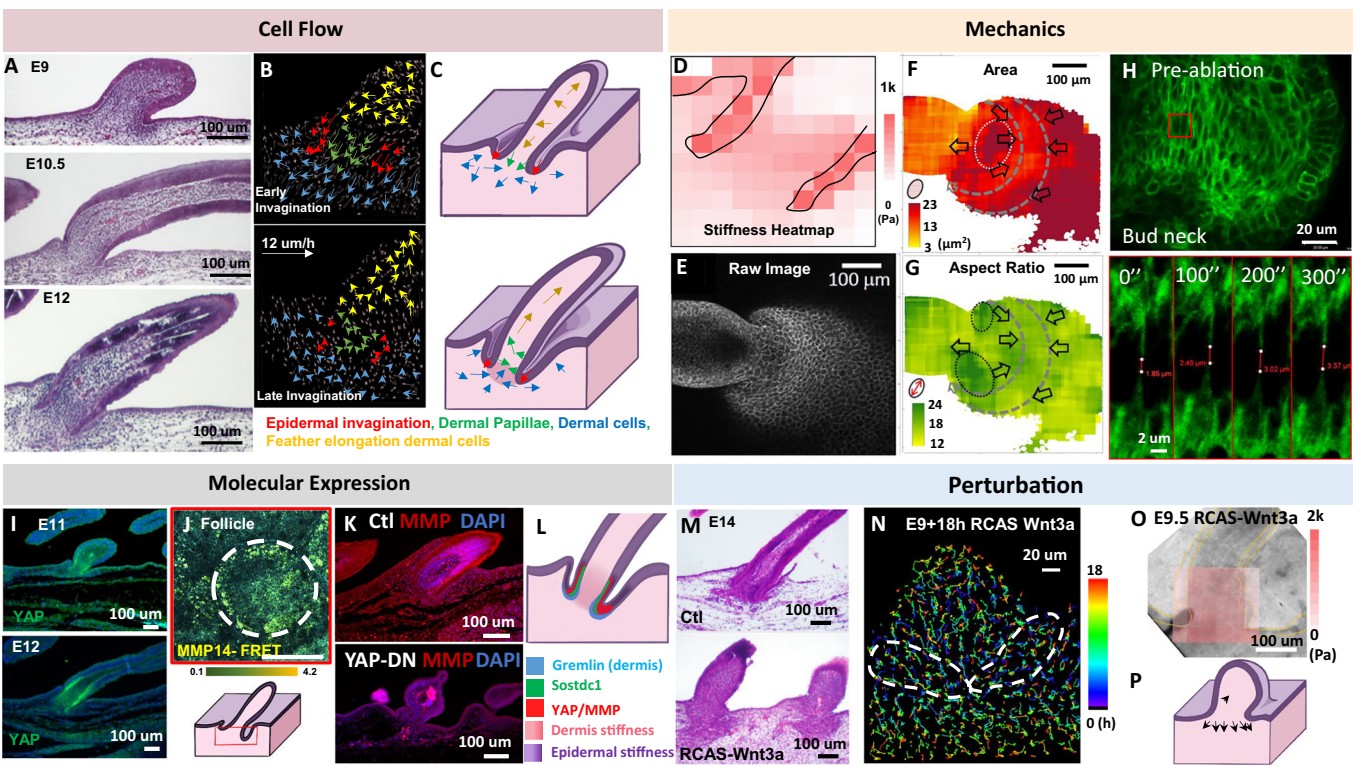

**Figure 2. Feather follicle invagination: Invaginating epidermal cells compartmentalize and create differential dermal cell flows.**

(A) H&E images of feather follicle invagination at E9, E10.5, and E12. $n = 3/3$, biological replicates. (B) Representative result of E10 + 24 h cell track analysis showing different cellular flows: (red arrows) epidermal invaginating cells, (green arrows) dermal papillae cells, (blue arrows) dermal cells, and (yellow arrows) feather elongation dermal cells. $n = 3/3$, biological replicates. (C) Illustration of cellular flows during early (upper) and late (lower) follicle invagination. The red arrows indicate invaginating epidermis that form as physical barriers and creates differential cell flows in the dermis, DP and feather bud. (D) Stiffness heatmap of E10 follicle invagination in a sagittal section of a follicle. Measured using AFM. Black lines demarcate the epidermis. $n = 3/3$, biological replicates. (E) Whole mount E10 feather bud with invagination stained with E-cadherin. $n = 3/3$, biological replicates. (F) QMorF analysis of panel (E) showing the spatial patterns of cell deformation in epidermal cells during feather follicle invagination. Arrows indicate directions of cell deformation, which are consistent with principal stress directions. $n = 3/3$, biological replicates. (G) QMorF analysis. Aspect ratio of E10 epidermal cells in and around a feather bud and invagination site. $n = 3/3$, biological replicates. (E–G) is also shown in Fig. EV3J–L. (H) Laser ablation analysis showing the ablated epidermal cell membrane retracted in the E9 feather bud neck. Representative photos shown, the complete serial images are shown in Fig. EV7A. Red block: region of ablation and analysis. (I) Immunohistochemistry of YAP in E11 and E12 feather bud during follicle invagination. $n = 3/3$, biological replicates. (J) The base of the E11.5 feather follicle shows MMP14-FRET activity. The illustration shows where FRET activity is recorded. $n = 3/3$, biological replicates. (K) E13 control and RCAS-dnYAP1 feather buds with MMP2 staining. YAP-DN feather bud shows limited invagination, abnormal morphology, and the lack of MMP at the invaginating epidermis (lower panel). $n = 3/3$, biological replicates. (L) Schematic summary showing Gremlin, Sostdc1, YAP, and MMP expression and spatial tissue stiffness of an E12 follicle. Darker color indicates higher tissue stiffness. (M–P) Perturbation of follicle formation by interrupting mechano-chemical coupling between epidermis and dermis. (M) H&E photos of E14 control and Wnt3a overexpression abrogated epidermal invagination, leading to dispersed dermal condensate cells, shortened feather buds and no follicle wall invagination. $n = 3/3$, biological replicates. (N) Representative result of cell tracking analysis showing dermal condensate cells (white dotted lines) dispersed pattern in E9 + 18 h feather bud, shown by time-scaled dragon tails. Color scale: 0–18 h. $n = 3/3$, biological replicates. (O) AFM stiffness map of E9.5 RCAS-Wnt3a feather bud at the invaginating region, showing the dermis is stiffer (~2000 Pa) than the base of epidermal cells (800 Pa). $n = 3/3$, biological replicates. (P) Illustration of abrogated epidermal invagination via softening epidermis causes dispersal of dermal condensate cells, and in turn leads to shortening and widening of feather buds. Source data are available online for this figure.

deformation organize cells concentrically, accommodating the azimuthal symmetry of the invaginating bud.

Formation of the azimuthal neck appears to involve a cellular band that shrinks circumferentially around the bud cylinder at the invagination site. Consistently, Fig. 2F shows an orange band of small cells between two gray dashed curves, corresponding to a region of reduced cellular area. Opposing force vectors at either side of this band likely form a force field interface that contributes to macroscopic follicle invagination (Figs. 2F,G and EV3).

Laser ablation at the bud neck (Figs. 2H and EV7A,B, two-way ANOVA, F (1.422, 5.687) = 7.419, $P = 0.0308$, $n = 3/3$) confirmed significantly faster cell membrane retraction compared to interbud

regions, indicating high tensile forces stored in invaginating epidermal cells that may serve as mechanical triggers for down-stream signaling. These findings suggest that cell shape anisotropy detected by QMorF correlates with stored mechanical forces (Figs. 2H and EV7A,B).

We next searched for molecules involved in mechano-chemical signaling at the bud neck. YAP, a stretch-activated transcriptional regulator, is known to mediate such coupling (Halder et al, 2012). YAP expression was strongly induced at the bud base, particularly in the invaginating epithelial walls (Fig. 2I), where MMP activity was also elevated (FRET ratio = 3.57 ± 0.58, Fig. 2J), in contrast to the apteric region (0.34 ± 0.07, $t$-test, $p = 0.0015$, Fig. EV7C, $n = 3/3$).

Gremlin and Sostdc1 were also enriched in the invaginating dermis and epidermis, respectively (Fig. EV7D). Overexpression of a dominant-negative YAP disrupted both epidermal invagination and MMP expression, producing shortened, malformed buds (Fig. 2K). Consistently, inhibition of MMPs impaired ECM remodeling and follicle invagination (Jiang et al, 2011). These findings suggest that bud elongation-induced stretching activates YAP MMPs, and other effectors, which in concert stiffen the epidermis and soften the dermis around the DC, thereby facilitating downward extension of the epidermal tongue (Fig. 2L).

Recent studies indicate that YAP/TAZ signaling cross-talks with Wnt/β-catenin pathways to regulate renewal and tissue homeostasis (Azzolin et al, 2014; Azzolin et al, 2012). To test whether Wnt signaling affects follicle mechanics, we overexpressed Wnt3a in developing skin. This produced broadened buds with wide bases and failed invagination (Fig. 2M). Longitudinal cell tracking confirmed that Wnt3a-overexpressing follicles lacked invagination, leading to dispersed dermal cell flows (Figs. 2N and EV8A) and increased average nearest neighbor distances (Fig. EV8B). AFM measurements showed that Wnt3a-overexpressing epidermal walls at E9.5 were significantly softer than the DC and surrounding dermis (821 ± 102 Pa vs. 1747 ± 225 Pa; Figs. 2O and EV8C,D, $P = 0.0048$. $n = 3/3$, biological replicates). This reduced stiffness affected invagination and produced shortened, widened buds without proper DCs (Fig. 2M,P). Overexpression of Snai1, which also softens epidermal cells (Fig. 1L), generated a similar phenotype (Fig. EV7E).

Together, these results indicate that follicle formation depends on stretching at the bud/interbud boundary, manifested as a stiffened epidermal wall, YAP activation, and elongated cell morphology. These mechanical cues trigger epithelial invagination, while sustained stiffness maintains continuous downward extension of the epithelial tongue. In concert with dermal cell migration, these events transform a 2D planar skin sheet into a 3D vertical follicular architecture (Fig. 2L).

## The stiff epidermal wall invaginates and compacts the Tgf-β emitting dermal papilla to form the base of the follicle and dermal papillae

To complete follicle formation (E17 in flight feathers), the epidermal wall (white line, Fig. 3A) extends to the follicle base, clamping around the DP (SMA+ cells, Fig. 3A), while the DS wraps around from the outside (Fig. 3A–C). Dermal cells condense as intercellular distances decrease and express SMA (α-smooth muscle actin) (Fig. EV9A,B), indicating higher compaction and stiffness (late invagination, green arrows; Figure EV9C, cyan line). Consistent with this, Martino et al (Martino et al, 2023) showed that dermal sheath–generated contractile forces orchestrate homeo-static tissue regression, suggesting that SMA+ cells—abundant during follicle development—serve as important force generators, contributing to tissue stiffness differences.

We next analyzed cellular flows and tissue rigidity in this final stage. Cell tracking revealed new, complex flows (Fig. 3D; Movie EV3). Epidermal wall cells at the base (lower red arrows) migrated downward and bent toward the putative DP, while cells above the DP migrated upward (upper red arrows), along with dermal cells inside the follicle (yellow arrows). By contrast, putative DP cells (green arrows) migrated more slowly. These movements could be divided into three subflows: below the epidermal wall (dark green arrows, Fig. 3D,E), level with the wall endings (green arrows), and within the wall (light green arrows). DP cells below the epidermal wall mostly migrated horizontally with slight downward shifts; those at the wall level moved horizontally inward; and those inside the wall migrated moderately upward (Fig. 3D,E). The illustration (Fig. 3D) depicts these intricate flows during DP formation, which parallel the cellular movements seen during adult feather cycling (Wu et al, 2021b).

Tissue stiffness was assessed in longitudinal sections using AFM. The epidermal wall and invaginating tongues exhibited the highest stiffness (1224 ± 132 Pa, $n = 3/3$, biological replicates), followed by the DP (Fig. 3F). Within the DP, the bottom region where epidermal tongues met the DP displayed the greatest stiffness, suggesting that these structures may not only act as physical barriers but also apply stress that compacts and seals the follicle floor (Fig. 3E,F). High stiffness correlated with high cell density, including DP cells (Fig. EV9D). SMA was strongly expressed by upper DP cells (Fig. 3G), aligned with observed flows (Fig. 3D), dispersing radially from the narrowest DP region. Beneath the follicle floor, dermal cells expressed moderate SMA and were circumferentially oriented (Fig. 3G), forming the DS cup to further seal the follicle base.

These findings indicate that dermal flows, coordinated with epithelial tongues, compartmentalize tissues into DS along the outer wall, pulp cells along the inner wall, and DP plus DS cup at the base. Pulp and DS cells were oriented longitudinally, whereas DP and DS cup cells were oriented circumferentially. Thus, local cell flows correlated with stiffness differences, and dermal cell orientation suggested that mechanical factors strongly influence both behavior and fate specification (Fig. 3E–G).

We then examined molecules mediating these biomechanical–biochemical interactions during DP completion. Tgf-β2 is known to induce mesenchymal condensation (Ting-Berreth and Chuong, 1996), activate SMA, and promote contraction and EMT (Massague and Sheppard, 2023). In lung fibroblasts, Tgf-β signaling induces expression of Fzd8 and its ligand Wnt5b, which together activate non-canonical Wnt signaling (Spanjer et al, 2016). We found Tgf-β2 expressed in dermal cells at the follicle base beginning at E14 and persisting through E17 (Fig. 3H), coinciding with high SMA in the DP and Fzd8 at the epidermal tip. These observations raise the possibility that DP cells-secreted Tgf-β2 may act as a chemotactic cue for epidermal invagination, consistent with Fzd8 expression at the epidermal tip. High YAP expression at the epidermal tongue tip and DP region of newborn flight feathers further implies that these cells experience high mechanical stress, potentially shaping DP into its final form (Fig. 3G–J).

To test Tgf-β2 function, we inhibited its activity with LY2109761, a Tgf-β I/II kinase inhibitor (Melisi et al, 2008), was applied to explants from E12 to E16. Tgf-β2 suppression reduced epidermal invagination and eliminated narrowing of the DP at the follicle base (Fig. 3K,L). Inhibiting signaling blocked DP-to-epidermis chemoattraction and prevented epidermis-mediated "DP shaping," producing a loosened DP and widened follicle base (Fig. 3K,L). Feather elongation was also impaired (shortened buds), consistent with mechanical cues being critical for initiating invagination (Fig. 2).

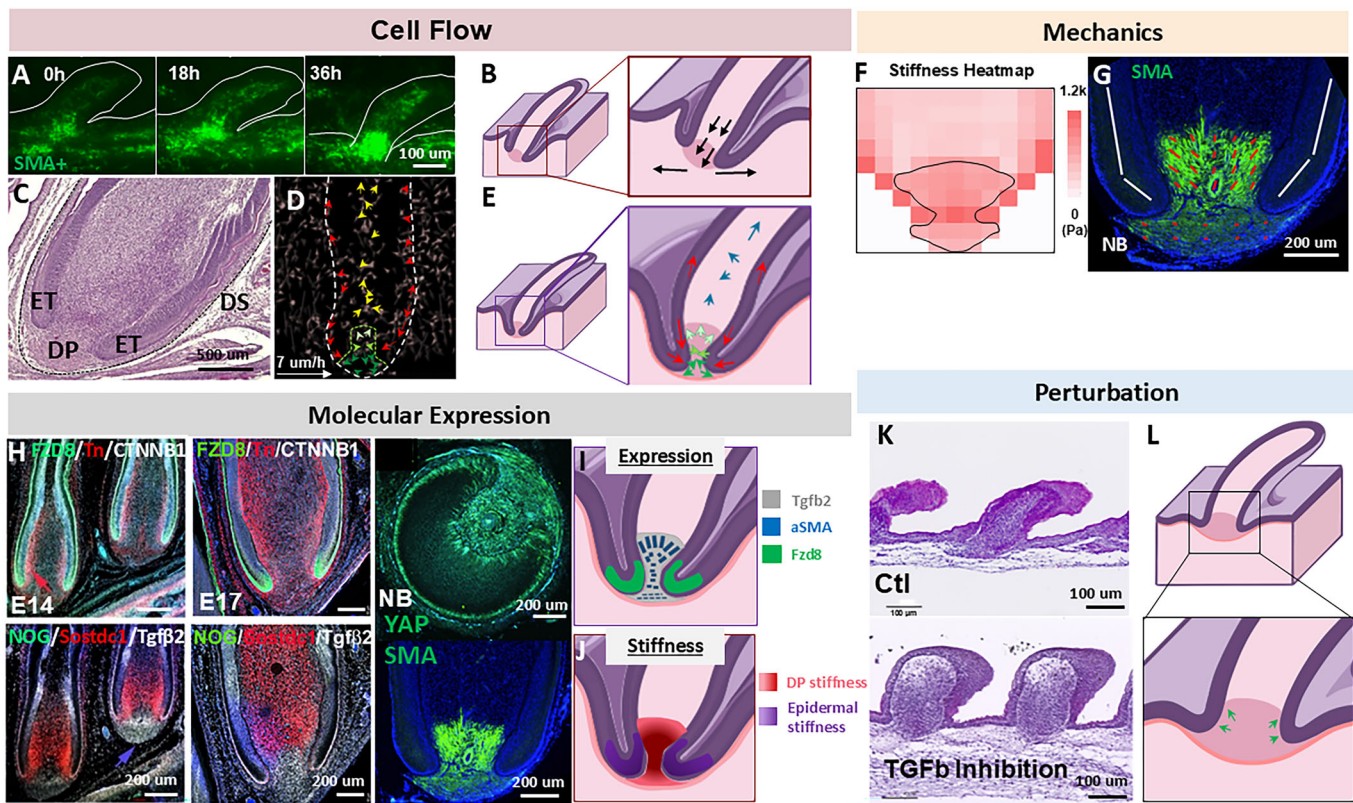

**Figure 3.  Formation of the follicle base: establishing the dermal papilla through compaction of epidermal tongues on dermal condensation cells.**

(A) Lateral view of time-lapse photos of lentivirus-labeled SMA+ cells (green) in E9 + 36 h explant. The white lines demarcate the outline of a feather bud. The panel is also shown in Fig. EV9A. n = 4/4, biological replicates. (B) Illustration of SMA+ cell migration pattern during early DP formation. (C) H&E of an E17 flight feather follicle to highlight the epidermal tongue (ET), DP and dermal sheath (DS, dotted line). n = 3/3, biological replicates. (D) Cell tracking of E17 flight feather to reveal differential cellular flows: (green) DP-forming cells migrate downwards. Epidermal tongue cells migrate downwards while distal feather epidermal cells migrate upwards (red). Feather elongating dermal cells migrate distally (yellow). The green dotted line demarcates presumptive DP, and top (light green), middle (green), and lower (dark green) DP cells show differential migration patterns. n = 3/3, biological replicates. (E) Illustration of feather follicle formation and differential cell movements in late stages. Blue: feather elongation dermal cells. Red: epidermal cells. Light, green, and dark green: top, middle and lower DP cells, respectively. (F) Stiffness heatmap of E17 flight feather dermal papillae. Sagittal section of follicles measured by AFM. Black lines demarcate dermal papillae. n = 5/5, biological replicates. (G) SMA+ cells in the DP region of a newborn (NB) feather. The orientation of SMA cells is highlighted by red lines, and the expression level by the length of the arrow. White line: alignment of the epidermal wall. (H) Molecular expression in feather follicles. FZD8, TNC, CTNNB1, NOG, SOSTDC1, Tgf-β2, YAP (longitudinal section), and SMA expression in E14, E17, and newborn (NB) flight feathers, particularly around the dermal papillae region. n = 3/3, biological replicates. (I) Illustration of the spatial expression of Tgf-β2, SMA and Fzd8 during late follicle formation. (J) Illustration of DP and epidermal spatial stiffness during late follicle formation. (K, L) Perturbation of the follicle base formation by disruption of Tg-fβ2 signaling. (K) H&E images of control (Ctl) and Tgf-β2-inhibitor-treated explant (E12 + 4 d) showing disrupted late epidermal invagination and failure of dermal condensation to compact into dermal papillae. n = 3/3, biological replicates. (L). Illustration of disrupted late epidermal invagination and expanded DP formation. Source data are available online for this figure.

Together, these results indicate that follicle base closure is achieved through mechanically and chemically interconnected events that guide cellular flows, tissue folding, and fate specification. Mechanical and molecular cues act in concert to induce shape changes. This process also establishes molecular patterning in distinct topological zones of the proximal follicle, positioning tissues for future cyclic renewal (Wu et al, 2021b).

## From 2D plane to 3D follicle: a mechano-chemical coupling model for topological transformation during follicle formation

Upon dermal condensation (DC) formation, we observed a switch in dermal cell movement accompanied by changes in epidermal cell shape and stiffness. These events were followed by epidermal invagination around the DC, which ceased once the invagination reached the DC base (Fig. 4A). We asked whether the causality of these spatiotemporally entangled events could be explained by a simple mechano-physical principle.

Mechanically, condensed dermal cells at the DC exert traction forces on surrounding tissue. While horizontal or downward-directed forces are counterbalanced by neighboring DCs or dermal resistance, upward-directed forces reach the epidermis and act as downward pulls on epidermal cells. At the same time, dermal-secreted factors such as Tgf-β stimulate epidermal morphogenetic movements and signal surrounding dermal cells to secrete MMP (Moore-Smith et al, 2017), which in turn activates Tgf-β (Kobayashi et al, 2014). Because MMP degrades the basement membrane (BM) (Strzyz, 2019), the combined activity of Tgf-β and MMP produces mechanically weakened regions in the BM near

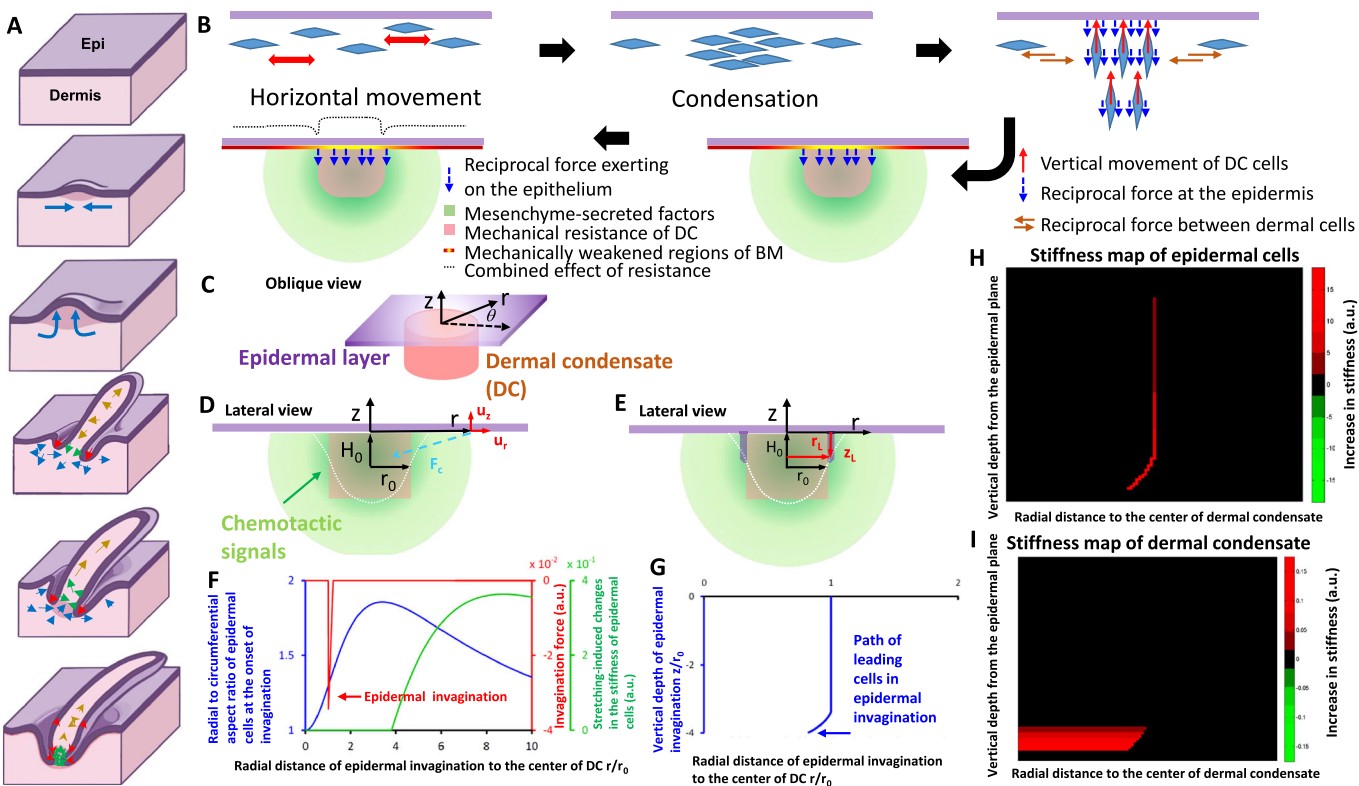

**Figure 4.   A mechano-chemical coupling model of feather follicle formation.**

(A) Morphological transitions of topological transformation are shown in five steps. a, b are studied in Fig. 1. c, d are studied in Fig. 2. e is studied in Fig. 3. Through these interconnected processes, a complex follicle is formed from the planar epithelium. (B) The schematic depicts the model assumptions. Following the horizontal movement, the FGF-stimulated dermal cells form dermal condensates (DCs) and generate migratory/contractile forces therein. While the forces in the horizontal (orange) or downward (red) directions are counteracted by the forces from neighboring condensates or the resistance from the deep dermis, the forces in the upward (red) direction create reciprocal traction forces at the epidermis (blue). Meanwhile, the diffusive Tgf-β-MMP signaling (green) creates a weakening effect (yellow) at the basement membrane (BM), where the epidermal cells experience the largest mechanical resistance (light pink) from the DC. The model aims to investigate whether the combination of BM weakening and DC resistance (dotted gray line) and the diffusive Tgf-β-MMP signaling (green) can: a) induce changes in cell shape and stiffness as observed in the experiment, and b) facilitate epidermal invagination at the periphery of the DC. (C, D) The model simplified the epithelial layers as a flat, planar sheet (Purple) and the dermal condensate as a cylindrical structure (Pink) with a radius denoted as "$r_0$." The chemoattractants (Green) secreted from the dermal cells to induce epidermal invagination were approximated as originating from a point source at a distance denoted as "$H_0$" from the epidermal layers. A cylindrical coordinate system ($r, \theta, z$) with azimuthal symmetry was employed to model the local deformation of cells within the epidermal layers, represented as ($u_r$, 0, $u_z$), during the onset of chemoattractant-induced invagination. (E) The model used the trajectory of epidermal leading cells at the forefront of invagination, denoted as ($r_L, \theta, z_L$) with $\theta$ ranging from 0 to $2\pi$, to depict the progression of invagination along the sidewall of the dermal condensate and the subsequent closure of invagination at the base of the newly formed follicle. (F) The numerical results for panel (C) indicate an increased radial to azimuthal aspect ratio (Blue) and enhanced cell stiffness (Green in panel F) resulting from cell stretching in epidermal cells located distantly from the dermal condensate. The maximal force for invagination was observed near the border of the dermal condensate (Red arrow in panel F). (G–I) The numerical results show (G) the path of the invagination (Blue) and the increase of stiffness represented as heatmaps for (H) the epidermal cells and (I) the dermal cells during invagination. The parameters utilized in the numerical simulations are detailed in the model section.

DCs, facilitating epithelial invagination. We hypothesized that these dermal mechano-chemical activities could explain early changes in epidermal cell shape and stiffness, weak point formation, invagination site specification, and invagination termination. Testing these possibilities experimentally is challenging, so we turned to coarse-grained phenomenological modeling (Fig. 4B).

At the epidermal–dermal interface, we assumed the epidermis receives both traction and chemoattracting forces from the underlying DC (Fig. 4B, blue lines and green marks). Epidermal cells attempting to move downward encounter DC resistance; therefore, only those at the DC periphery move downward, establishing the ring-shaped invagination zone. The highest Tgfβ–MMP signaling is expected near the DC base, guiding epidermal cells toward this target. Initially, epidermal cells are too

distant for chemoattractant forces to deform the DC, so they move along its periphery. As cells migrate toward regions of higher chemoattractant concentration, the increasing force enables them to compress the DC, ultimately closing the invagination.

Changes in stiffness occur through distinct mechanisms in dermal and epidermal cells. Dermal stiffness increases primarily via ECM compression, a process shown to enhance rigidity (Kim et al, 2017; Tronci et al, 2013; Vinciguerra et al, 2014; Wollensak et al, 2003). This compression results from centripetal migration and constriction forces exerted by epidermal cells along the DC boundary, which crowd and narrow the DC and align dermal cells convergently near its base. Epidermal cells, by contrast, stiffen through two processes: (1) elastic stretching within epidermal layers when distant from the DC base, where Tgf-β–MMP signaling

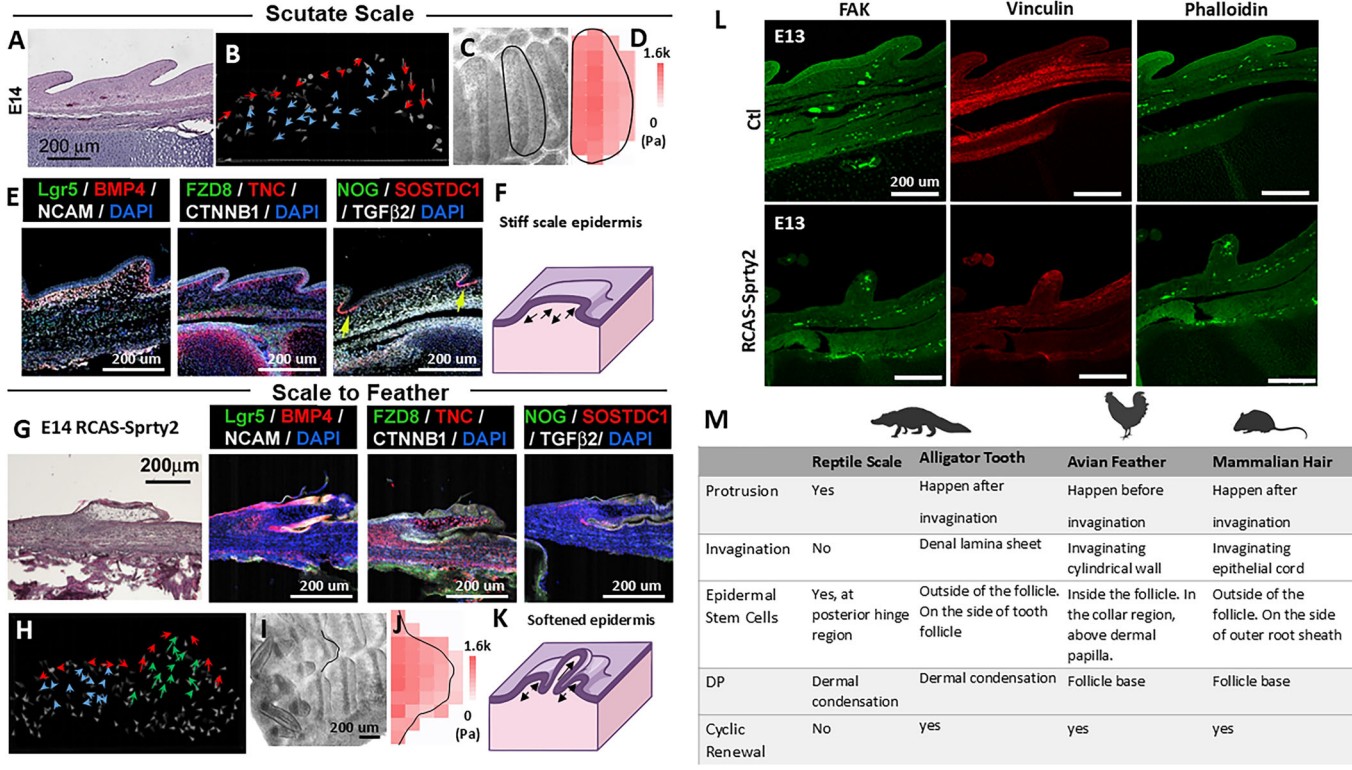

**Figure 5. Scale–feather transition: biophysical events following biochemical signaling alterations redirect morphogenetic consequences.**

(A–F) Biophysical characterization of scutate scales and their formation. (A) H&E of E14 scutate scale. $n = 3/3$, biological replicates. (B) Cell tracking analysis of E10 + 24 h quail scale showing limited dermal cell migration during scale formation. $n = 3/3$, biological replicates. (C) Bright field top-down view of scutate scale in E14 explant. The black line demarcates the scale measured by the AFM. $n = 5/5$, biological replicates. (D) Stiffness heatmap of an E14 scutate scale. $N = 5/5$, biological replicates. (E) Lgr5, BMP4, NCAM, FZD8, TNC, CTNNB1, NOG, SOSTDC1, and Tgf-β2 expression in E14 scale. Yellow arrows: SOSTDC1. The panel is reproduced from Liu et al, Journal of Developmental Biology 2023, 11, 30, Fig. 2H (CC BY 4.0; https://doi.org/10.3390/jdb11030030) (Liu et al, 2023). (F) Illustration of stiff scale epidermis limiting dermal cell flow. (G–L) Scale-feather conversion is accompanied by epidermal softening and redirected dermal cellular flows. (G) H&E section and molecular expressions (shown by RNAscope) of scale to feather conversion by overexpressing Spry2 in the epidermis. $n = 3/3$, biological replicates. (H) Cell flow analysis of newly induced feather buds on top of the scutate scale. It shows dermal cells are redirected to generate vertical flow in the newly formed feather bud. $n = 3/3$, biological replicates. (I) Bright field, top-down view showing scutate scale-to-feather conversion in spry2 overexpressing explant. The black line demarcates the scale measured by the AFM. (J) Stiffness heatmap of an E14 scutate scale-converted feather buds. $n = 5/3$, biological replicates. (K) Illustration of scale to feather transition showing softened epidermis on the scutate scale allows dermal cells to flow vertically and generate feather buds. (L) Representative results of reduced FAK, vinculin, and phalloidin expression in E13 RCAS-Spry2-induced scale to feather skin. Scale bar: 100 um. $n = 3/3$, biological replicates. (M) Comparison of cellular flows during the formation of reptile scale, avian feather buds and mammalian hair primordia. Bud protrusion, follicle invagination, and dermal papillae formation are compared. Source data are available online for this figure.

is low, and (2) signaling-induced stiffening near the DC base, where Tgf-β–MMP activity is high, even without strong chemoattractant gradients (Gladilin et al, 2019).

Using these assumptions, we constructed a model to capture the biophysics of invagination onset and closure. The model focuses on invagination and omits bud protrusion. For invagination onset, we considered the elastic responses of epidermal cells to dermal traction and chemoattracting forces. Using a cylindrical coordinate system (Fig. 4C–E), we analyzed epidermal cell aspect ratio (Fig. 4F, blue curves), invagination initiation sites (Fig. 4F, red curve), and stiffness changes (Fig. 4F, green curve) through a steady-state approach. The aspect ratio increased gradually with radial distance r from the DC center, then returned to baseline near the inter-bud region. Cell stiffness increased away from the DC, reflecting greater stretching in those regions.

For invagination closure, we focused on leading cells at the invagination front (Fig. 4E). These cells exhibited centripetal

movement near the DC base (Fig. 4G). Their stiffness gradually increased as they approached the base (Fig. 4H), while local DC stiffness also rose due to centripetal epidermal closure (Fig. 4I).

Overall, the model describes how dermal mechano-chemical cues distribute forces and deform epidermal cells to initiate, extend, and complete invagination, thereby enabling DP formation. Full methodological details are provided in the Methods and Appendix.

## Scale to feather conversion: softening scale epidermis enables feather follicle formation

Among the many amniote skin appendages (Wu et al, 2004), only a few form follicle configurations that allow cyclic renewal, including hairs (Fuchs, 2009), feathers and alligator teeth (Wu et al, 2013). Scales exhibit epithelial folding and contain epidermal progenitors that support scale growth, but not molting or regeneration. A unique feature of birds is that scutate scales can be experimentally

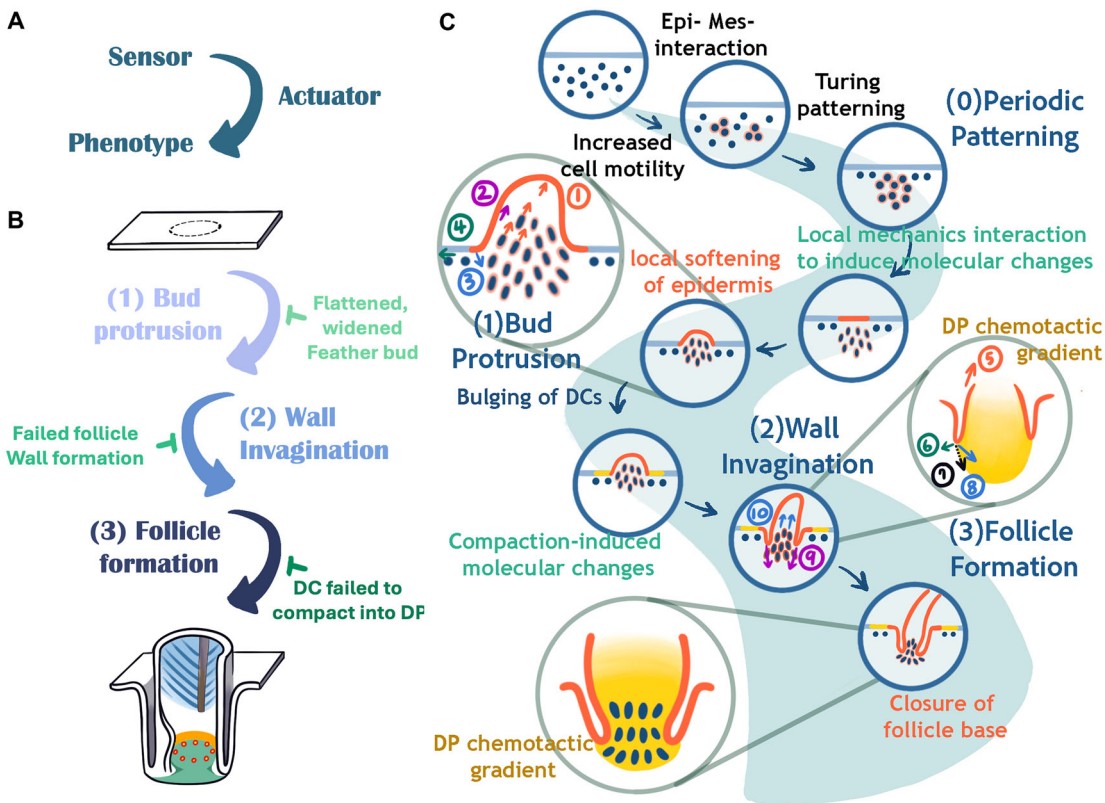

**Figure 6.  Schematic illustration showing a series of self-organizing cellular flows, driven by mechano-chemical coupling, can build the complex topology of feather follicles.**

(A) Self-organizing signaling circuit. The circuit requires a sensor and an actuator to undergo phenotypic changes. This is the core circuit. (B) Multiple self-organizing signaling circuits can be coupled to achieve a more complex morphogenetic process. The coupling happens when the product of one process becomes the initiator of the next process. (C) Here, the formation of a feather follicle is used to illustrate the coupling process. During feather follicle development, an originally homogenous skin tissue undergoes Turing periodic patterning to generate a population of feather primordia. Here, we focus on one primordium. We study how an individual 2D placode is transformed into a 3D follicle architecture. We highlight three stages of morphological transition: (1) bud protrusion, (2) invagination to reach, (3) follicle floor formation. The involved mechanical forces, some are hypothetical, are shown in arrows: (1) Protruding stress, (2) Cell flow-induced shear stress, (3) Chemotactic force, (4) Inter-bud skin tension, (5) Epidermal proliferation dermal cell flow-induced epidermal migration, (6) Elastic resistance force from the DC, (7) Net force, (8) Chemotactic force, (9) Expansion force from cell sheet + ECM-induced migration force, (10) Rebounding force from compacted dermal cells. Epi epidermis, Mes Mesenchyme, DC(s) dermal cell(s), WP weak point.

converted into feathers (Dhouailly, 2009; Widelitz et al, 2000; Wu et al, 2018a; Wu et al, 2018b). Here, we revisited this scale-to-feather transition (Lai et al, 2018; Wu et al, 2018b) from a biomechanical perspective, focusing on cell flows and tissue rigidity.

Scutate scales are characterized by a flat epithelial protrusion with limited posterior elongation (Fig. 5A). Cell tracking during scale development revealed that epidermal cells migrate anterior-to-posterior, while dermal cells show only limited posterior and vertical migration (Fig. 5B; Movie EV4). This suggests that the relatively "soft" region of the scale lies at its posterior end. AFM measurements confirmed that the anterior edge of the scutate scale is 23% stiffer (1632 ± 176 Pa) than the posterior edge (1231 ± 114 Pa, Fig. 5C,D, $n = 3/3$, biological replicates), and that both regions are stiffer than feather buds (845 ± 96 Pa, Fig. 2A, $n = 3/3$, biological replicates). The overall rigidity of scales likely limits dermal cell flow and restricts "stretching-induced" mechanical signaling.

We next examined biochemical circuits that might transduce mechanical forces in scales. Tgf-β2 is expressed in the basal dermis (Fig. 5E), but FZD8, a WNT receptor that can modulate TGF-β signaling through receptor complex formation (Spanjer et al, 2016), is not strongly expressed. Instead, NCAM is enriched at the anterior end of scutate scales, while TNC is expressed in dermal cells at the epidermal–dermal interface (Bao et al, 2016), and SOSTDC1 is restricted to the epidermal "anchors" of the scale, contrasting with its strong expression in DP cells during feather follicle formation. The high levels of NCAM and TNC at the dermal–epidermal interface suggest strong adhesion and tight binding between the two tissues. At the same time, the relatively stiff anterior epidermis likely guides dermal cells posteriorly, where the epidermis is slightly softer and permits limited protrusion (Fig. 5D,F). However, the absence of a signaling center that coordinates biochemical–mechanical crosstalk prevents spatial modulation of stiffness, dermal mobility, and epidermal invagination in scales.

Could altering scale tissue mechanics enable feather follicle formation? Specifically, would softening the scale epidermis via

Spry2 promote this transition? Our previous study (Wu et al, 2018b) showed that localized Spry2 overexpression in scale epidermis can induce scale-to-feather conversion (Fig. 5G). Cell tracking revealed vertical dermal migration resembling that seen in feather bud formation (green arrows, Fig. 5H; Movie EV5). This was accompanied by a 65.6% reduction in local epidermal stiffness (from 1620 to 481 Pa, Fig. 5I,J). We interpret these findings as Spry2-induced softening of the epidermis, which allowed dermal migration to overcome the physical barrier of the rigid epidermis and form feather bud–like structures (Fig. 5K). Consistent with this, Spry2 overexpression downregulated focal adhesion and cytoskeletal molecules in the scale epidermis (Fig. 5L), similar to what we observed in quail feather buds (Fig. EV6D).

By comparison, reptilian scales exhibit short, broad dermal protrusions, minimal epidermal invagination, and no DP or follicle-like structures (Di-Poi and Milinkovitch, 2016). Mammalian hairs, in contrast, form follicles and undergo robust cyclic renewal (Fuchs et al, 2001; Stenn and Paus, 2001). During hair development, hair germs invaginate deeply into the dermis, and the follicle wall wraps around the DP through morphogenetic processes distinct from feathers (Morita and Fujiwara, 2022). Ultimately, hair stem cells are maintained in the bulge region (Li and Tumbar, 2021; Morrison and Spradling, 2008), distinct from the feather collar bulge (Yu et al, 2002). While both hairs and feathers achieve cyclic renewal, their divergent topology and morphogenetic origins highlight convergent evolution of follicle formation strategies (Fig. 5M).

## Discussion

We propose that morphogenetic transition circuits require both a sensor and an actuator to drive phenotypic change (Fig. 6A). The chemo-mechanical coupling events observed in developing systems exemplify such self-organizing sensor–actuator signaling circuits (Fig. 6B). A morphogenetic system senses and responds through this mechanism to achieve each stage of development. Once actuated, it becomes the sensor for the subsequent stage. During feather follicle development, initially homogeneous skin tissue undergoes three transitions: (1) bud protrusion, (2) invagination, and (3) follicle formation (Fig. 6B). Importantly, completion of each stage is required for initiation of the next.

This sequence is summarized in Fig. 6C. Feather follicle morphogenesis involves topological transformations that can be classified into patterning, protrusion, invagination, and follicle floor formation. Each stage is coupled such that the outcome of one becomes the trigger for the next. This interdependence is illustrated in the scale-to-feather conversion, where epidermal softening not only enabled initial bud protrusion but also drove sequential invagination and eventual follicle architecture (Fig. 5).

Stage 1: Patterning. dermal condensation increases stiffness at its center. These condensed cells must displace—either downward into the dermis (as in hair follicles) or upward (as in feathers). In feathers, the placodal epidermis softens and deforms, as shown by AFM measurements and QMorF analysis, allowing dermal cells to actively move vertically upward and push out the feather bud.

Stage 2: Invagination. As the bud elongates, the ring-shaped basal epithelial cells are passively stretched, evidenced by changes in cell shape and YAP and MMP expression. This stretching drives the surrounding epithelium, organized in a cylindrical wall, to invaginate into the dermis.

Stage 3: Follicle floor formation. Continued invagination of the cylindrical follicle wall and enclosed dermal condensates leads to additional changes (Fig. 3A–E). Dermal condensation cells begin expressing SMA, arrange in circular patterns (Fig. 3A,G,H), stiffen (Fig. 3F), and recruit adjacent fibroblasts to express YAP (Fig. 3H). Elevated Tgf-β expression attracts the invaginating epithelial wall toward the dermal papilla, resulting in dermal papilla formation and sealing of the follicle base.

Our prior work showed that optimal tissue stiffness is required for regenerative wound healing (Harn et al, 2019; Harn et al, 2021). A soft dermis permits stiff epidermal hair placodes to invaginate and form hair germs, while proper epidermal stiffness is also necessary for hair follicle development. By analogy, feather development requires an optimal stiffness balance between epidermis and dermis at each stage. Specifically, stiffness contrasts—force imbalances between the two tissues—drive folding, cell flows, and morphogenesis. At E7, actively dividing and migrating dermal cells generate stochastic events that enhance adhesion and promote dermal condensation. This increases DC stiffness from ~175 to ~212 Pa, enabling local mechanical interactions with the epidermis, reflected in epidermal cell orientation, Shh/Slug activation, and epidermal softening (<196 Pa). These changes allow dermal cells to bulge upward, forming the protruding feather bud. The softening factor Snail, downstream of Shh (Riaz et al, 2019), explains the colocalized expression of Shh and Snai1 (Fig. 3A). Since Snai1 downregulates epidermal junctions (Cano et al, 2000), its expression plausibly accounts for the observed reduction in epidermal stiffness.

On the dermal side, FGF-induced "solidification" of the DC core, coupled with BMP-dependent contraction of surrounding cells, has been proposed to drive bud protrusion (Yang et al, 2023).

Here, we focus on the transition from DC to short feather buds (with a broad base), whereas elongation into long buds depends on Shh- and gap junction–mediated calcium signaling and dermal migration (Li et al, 2018). As the bud continues to elongate, dermal flows not only sustain growth but also impose shear stress that stretches the basal epidermis (Fig. 6). This activates YAP and its downstream targets—proliferation, motility, and MMP production—leading to epidermal stiffening (~1000 Pa). The stiffened epidermis buckles into the softened dermis (~500 Pa), initiating invagination, which is then extended by chemoattractant-mediated signaling. YAP, a canonical mechanosensory (Panciera et al, 2017), is active at invaginating epithelial tongues, consistent with its role in stiffening and remodeling. YAP also induces MMP expression (Lei et al, 2023), further softening the invagination zone (Jiang et al, 2011). Together, these molecules create concentric ring patterns (Fig. 1I), reminiscent of Fujiwara's "telescope model" (Morita et al, 2021). We propose that these ring-like domains establish mechanical contrasts that guide dermal cell flows into the invaginating region.

Beyond feather follicles, the scutate scale study underscores the importance of tissue folding and compartmentalized flows in fate specification. Limited dermal mobility confines scale morphogenesis within a rigid epidermal compartment (Fig. 5B). By contrast, Spry2 overexpression softens the epidermis, enabling dermal flows, protrusion, and activation of sequential circuits that form ectopic feather follicles (Fig. 5G–K). The ability of both avian feathers and mammalian hairs to form dermal papillae—via epidermal

invagination that envelops specified dermal cells—illustrates convergent evolution toward stem cell–based follicles, conferring cyclic renewal and adaptive versatility.

We therefore propose the concept of a "mechanical signaling organizer" operating in parallel with molecular organizers. A molecular signaling center typically functions through localized morphogen diffusion, establishing gradients that regulate cell fate. By analogy, a mechanical organizer is defined by localized stiffness contrasts, capable of transmitting forces through tissue deformation or channel activity to influence distant regions. In this framework, dermal condensation, the feather bud tip, and the dermal papilla act as mechanical organizers. Each redirects cell flows, enabling sequential morphogenetic events and the emergence of complex architectures.

In summary, our findings reveal that dynamic cell flows beneath the skin play underappreciated roles in development, regeneration, and organoid morphogenesis (Lei et al, 2023). Each morphogenetic transition requires both a sensor and an actuator to initiate and complete the process. While diverse amniote scales evolved distinct configurations, none acquired follicular architecture capable of stem cell–based renewal. Here, we show that altering tissue mechanics can generate novel mechanical organizers that, together with molecular organizers, redirect flows and couple morphogenetic stages. Once optimal conditions are established, the completion of one stage reliably initiates the next, exemplified by the three consecutive stages of feather follicle formation (Fig. 6A–C). From feathered dinosaurs to early birds, this follicular design enabled evolutionary experimentation with diverse feather forms across regions and life stages, enhancing integumentary adaptability. We further propose that newly induced mechanical organizers—during molting or wound repair—may similarly redirect flows to achieve repair or regeneration of the integumentary organ.

## Methods

### Reagents and tools table

| Reagent/resource | Reference or source | Identifier or catalog number |
|---|---|---|
| **Experimental models** | | |
| White Leghorn eggs | SPAFAS, Preston, CT | |
| Membrane-GFP transgenic quails | Huss et al, Development, 2015 | |
| **Recombinant DNA** | | |
| FRET-MMP14 | Chung et al, 2015 | |
| acta2_GFP_pA_pDestTolpA2 | Addgene | 78684 |
| FURW | Addgene | 14883 |
| RCAS-H2B-mOrange | Bronner lab. Tang et al, 2019 | |
| RCAS-Wnt3a | Tabin Lab. Kengaku et al, 1998 | |
| RCAS-Pry2 | Martin Laboratory. Minowada et al, 1999 | |
| RCAS-dnYAP1 | Wu et al, 2025 | |
| **Antibodies** | | |
| E-cadherin | Proteintech | 20874-1-AP |

| Reagent/resource | Reference or source | Identifier or catalog number |
|---|---|---|
| YAP1 | Proteintech | 13584-1-AP |
| MMP2 | Proteintech | 10373-2-AP |
| SMA | Proteintech | 14395-1-AP |
| FAK | Proteintech | 12636-a-AP |
| Vinculin | Proteintech | 26520-1-AP |
| **Oligonucleotides and other sequence-based reagents** | | |
| RCAS-Snai1 primers | This study | Materials and methods |
| DKK1 primers | This study | Materials and methods |
| SOSTDC1 primers | This study | Materials and methods |
| Snai2 probe | Addgene | 14015 |
| Shh probe | Tabin Lab. Roberts et al, 1995 | |
| Gremlin probe | Capdevila et al, 1999 | |
| BMP4 | Advanced Cell Diagnostics | 558411-C2 |
| CTNNB1 | Advanced Cell Diagnostics | 458711-C3 |
| FZD8 | Advanced Cell Diagnostics | 1055381-C1 |
| LGR5 | Advanced Cell Diagnostics | 480781-C1 |
| NCAM1 | Advanced Cell Diagnostics | 1055371-C3 |
| NOGGIN | Advanced Cell Diagnostics | 480101-C1 |
| SOSTDC1 | Advanced Cell Diagnostics | 1055361-C2 |
| TNC | Advanced Cell Diagnostics | 1055441-C2 |
| TGFB2 | Advanced Cell Diagnostics | 1055431-C3 |
| **Chemicals, enzymes and other reagents** | | |
| Blebbistatin | Cayman | 13186 |
| ADH-1 | Cayman | 29981 |
| LY2109761 | Cayman | 15409 |
| **Software** | | |
| Imaris | Oxford Instruments https://imaris.oxinst.com/ | |
| Leica LAS X | Leica https://www.leica-microsystems.com/products/microscope-software/p/leica-las-x-ls/ | |
| **Other** | | |
| Phalloidin-488 | Proteintech | PF00001 |
| DAPI | Thermo Fisher | D1306 |
| RNAscope Multiplex Fluorescent v2 System | Advanced Cell Diagnostics | 323100 |

## Avian eggs

Chicken embryos were harvested from White Leghorn pathogen-free fertilized eggs (SPAFAS, Preston, CT). The membrane-GFP

transgenic quail eggs were gifted from the Lansford Lab (Huss et al, 2015). Embryos of both sexes were included in the study randomly, as it is very difficult to determine the sex of the embryos at such an early stage.

An avian skin explant model was performed according to Jiang et al (Jiang et al, 2023). In brief, skin tissues were harvested from avian embryos and cultured on PET track-etched 0.4-um pore size culture-insert mesh (Corning, NY, USA) in six-well plates. About 1 ml culture medium of DMEM, 10% FBS, 2% chicken serum, and 1% pen/strep antibiotics were added to the outside of the insert, creating a fluid-air interface for the explant. The tissue is incubated at 37 °C, 5% $CO_2$, and 90% humidity.

## Atomic force microscopy, data analysis, and stiffness heatmap

The AFM (NanoWizard 4a/CellHesion, JPK, Berlin, Germany) combined with a stereomicroscope was set up in contact mode indentation to measure tissues in PBS. The stereomicroscope is aligned with the cantilever and sample, so the location of indentation can be precisely observed. SAA-SPH-5UM AFM cantilevers (Bruker, MA, USA) of cylindrical tip with 5-um end radius and spring constant between 0.1 and 0.2 N/m. The spring constants of all cantilevers were calibrated via the thermal noise method with a correction factor in liquid (Butt and Jaschke, 1995; Hutter and Bechhoefer, 1993) prior to each measurement. A force series identified a maximum indentation force of 5 nN to show the most consistent results on test samples. A constant rate of 1 μm/s was used for the entire approach and retract sequence. Force-distance curves were collected and post-processed using the JPK package software (Data Processing, 6.3.11). The force curves were analyzed using the Hertz model with a spherical indentation (Lin et al, 2009; Tripathy and Berger, 2009).

The force on the cantilever F(h) is given by:

$$F(h) = \frac{E\text{sample}}{1 - v^2\text{sample}} \frac{4\sqrt{R}}{3} h^{3/2}$$

where h is the depth of the indentation, $E$ is the effective modulus of the system tip-sample, $v$ is the Poisson ratio for the sample, and $R$ is the radius of the AFM tip. The unit of Young's modulus is calculated as N/m², and expressed as pascal (Pa) or kilopascal (kPa). Poisson ratio was set at 0.5 since the spherical tip was incompressible relative to the sample. The temperature of the measurement was controlled at 37 °C to mimic cultured conditions.

For force curves obtained from the epidermis, an additional Gaussian smoothing step (smoothing width = 3.00) and baseline adjustments (Offset + Tilt) were implemented prior to fitting.

Data analysis and heatmap generation were done in reference to Harn et al (Harn et al, 2021). At least five indentation points were taken for each region of interest, and at least three biological replicates were measured for each condition. The data from the same biological locations (e.g., dermal condensate, epidermal invagination site, and epidermal protrusion) were averaged and used to plot the eventual stiffness heatmap.

The interpolation of tissue stiffness was performed by using the 3D meshgrid function of MATLAB (R2015b, Natick, MA, USA). After obtaining a Young's modulus (z) at a specific spatial location

(x, y) in the wound, a three-dimensional matrix was defined. When the positions and stiffness of all the measured spots were identified, we could interpolate the stiffness of the positions in between to average the stiffness of the nearest parameters using the 3D meshgrid function, by defining (x, y) as meshgrid and (z) as griddata. In the end, the heatmap was generated by defining the representative color of stiffness.

### Accounting for surface curvature effects in AFM measurements

We measured the curvature of the E8 feather bud using its longitudinal H&E image, and calculated the ratio of average bud tips radius (R_sample, ~50–80 um, $n = 5$) to AFM tip radius (R_tip = 5 μm), and ensure the ratio is <0.1. We also maintained maximum indentation depths of <1 μm (controlled by 5 nN force limit), ensuring indent depth is much less than both R_tip and R_sample, minimizing geometric artifacts.

We also measured the tissue angle of the E8 bud vs the adjacent interbud region, and it averaged to be less than 34 degrees (~33.4 degrees, $n = 5$), which would minimize the underestimated effects of using a spherical tip on a tilted (curved) biological substrate (Ahmine et al, 2024).

### Sample preparation for AFM measurement
Unprocessed sample. Skin explants were harvested, placed on culture-insert meshes, and incubated under standard culture conditions for at least 2 h until firmly attached. Immediately before AFM measurement, the mesh was cut out with a surgical blade and mounted on a glass coverslip compatible with the JPK temperature control system.

Separated dermis/epidermis. Skin explants were harvested and incubated in calcium–magnesium-free (2× CMF) solution containing 0.25% EDTA for 15–25 min on ice. Epidermis and dermis were then mechanically separated under a dissecting microscope. The tissues were laid flat on a glass coverslip and loaded onto the AFM.

Longitudinal sections. Skin explants were sectioned longitudinally using a surgical blade. Sections were placed on their side on a culture-insert mesh, and the epidermal–dermal interface was visualized by phase-contrast stereomicroscopy. The mesh was then cut out and transferred to a glass coverslip for AFM measurement, allowing the cantilever to contact the cross-section of feather bud structures.

### Live imaging and cell tracking analysis
Imaging. Live xyzt imaging was performed on a confocal microscope (Stellaris 5, Leica) using 10× air objectives. Z-stacks (50 μm) were collected at 10-min intervals for 18–48 h. Laser power, gain, and exposure time were kept minimal and constant to prevent photobleaching. Samples were maintained at 37 °C, 5% $CO_2$, and 90% relative humidity in an Oko-Lab chamber.

Tracking. Cell migration analysis was conducted using Imaris (Version 9.2, Oxford Instruments). Drift correction was applied using the Correct Drift function, based on reference time points and structural alignment across the time series. Background subtraction was performed with a rolling ball algorithm (radius = 10–12 μm) to improve signal-to-noise ratio.

Approximately 20,000–25,000 cells were tracked and analyzed per sample. Particles were detected using the Spots function in Imaris,

based on nuclear RCAS-mOrange signal. All analyses were performed on particle trajectories. For each tracked particle (cell nucleus or centroid), we extracted 3D displacement vectors and positions (X, Y, Z) over time, from which displacement magnitude and direction were computed as described below:

Three-dimensional vectors:

$$\vec{r} = [r_x, \, r_y, r_z]$$

Length of a vector:

$$|\vec{r}| = \sqrt{r_x^2 + r_y^2 + r_z^2}$$

Cell–cell displacement x,y,z, where the position of an object at time index is $\vec{p}(t)$, can be calculated by subtracting the first time point position (1) from a selected time point position.

$$\vec{d}(t) = \vec{p}(t) - \vec{p}(1)$$

Hence, the object's track displacement can be calculated as:

$$\vec{tdl} = \vec{p}(n) - \vec{p}(1)$$

Local spatial organization was quantified by calculating the average nearest neighbor distance for each tracked particle (cell) at every time point. This provided a proxy for local density changes and spacing uniformity, which reflects spreading and compression behaviors. Results were plotted with time on the x-axis for each sample.

The *dragon tail* function in Imaris was used to visualize particle trajectories. Particles were color-coded according to their position and cell type (e.g., epidermis, dermis, or dermal papilla–forming cells).

Cells in different regions of the feather bud were manually selected and grouped by color based on their spatial position. Migration behaviors of each group (e.g., dermal cells and invaginating epidermal cells) were then analyzed independently.

Imaris uses object-based tracking of individual nuclear centroids. Nuclei were tracked using the Autoregressive Motion algorithm, which models object displacement based on motion in the immediately preceding time point. Each nucleus is designated as a particle for tracking. In bud formation, invaginations, DP and scutate scale videos, the mean particle diameter was estimated to be 14.8, 22.7, 22.7, and 9.10 µm, while the maximum allowed inter-frame displacement was set to 8.35, 19.3, 15.7, and 6.15 µm, respectively. These parameters minimize erroneous linking between neighboring nuclei. The maximum gap size allowed between frames were set to 3, and gap filling was disabled to avoid artificial interpolation of trajectories. Tracking results were manually inspected to confirm continuity of motion and absence of identity switching.

Cell division events were measured using Imaris' lineage tracking function, focusing on dermal cells within the protruding feather bud. The cell division events were divided over the final cell number within the region of interest.

### Viral production and transfection

Lentivirus was made in reference to Harn et al (Harn et al, 2021). Plasmid for the reporter smooth muscle actin (ACTA2) was purchased from Addgene (acta2_GFP_pA_pDestTolpA2 (Cat# 78684). FURW (flap-Ub promoter-RFP-WRE) plasmids (Cat. #14883) were made by replacing the EGFP vector with that of RFP. To co-transfect the embryo, the two viruses were mixed at a 1:1 ratio and injected into the chicken embryo somite at E3.

RCAS injection is performed on an E3 embryo to transfect the epidermis, dermis or both, the virus is injected into the amniotic sac, dermis, or both.

RCAS-H2B-mOrange is a gift from the Bronner lab (Tang et al, 2019).

To generate RCAS-Snail1, the full-length chicken Snail1 coding sequence was amplified by PCR using cDNA from embryonic day 8 (E8) chicken skin as the template. The primers used were:

Forward –GGGACAAGTTTGTACAAAAAAGCAGGCTTCAC CATGCCGCGCTCGTTCCTGGT, Reverse – GGGGACCACT TTGTACAAGAAAGCTGGGTCTCAGCGTGCCCCTGAGCAG.

The RCAS-Snail1 construct was assembled using BP and LR recombination reactions (Invitrogen, Gateway® Technology with Clonase™ II) according to the protocol (Loftus et al, 2001).

RCAS-Wnt3a was kindly provided by the Tabin laboratory (Kengaku et al, 1998), RCAS-Sprouty2 (Spry2) by the Martin laboratory (Minowada et al, 1999), and RCAS-dnYAP1 by Wu et al (Wu et al, 2025).

RCAS-control is prepared with the same backbone used for other RCAS clones, as used in Wu et al (Wu et al, 2018b).

### Laser ablation assay

Membrane-GFP-labeled transgenic quails were used for laser ablation experiments on skin explants. Skin explants were harvested and cultured on culture-inserts for at least 2 h as described above. Immediately prior to ablation, the mesh was removed, and the tissue was placed in a 35 mm glass-bottom culture dish (MatTek Life Sciences, MA, USA) with feather buds facing the glass. Approximately 100 µL of medium was added to the explant, and the dish was loaded onto the microscope, which was equipped with a temperature-controlled system maintained at 37 °C.

Laser ablation was performed using a Leica TCS SP8 confocal microscope (Wetzlar, Germany) with a 63X oil immersion lens, following previously described procedures (Liang et al, 2016). GFP fluorescence was used to capture an overview of the feather bud and identify the region of interest (ROI) for ablation. The ROI was then zoomed in at 20X, and the point bleach function was used to create damage using VIS, UV, or IR pulsed lasers. Laser intensity was set to 99%, and ablation duration was optimized between 20 and 60 s. Post-ablation imaging was recorded at 10 s per frame with 2% laser intensity for 5 min to monitor changes in gap distance. Gap distances were quantified using Leica LAS X software (Wetzlar, Germany).

### In situ hybridization and RNAscope

To generate RNA probes for in situ hybridization, polymerase chain reaction (PCR) was performed using cDNA prepared from embryonic day 8 chicken skin. The following primers were used: DKK1, forward 5'-gtgaggagggcgacttctg-3' and reverse 5'-

gaaattaatacgactcactatagggaaactcagcgcgtaccac-3'; SOSTDC1, forward 5'-tctctccgccattcacttct-3' and reverse 5'-gaaattaatacgactcacta- taggggacaggctttgcttgagagg-3'. Antisense RNA probes were synthesized using T7 RNA polymerase (Roche, Switzerland). Plasmids for SHH and Gremlin probes were kindly provided by Dr. Cliff Tabin (Roberts et al, 1995) and Capdevila et al (Capdevila et al, 1999), respectively. The plasmid for the Snail2 probe was obtained from Addgene (Plasmid #14015), originally generated by the Tabin laboratory. Section and whole mount in situ hybridization were performed as described in Wu et al (Wu et al, 2018b).

RNAscope was performed using the Multiplex Fluorescent v2 system (Advanced Cell Diagnostics, Cat. #323100), following the manufacturer's standard protocol. The following probes were used: BMP4 (Cat. #558411-C2), CTNNB1 (Cat. #458711-C3), FZD8 (Cat. #1055381-C1), LGR5 (Cat. #480781-C1), NCAM1 (Cat. #1055371- C3), NOGGIN (NOG, Cat. #480101-C1), SOSTDC1 (Cat. #1055361-C2), TNC (Cat. #1055441-C2), and TGFB2 (Cat. #1055431-C3).

## Inhibitor treatment

Blebbistatin (Cat# 13186, Cayman, MI, USA), ADH-1 (Cat# 29981, Cayman, MI, USA), and LY2109761 (Cat# 15409, Cayman, MI, USA) were dissolved in DMSO and added to the medium to reach the desired concentration.

## IHC and antibody

All the antibodies purchased were unconjugated and were performed according to standard IHC protocol at a 1:50 titer, 4 °C hybridization overnight, and 1 h, room temperature for secondary hybridization. Phalloidin and DAPI were added in the secondary hybridization step at a 1:1000 titer. E-cadherin (Cat# 20874-1-AP, Proteintech), YAP1 (Cat# 13584-1-AP, Proteintech), MMP2 (Cat# 10373-2-AP, Proteintech), SMA (Cat# 14395-1-AP, Proteintech), FAK (Cat# 12636-a-AP, Proteintech), vinculin (Cat# 26520-1-AP, Proteintech), Phalloidin-488 (Cat# PF00001, Proteintech), DAPI (Cat# D1306, Thermo Fisher).

## FRET-biosensor, imaging and analysis

MMP14-FRET-biosensor (Chung et al, 2015) is a kind gift from Yingxiao Wang's lab. The lentivirus-backboned (FUGW, plasmid #14883, Addgene) MMP14-FRET were cloned by replacing the EGFP segment with the MMP14-FRET vector. The biosensor consists of CFP and YFP separated by an MMP14-cleavable peptide linker. MMP14 protease activity cleaves the linker, separating the fluorophores and reducing FRET efficiency. The virus was generated as described in Harn et al (Harn et al, 2021). The virus was injected into the chicken somite at E3. The explant was harvested on the desired embryonic day and harvested on the culture insert for imaging.

Imaging protocol is adopted from Chung e al (Chung et al, 2015). Briefly, the Leica Stellaris 5 confocal microscope with a 10× air objective was used, with an incubation chamber set at 37 °C. Laser excitation was set at 453 nm for CFP excitation, and emission channels were set at 475-500 nm for CFP (donor) and 545–570 nm for YFP (acceptor). Laser power was set at 5% to minimize photobleaching. Z-stacks were set at 50-μm depth, 5-μm step size.

FRET ratio was calculated as:

$$\text{FRET ratio} = I\_YFP/(I\_CFP + I\_YFP)$$

where I_CFP and I_YFP are background-subtracted intensities in respective channels. Background was measured from regions outside the tissue and subtracted from all images. FRET ratio values were normalized to control (apteric) regions, which show minimal MMP activity (Fig. EV7C, FRET ratio = 0.1–0.5).

### Statistical analysis

FRET ratio heatmaps represent the average of three feather buds from three biological replicates ($n = 3/3$). Regions with FRET ratio >1.2 (>fourfold above apteric control) were classified as MMP- positive. Statistical comparisons used a t-test comparing bud base and apteric regions.

Statistics. Number and nature of the replicates are indicated as ($n =\_/\_$). The data were presented as mean ± SD unless stated otherwise. Two independent sample T-tests (two-tailed) were used for comparing unpaired sample groups. One-way or two-way ANOVA with post hoc Tukey's were used where appropriate. Results with $p < 0.05$ were considered significant. *$p < 0.05$. **$p < 0.01$. ***$p < 0.005$, ****$p < 0.001$, or as indicated in the figure legend.

Phenomenological model. To model the initiation of invagination, a cylindrical coordinate system ($r, \theta, z$) with azimuthal symmetry is applied to the system, with the epidermal layer located at $z = 0$ and the DC located underneath the origin (Fig. 4C, Oblique view). For simplicity, we neglected the proliferation of epidermal cells at this stage, which could potentially lead to epithelial buckling and undulation. Since the model focused on invagination, we also neglected the upward deformation of the epidermal layer caused by the protrusion of the DC. The region with the highest concentration of chemoattractant Tgfβ is represented by a point $Q$ located at $z = -H_0$, and the DC is approximated by a cylinder with a radius of $r_0$. Due to the synergistic effect between MMP and Tgfβ, the spatial profile of MMP activity was assumed to correlate with the profile of Tgfβ. The deformation of epidermal cells at the location $r$ is denoted as $u_r(r)$ and $u_z(r)$ along the radial and the $z$ directions, respectively, while $F_c$ represents the chemotactic force acting on the epidermal cells from the dermis (Fig. 4C, Lateral view). The spatial profile of Tgfβ/MMP was approximated by the steady-state solution. The gradient of this solution was used to indicate the force (refer to Equations (S.1–S.3) in the SI). The aspect ratio of the radial length to the azimuthal length of the epidermal cells at the location $r$, denoted as $AS(r)$, is expressed as follows (refer to Equation (S.4) in the SI),

$$AS(r) = (1 + \frac{du_r(r)}{dr})/(1 + \frac{u_r(r)}{r}) \quad \text{(M.1)}$$

while the changes in cell stiffness were assumed to be linearly proportional to the increase in cell length due to stretching (refer to Equation (S.5) in the SI). The steady-state profiles of $u_r(r)$ and $u_z(r)$ was numerically estimated using Lamé's constitutive equation (Sadd, 2014) (refer to Equations (S.9–S.12) in the SI). The results were used to calculate the aspect ratio (Fig. 4E, blue curves) and the changes in

cell stiffness (Fig. 4E, green curves). To model the invagination of epidermal cells into the dermis, we considered the combined effect of mechanical resistance from the DC and the mechanical weakening of the basement membrane (BM) by MMP (Fig. 4B). Due to the diffusion of MMP, epidermal cells on top of the DC encounter direct resistance, whereas epidermal cells at the border of the DC experience a reduction in resistance (Fig. 4B). Only when the force is larger than the resistance, the epidermal cells are allowed to deform in the $z$-direction. Using this approach, we numerically estimated the effective force for the invagination (Fig. 4E, red curve; refer to Equation (S.13) in the SI).

To model the termination of invagination, we tracked the position of leading cells at the forefront of invagination. The position is defined as $r_L(t)$ and $z_L(t)$ for the radial and $z$ coordinates, respectively, while the angular coordinate was ignored due to azimuthal symmetry (Fig. 4D). Since these cells are approaching the region with the highest concentration of Tgfβ, we assumed that the major force acting at these cells is the chemotactic force and the resistance from the dermis (refer to Equations (S.15) and (S.17) in the SI). We also assumed that these cells change stiffness mainly in response to the high concentrations of Tgfβ. For simplicity, we assumed that the increase in cell stiffness of these cells is directly proportional to the concentration of Tgfβ in their immediate vicinity (see Equation (S.2) in the SI). Furthermore, we considered the nonlinear effect in the compressive stress-strain relations of the dermis, given that the migratory force from the leading cells is significant. This nonlinear effect indicates an increase in the stiffness of the dermis (refer to Equation (S.16) in the SI), denoted as $\Delta\lambda_D$:

$$\Delta\lambda_D(r \leq r_L(t), z_L(t)) = \lambda_{Din1}(r_0 - r_L(t)) \qquad \text{(M.2)}$$

where $\lambda_{Din1}$ represents the coefficient of the nonlinear compressive effect of the dermis. Using these approaches, we numerically obtained the path of invagination (Fig. 4F), the changes in the stiffness of the leading cells (Fig. 4G), and the changes in the stiffness of the DC (Fig. 4G) (refer to the SI for more details).

## Data availability

This study includes no data deposited in external repositories.

The source data of this paper are collected in the following database record: biostudies:S-SCDT-10_1038-S44318-026-00771-7.

## Peer review information

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

## Acknowledgements

This work was supported by National Institutes of Health (NIH) grants R37 AR060306, R35GM153402, RO1 AR078050 (C.M. Chuong), and RO1 AR047364 (P. Wu and C.M. Chuong), R01 HL121365 and R35 GM140929 (Y Wang), and the research contract between USC and China Medical University in Taiwan, contract number 005884. WTJ would like to thank the services provided by the two-photon imaging facility, China Medical University Hospital, Taiwan. WTJ was supported by the National Science and Technology Council, Taiwan (NSTC 112-2112-M-039-001 and NSTC 112-2811-M-039-002), and China Medical University, Taiwan (CMU111-MF-18 and CMU112-MF-01). CLG is funded by NSTC 112-2112-M-001-069, NSTC 112-2314-B-001-001, NSTC 113-2327-B-038-001, NSTC 113-2112-M-001-038, NSTC 114-2112-M-001-020, AS-GCS-112-M01, GCS-112-M01, GCS-112-M05, and GCS-114-M09. TYL is supported by the Dragon Gate Program (112-2926-I-006-507-G) in Taiwan. We acknowledge the use of AFM at the Nano and Pico Characterization Lab in the California NanoSystems Institute at UCLA. Figure 5E contains an image reproduced from our previous publication (Fig. 2H in Liu et al, Journal of Developmental Biology, 2023) (Liu et al, 2023), which is published under the Creative Commons Attribution License (CC BY 4.0).

## Author contributions

**Hans I-Chen Harn**: Conceptualization; Data curation; Formal analysis; Validation; Investigation; Visualization; Methodology; Writing—original draft. **Ting-Xin Jiang**: Data curation; Validation. **Chih-Han Huang**: Data curation; Software; Formal analysis; Validation; Investigation. **Wen-Tau Juan**: Data curation; Formal analysis. **Tzu-Yu Liu**: Data curation; Visualization; Methodology. **Tsao-Chi Chuang**: Formal analysis. **Wan-Chi Liao**: Formal analysis. **Yingxiao Wang**: Resources. **Ji Li**: Data curation. **Cornelis J Weijer**: Writing—review and editing. **Ping Wu**: Conceptualization; Data curation; Formal analysis; Validation; Investigation; Methodology. **Chin-Lin Guo**: Conceptualization; Software; Formal analysis; Methodology; Writing—original draft. **Cheng-Ming Chuong**: Conceptualization; Resources; Supervision; Funding acquisition; Investigation; Writing—review and editing.

Source data underlying figure panels in this paper may have individual authorship assigned. Where available, figure panel/source data authorship is listed in the following database record: biostudies:S-SCDT-10_1038-S44318-026-00771-7.

## Disclosure and competing interests statement

The authors declare no competing interests.

# Expanded View Figures

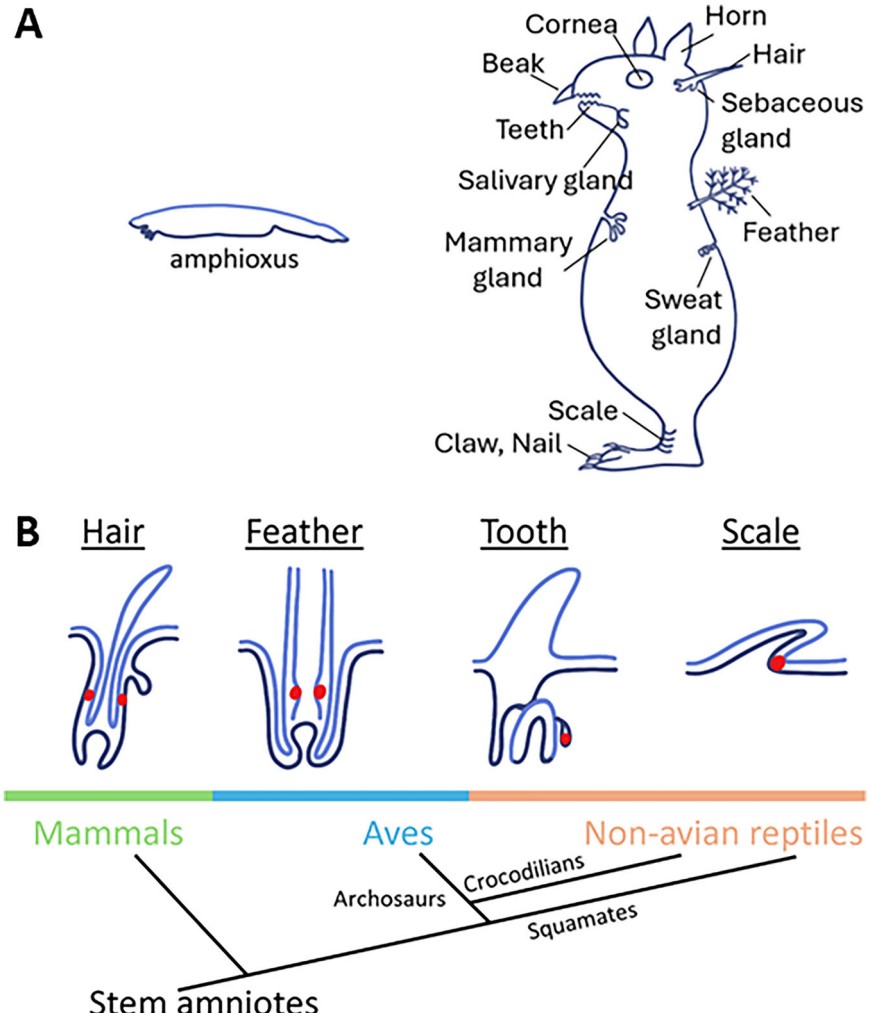

**Figure EV1. Evo-devo of integumentary organs.**

(A) Ancestral chordates such as amphioxus have a smooth integument. As vertebrates evolve, different types of skin appendages emerge to help animals interact with the environment. Shown here is a conceptual animal with different integumentary organs. Modified from Chuong CM edit, 1998. Molecular Basis of Epithelial Appendage Morphogenesis. Landes Biosciences. (B) Among these integumentary appendages, the follicle architecture provides epidermal stem cells (red dots) with a dermal niche, facilitating the molting of the distal differentiated structures and renewal of new appendages. Follicles in hair, feathers and reptile teeth result from convergent evolution. Scales do not have this configuration and keep homeostasis similar to the mechanism used in the epidermis.

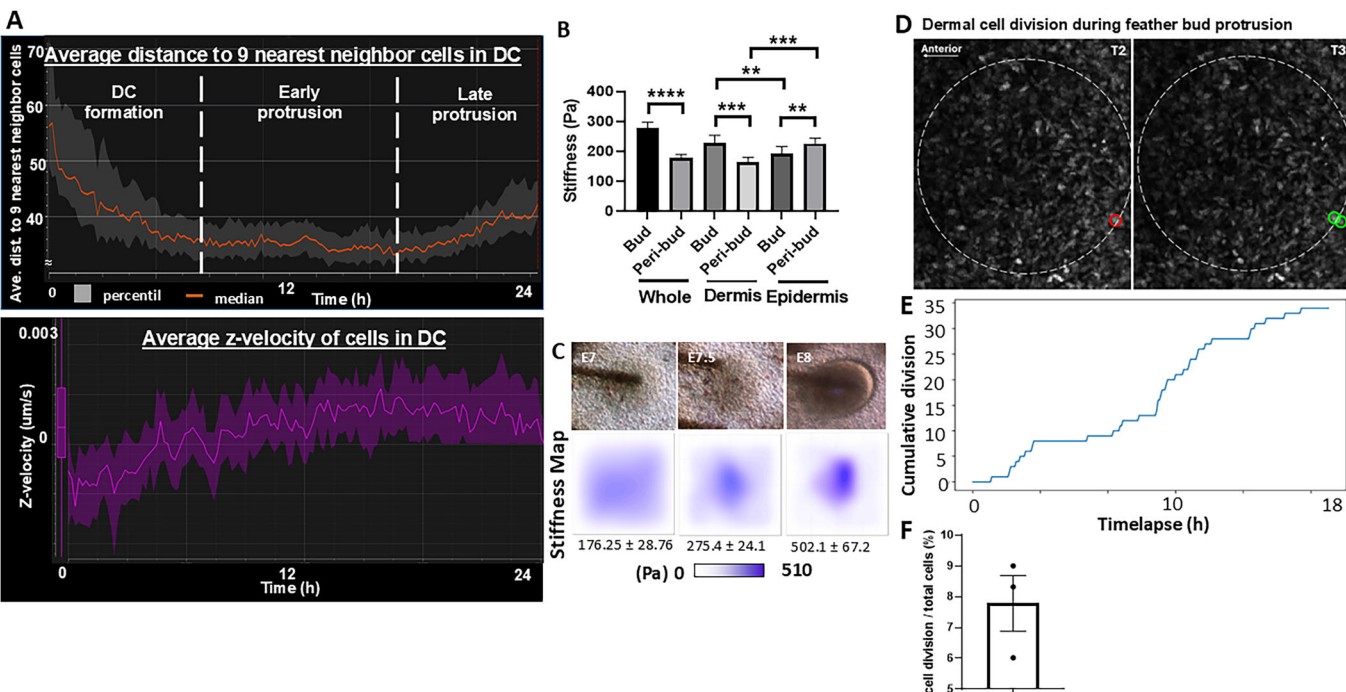

**Figure EV2. Biophysical characterization of feather bud formation.**

(A) Top panel, average distance to nine nearest neighbor cells. Lower panel, average z-velocity in DC during feather bud protrusion. Representative results. n = 3/3, biological replicates. (B) Average tissue stiffness of whole, dermis and epidermis of bud and peribud regions of E8 skin. Two-way ANOVA, Whole, bud vs peribud: F (1, 8) = 304.7 ****, <0.0001. Dermis, bud vs peribud: F (1, 8) = 35.79 ***P = 0.0003. Epidermis, bud vs peribud: F (1, 8) = 37.66 **P = 0.0031. Dermis bud vs epi bud, F (1, 8) = 11.47, **P = 0.0095. Dermis peribud vs epidermis peribud: F (1, 8) = 45.76, ***P = 0.0001. n = 9/5. Nine measurements from five biological replicates. (C) Stiffness map of E7, E7.5, and E8 feather buds showing during dermal condensation and feather bud protrusion, the regional stiffness gradually increases from 176 ± 28 Pa to 275 ± 24 and 502 ± 67 Pa, respectively. n = 3/3. (D) Representative photos of detected cell division at the posterior end of the feather bud during its protrusion at T2 and T3 of E8 + 18 h video. The arrow indicates the anterior end of the feather bud. n = 3/3, biological replicates. (E) Representative graph of cumulative cell division events observed in 511 dermal nuclei within the feather bud. n = 3/3, biological replicates. (F) Average percentage of cell division events over total cells in three E8 + 18 h feather buds. N = 3/3, biological replicates.

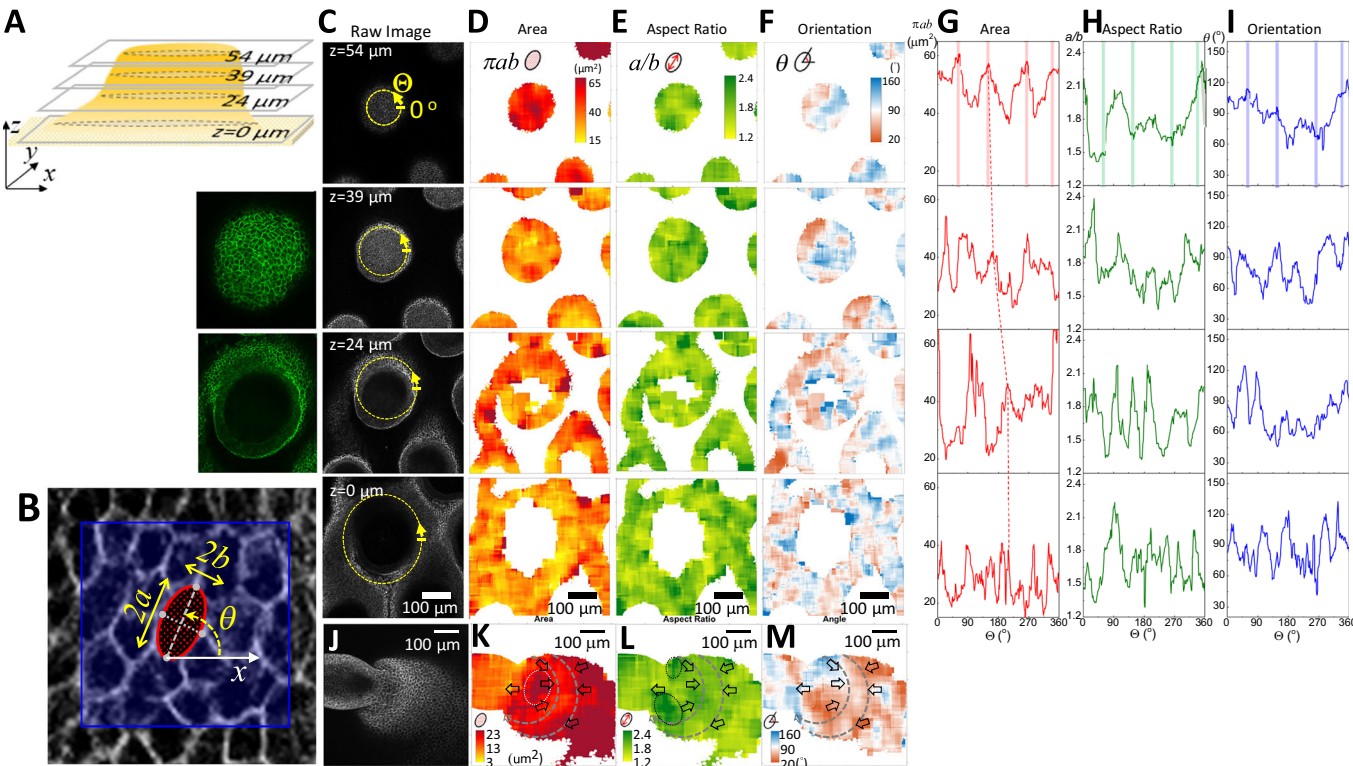

**Figure EV3. QMorF (Image-cased quantitative morphology field measurement).**

(A) Illustration of different section levels of an E9 feather bud that are used for analyses. (B) Representative quantifiable parameters of a cross-sectional cellular morphology in the QMorF analysis. $n = 3/3$, biological replicates. (C) Raw image of respective sections. The green lines point to the magnified original E-cadherin staining images of respective feather bud z-sections. $n = 3/3$, biological replicates. (D–F) Area, aspect ratio, and orientation of respective sections. $n = 3/3$, biological replicates. (G–I) Quantified area, aspect ratio, and orientation of cell shape over the azimuthal direction (Θ-axis) along the yellow dot circle in panel (C) of respective sections. Four peaks distributed over the azimuthal direction of the area plot in panel (G) at the bud tip (z = 54 μm) suggest a fourfold morphological symmetry of cells to accommodate the confined tip geometry. (J–M) Whole mount of E10 skin stained with antibodies to L-CAM (E-cadherin). Raw image used for quantification of area, aspect ratio, and orientation of invaginating E10 feather bud. $n = 3/3$, biological replicates. Please see the methods for QMorF analysis.

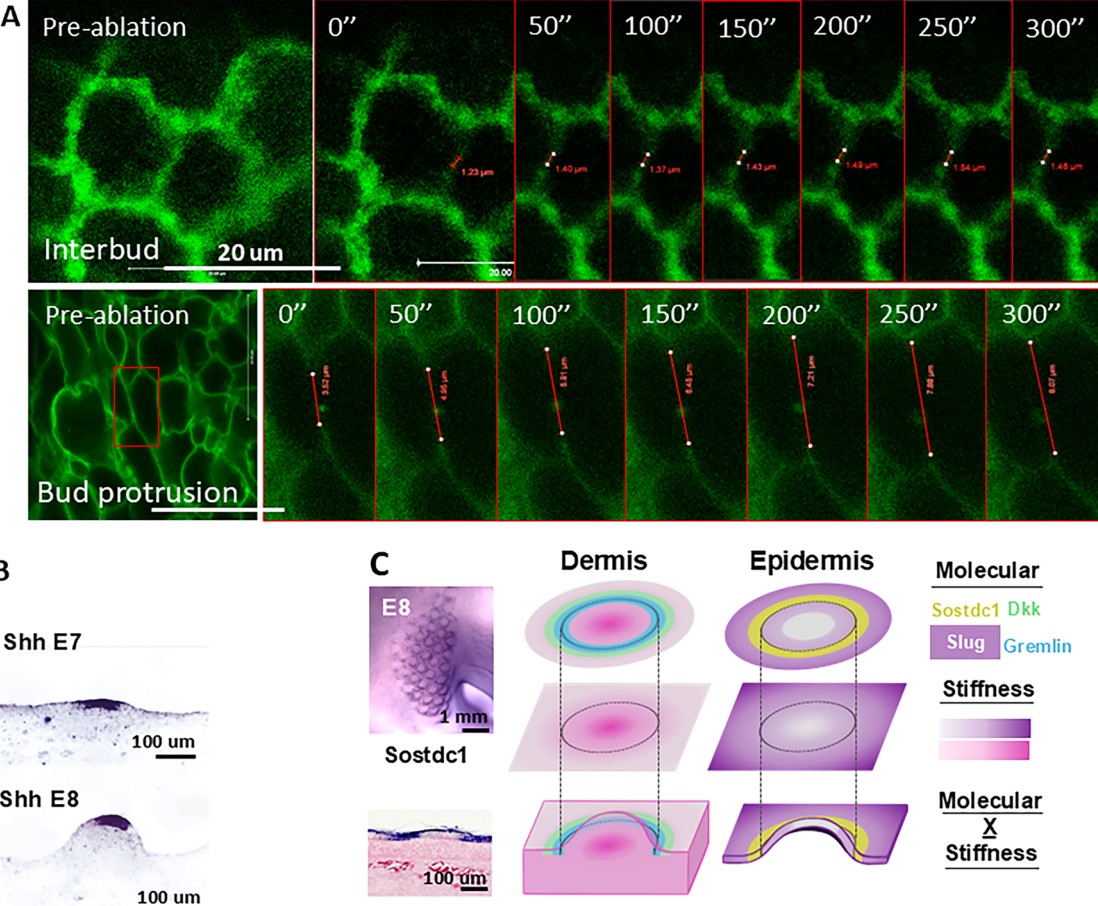

**Figure EV4. Cellular tension and molecular expression of protruding feather buds.**

(A) Laser ablation analyses showing ablated cell membrane retracted faster in epidermal cells of E8 protruding feather buds than in the interbud region. $n = 3/3$, biological replicates. (B) In situ hybridization of Shh of E7 and E8 chicken skin. $n = 3/3$, biological replicates. (C) Spatial expression pattern of Sostdc1, Slug, DKK1 and Gremlin in conjunction with spatial stiffness in the E9 dermis and epidermis. Notice these molecules demarcate bud boundaries.

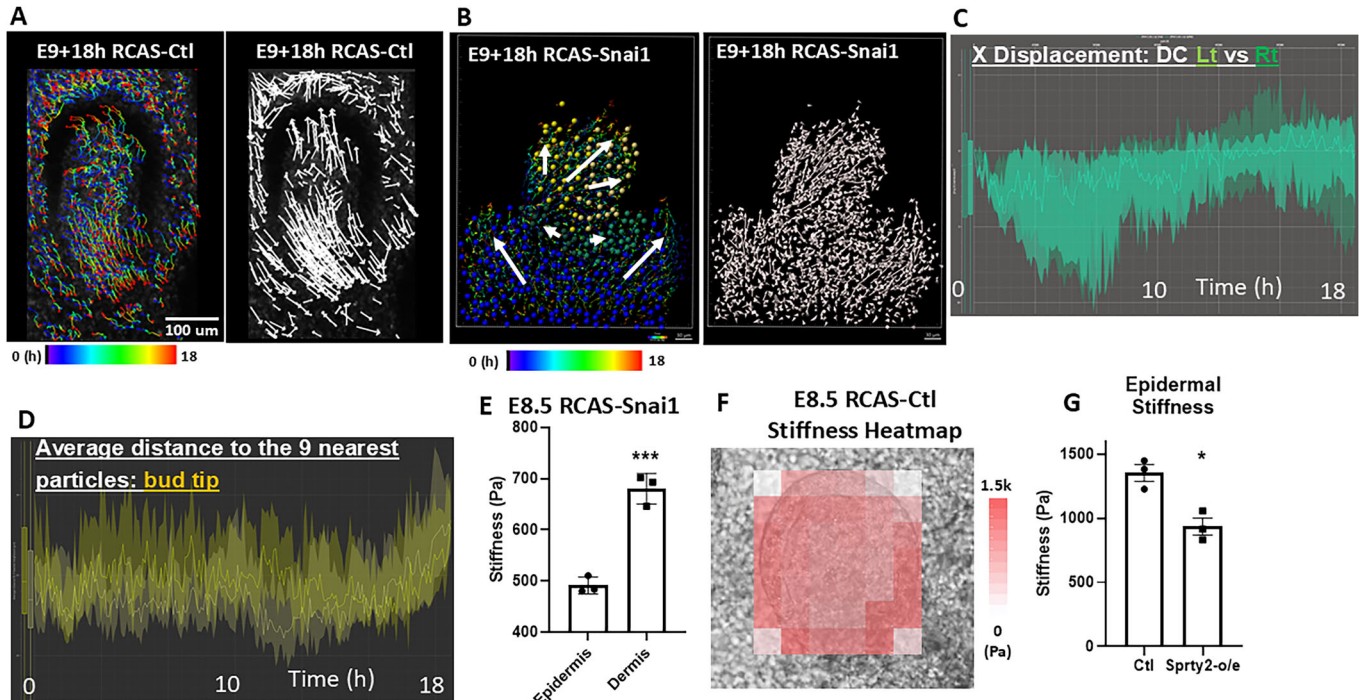

**Figure EV5. Dispersed cell migration pattern in the feather bud was demonstrated using Imaris cell track analysis of E9 + 18 h RCAS-Snai1 chicken skin.**

(A) Cell tracking of E9 + 18 h RCAS-Ctl. Left: Dermal cell tracked and color scale from 0 to 18 h. Right: arrowheads showing the initial to final displacement of dermal cells after 18 h of culturing. *N* = 3/3, biological replicates. (B) Cell tracking of E9 + 18 h RCAS-Snai1. Left: dermal cells of different regions labeled with specific colors. Yellow: feather bud dermal cells. Green: dermal condensate cells. Blue: non-feather bud dermal cells. Color scale: 0–18 h. Right: arrowheads showing the initial to final displacement of dermal cells after 18 h of culturing. *n* = 3/3, biological replicates. (C) E9 + 18 h RCAS-Snai1. Graph of X-axis (left-right) displacement of dermal condensate cells (DC) in left (Lt) and right (Rt) halves of DC over 18 h. *n* = 3/3, biological replicates. (D) E9 + 18 h RCAS-Snai1. The dispersed migrating pattern of feather bud dermal cells is demonstrated by the average distance to the 9 nearest particles in the feather bud dermal cells in the 15–18 h of cell tracking. Representative result. *n* = 3/3, biological replicates. (E) Stiffness comparison of E8.5 RCAS-Snai1 epidermis vs dermis. *T*-test, \*\*\**P* = 0.0007, *n* = 3/3, biological replicates. (F) Average stiffness heatmap of E8.5 RCAS-Ctl feather bud. *n* = 3/3, biological replicates. (G) Epidermal stiffness comparison of E8.5 RCAS-Ctl vs Sprty2 overexpression feather bud. *T*-test, \*P = 0.0109, *n* = 3/3, biological replicates.

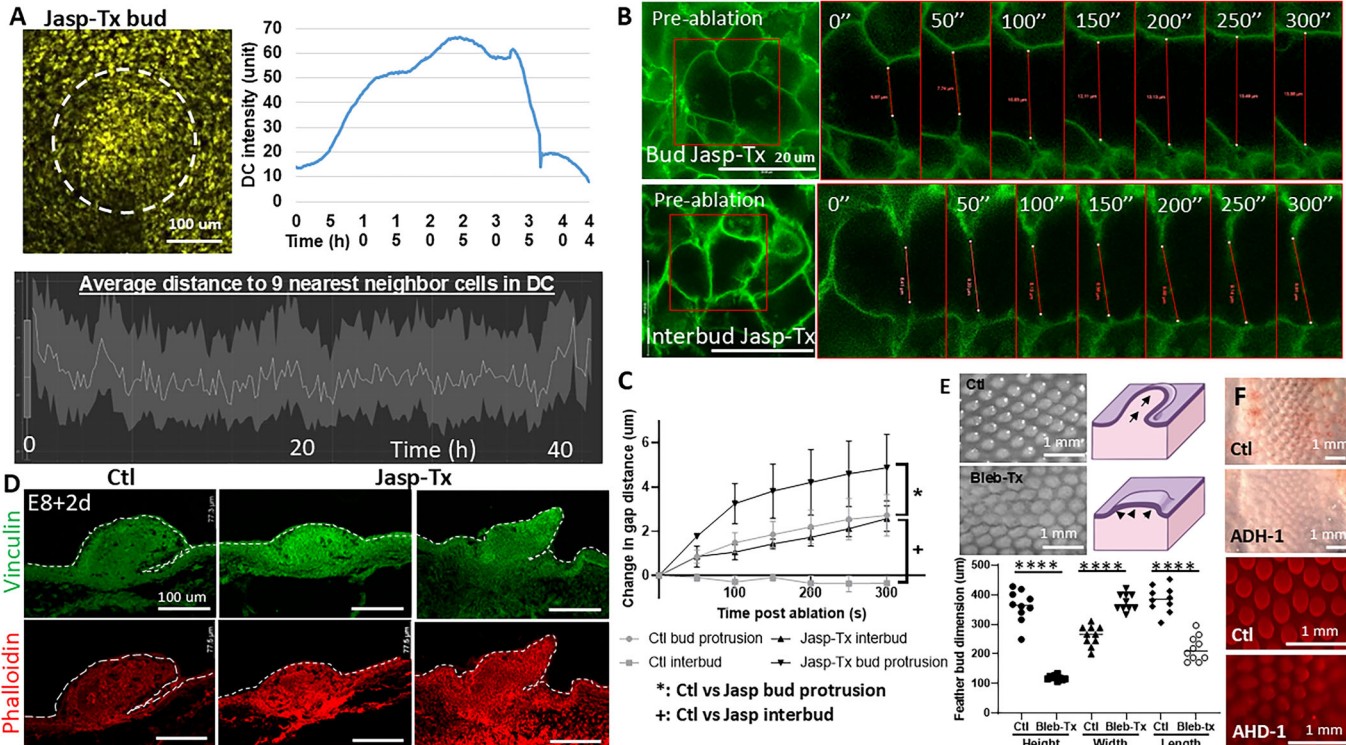

**Figure EV6. Perturbing the actomyosin system affects dermal condensation, cellular membrane tension dynamics, and normal feather bud growth.**

(A) Jasp-Tx E8 feather bud failed to progress after forming the initial dermal condensate (dotted line). Intensity of the DC and average distance to nine nearest neighbor cells in the DC both show an initial formation of DC and then disassemble after 30 h of Jasp-Tx. Representative image, $n = 3/3$, biological replicates. Scale bar: 100 um. (B) Laser ablation analyses showing Jasp-Tx increased epidermal cell membrane retraction rate in both bud and interbud region of E8 quail skin. Scale bar: 20 um. Representative results. $n = 3/3$, biological replicates. (C) Quantified results of the change in gap distance after laser ablation in various conditions. Two-way ANOVA. Ctl vs Jasp bud protrusion: F (1.177, 4.707) = 12.15, *$P = 0.0179$. Ctl vs Jasp interbud: F (2.021, 8.084) = 7.817, +$P = 0.0372$. $n = 3/3$, biological replicates. (D) Representative results of vinculin and phalloidin (F-actin) detection in E8 + 2 d control and Jasp-Tx quail feather buds. Vinculin is a focal adhesion protein, and phalloidin labels F-actin. Scale bar: 100 um. $n = 3/3$, biological replicates. (E) Blebbistatin treatment (Bleb-Tx) led to shortened and widened feather buds in E9 explants cultured for 48 h. Graph: quantified dot graph of feather bud dimension changes with and without Bleb-Tx. Scale bar: 1 mm. T-test, $P < 0.0001$ for all three comparisons. $n = 10/3$, biological replicates. (F) N-cadherin inhibitor ADH-1 treatment in E8 chicken explants for 2 days impeded normal feather bud development, leading to smaller feather buds. Scale bar: 1 mm. Representative result. $n = 3/3$, biological replicates.

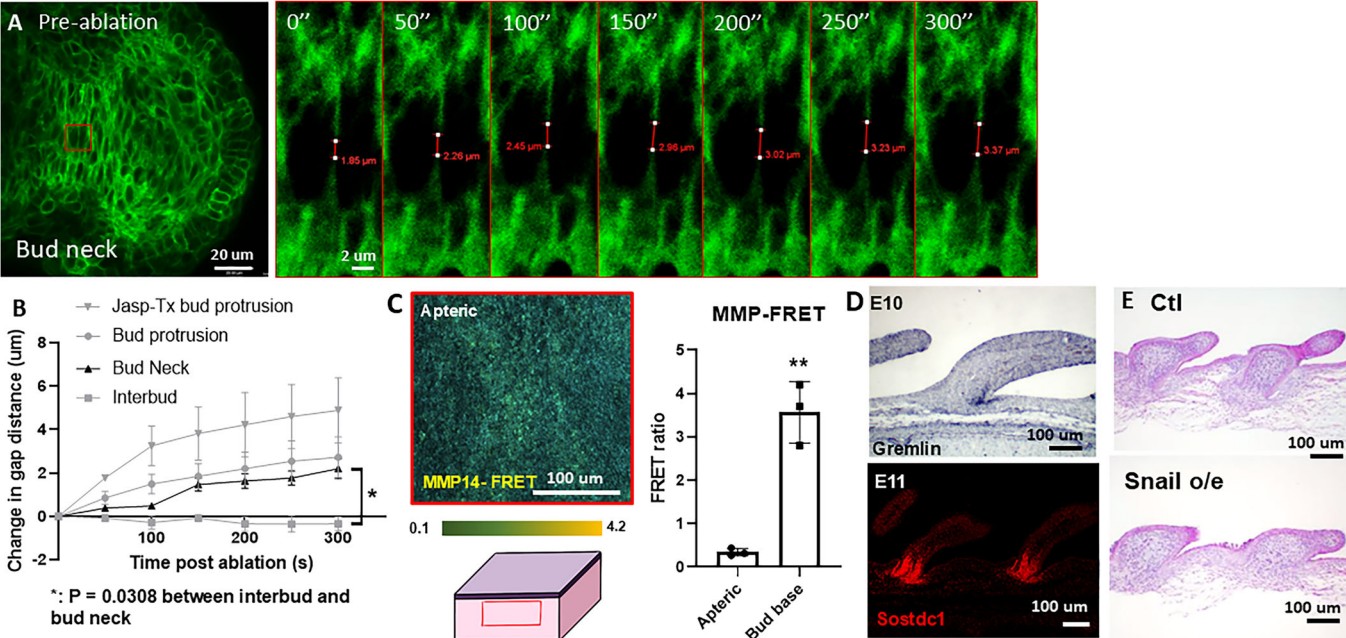

**Figure EV7. Molecular expression and cellular dynamics of invaginating feather follicle.**

(A) Laser ablation analyses showing epidermal cell membrane retraction in the bud neck region of an invaginating feather follicle of E8 quail skin. Scale bar: 20 um. $n = 3/3$, biological replicates. (B) Quantified results of the change in gap distance after laser ablation in the bud neck and various other conditions. Two-way ANOVA: Bud neck vs interbud, F $(1.422, 5.687) = 7.419$, *$P = 0.0308$. $n = 3/3$, biological replicates. (C) MMP14-FRET activity at the apteric region of E11 chicken skin, as illustrated. Graph: MMP14-FRET activity of apteric vs bud base region. Scale bar: 100 um. *t*-test, $P < 0.0015$. $n = 3/3$, biological replicates. (D) In situ hybridization of Gremlin in E10 feather bud and Sostdc1 in E11 chicken skin. Scale bar: 100 um. $n = 3/3$, biological replicates. (E) H&E photos of Ctl and Snail overexpression in the epidermis inhibited epidermal invagination in E7 reconstituted skin for 7 days. $n = 3/3$, biological replicates.

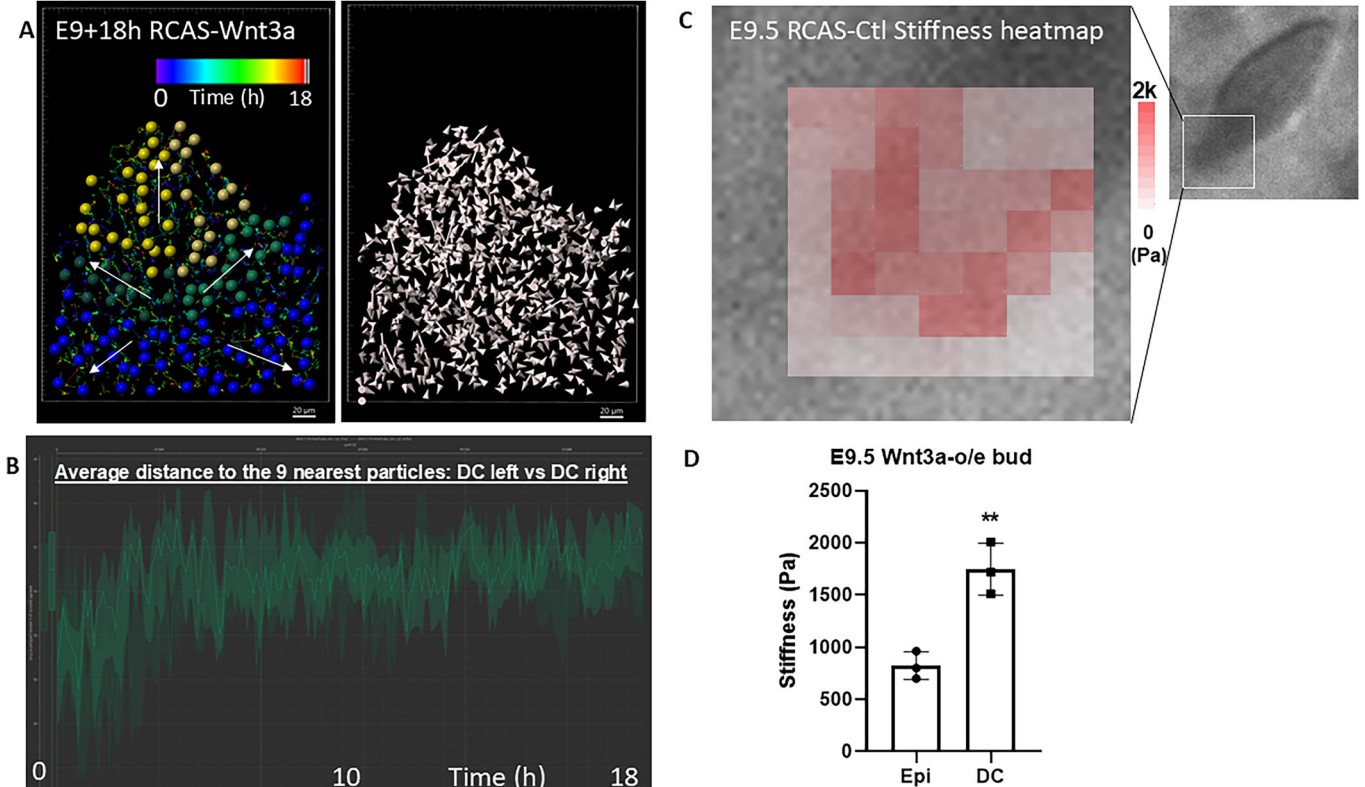

**Figure EV8.  RCAS-Wnt3a overexpression in the epidermis leads to dispersed dermal condensate cells.**

(A) Left: dermal cells of different regions labeled with specific colors. Yellow: feather bud dermal cells. Green: dermal condensate cells. Blue: non-feather bud dermal cells. Color scale: 0–18 h. Right: arrowheads showing the initial to final displacement of dermal cells after 18 h of culturing. $n = 3/3$, biological replicates. (B) Average distance to the nine nearest particles increased in both DC left and DC right after 3 h of culturing in RCAS-Wnt3a-overexpressing feather follicle. $n = 3/3$, biological replicates. (C) Top: average stiffness heatmap of E9.5 RCAS-Ctl bud neck and invagination area. $n = 3/3$, biological replicates. (D) Stiffness comparison of E9.5 Wnt3a-o/e bud epidermis vs DC. *T*-test, $P = 0.0048$. $n = 3/3$, biological replicates.

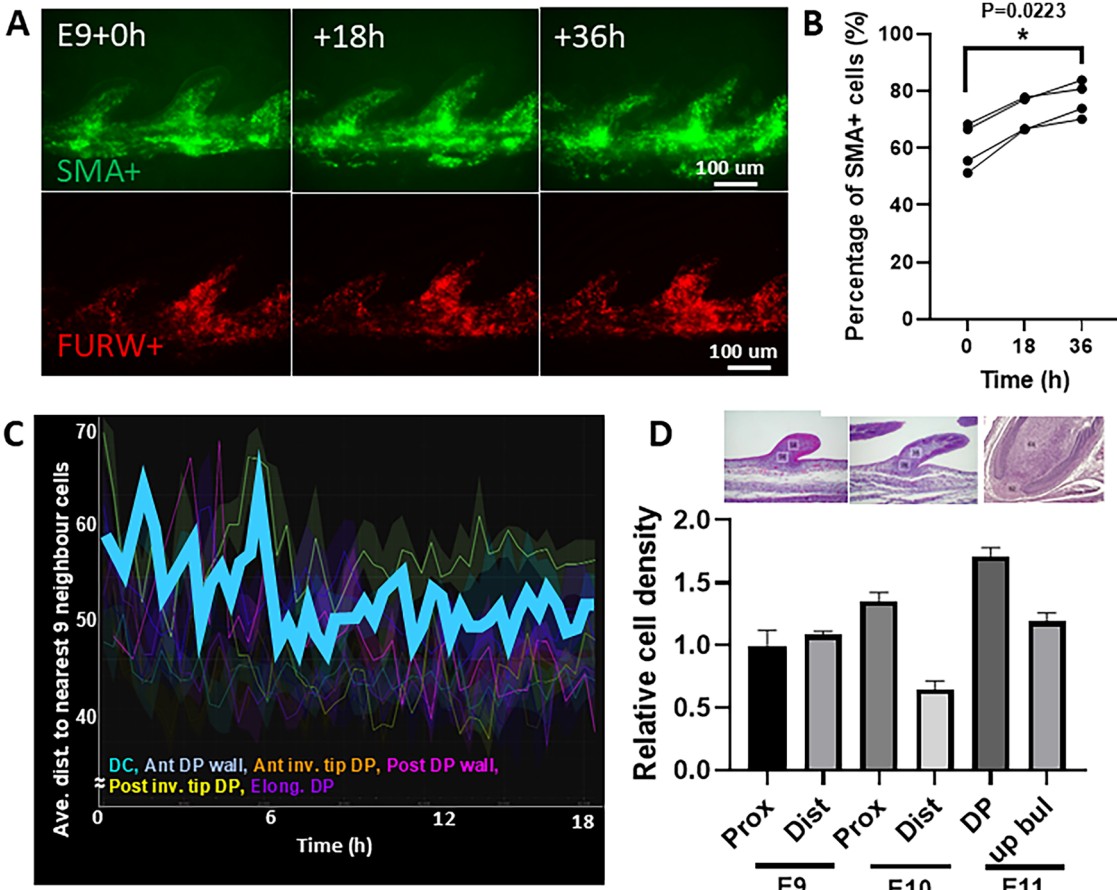

**Figure EV9. Molecular expression and cellular dynamics of dermal papillae formation.**

(A) The changes in SMA+ (green) and overall cell (red, FUKOW+) in growing E9 explant for 36 h. $n = 4/4$, biological replicates. (B) Graph showing quantified percentage of SMA+ over FUKOW+ cells in 4 E9 explant feather follicles. One-way ANOVA post hoc Tukey's. F = 5.974, 0 vs 36 h, $p = 0.0223$. $n = 4/4$, biological replicates. (C) Average distance to nearest neighbor cells in the invaginating E11 feather follicle. The bold cyan line indicates the median value of presumptive dermal papilla cells. Representative result. $n = 3/3$, biological replicates. (D) Relative cell density at proximal, distal, DP or upper bulge regions of the feather bud in E9, E10, and E11 feather bud. $n = 3/3$, biological replicates.

