## [Peer Review File · The EMBO Journal]

Novel tissue-mechanics-guided cellular flows drive the formation of feather follicles

Hans I-Chen Harn, Ting-Xin Jiang, Chih-Han Huang, Wen-Tau Juan, Tzu-Yu Liu, Tsao-Chi Chuang, Wan-Chi Liao, Peter Yingxiao Wang, Ji Li, Cornelis Jan Weijer, Ping Wu, Chin-Lin Guo, and Cheng-Ming Chuong

Corresponding author: Cheng-Ming Chuong (cmchuong@usc.edu)

Review Timeline:

Submission Date:	13th Dec 24
Editorial Decision:	28th Feb 25
Revision Received:	18th Sep 25
Editorial Decision:	19th Nov 25
Revision Received:	12th Jan 26
Editorial Decision:	5th Mar 26
Revision Received:	20th Mar 26
Accepted:	25th Mar 26

Editor: Ieva Gailite

Transaction Report:

Dear Ming,

Thank you for submitting your manuscript for consideration by the EMBO Journal. We have now received comments from a full set of reviewers, which are included below for your information.

Based on the overall interest expressed in the reports and your willingness to engage in a major revision as expressed in the preliminary revision plan provided during the pre-decision consultation, I would like to invite you to revise the manuscript as proposed in your point-by-point response. I should add that it is The EMBO Journal policy to allow only a single major round of revision and that it is therefore important to resolve the main concerns at this stage.

We generally allow three months as standard revision time, which I have now extended to four months based on the revision plan. Should you foresee a problem in meeting this deadline, please let me know in advance to discuss an extension. As a matter of policy, competing manuscripts published during this period will not negatively impact on our assessment of the conceptual advance presented by your study. However, please contact me as soon as possible upon publication of any related work to discuss the appropriate course of action.

When preparing your letter of response to the referees' comments, please bear in mind that this will form part of the Review Process File and will therefore be available online to the community. For more details on our Transparent Editorial Process, please visit our website: <https://www.embopress.org/page/journal/14602075/authorguide#transparentprocess>. Please also see the attached instructions for further guidelines on preparation of the revised manuscript.

Please feel free to contact me if you have any further questions regarding the revision. Thank you for the opportunity to consider your work for publication. I look forward to receiving your revised manuscript.

With best regards,

Ieva

Ieva Gailite, PhD
Senior Scientific Editor
The EMBO Journal
Meyershofstrasse 1
D-69117 Heidelberg
Tel: +4962218891309
i.gailite@embojournal.org

We realize that it is difficult to revise to a specific deadline. In the interest of protecting the conceptual advance provided by the work, we recommend a revision within 4 months (30th June 2025). Please discuss the revision progress ahead of this time with the editor if you require more time to complete the revisions.

Referee #1:

This manuscript addresses the question of how the feather follicle arises from a planar embryonic skin composed of epithelium and mesenchyme. It takes multiple distinct experimental approaches to investigate the role of cell signalling, cell movement and tissue mechanics to aim at an integrated understanding of tissue morphogenesis, from 2D to 3D. Addressing morphogenesis in this integrated way is a major challenge and the feather system a great model to do this. In my view the approach is novel, commendable and important in this breadth, but more is needed to tighten the connections between some elements, and the methods and experimental details need to be made clearer so that they can be assessed rigorously.

The numbers of experimental replicates and statistical significance of results are not clear for many experiments and the methods section is very short and incomplete.

It is unclear to me how "cell migration analysis" was done with Imaris. If individual cells were tracked, how many cells and how long the average tracks were, or if Imaris used a particle image velocimetry method to trace cell displacements is not specified here. The imaging itself should be explained more clearly, with references for the transgenic lines used and microscopy methods and post-processing explained more clearly (for example, if drift correction was applied).

It is not clear how the tissue was prepared for the AFM for sections, or whole mount, or how epidermis and dermis were separated. I can't find the statistical significance of reported differences in stiffness or what the error bars drawn represent (e.g. for SI Fig 2B), or what number of samples were assessed for Fig 5 D and J. These need to be explained.

The viral transductions should include some evidence of where the viral infection is, and the use of lentivirus (Figure 3A) explained in the methods and more clearly in the text. How did the lentivirus infect SMA positive cells, or did the lentivirus carry a SMA reporter, or were many cells infected and only the SMA positive ones analysed after staining?

These issues I am sure can be clarified by the authors.

Experimentally, the most interesting and important observation is the lack of feather follicle formation upon overexpression of Snail (or Wnt3a). This is the data in Figure 2J-K and the image is very convincing. This is the point where a clear understanding of what happens in this case would anchor the whole paper and argument, as the feather buds are present but the follicle is not, and the cause is clearly experimental expression of Snail. The text on p.8 states that "we overexpressed Snail and observed lowered follicle wall stiffness, widened buds with a broader base, and failed follicle invagination (Fig 2J)" but I can see only the image of the widened buds and failed invagination. The issue of whether the follicle wall is less stiff, as stated, is key as this would link mechanics to the upstream, causative expression of Snail, but I can not see the stiffness measurements for this presented in the paper. If these could be included and shown to be robustly significant, then this would strengthen the paper, together with a demonstration that the epidermis is expressing Snail (or at least that it is expressing RCAS proteins and thus infected).

For the Wnt3a expression experiment the text on p.8 states that Wnt3a overexpression causes "dispersed dermal cell flows (Fig 2K)" but it is not clear if these cell movements were actually observed directly or if Fig 2K is a summary of inferred cell movements that were not directly observed. This should be made clear.

The legend for the supplementary videos should be more informative and include the significance of all the coloured balls (is

each one a specific cell, tracked throughout?) and what the colours signify (e.g. the blue balls/cells in Video S3).

Referee #2:

General summary and opinion about the principal significance of the study

The authors determine cellular flows, tissue mechanics and morphogenetic signals to build a model for feather follicle formation. The model includes epidermal invagination around the dermal condensate and the formation of the dermal papilla. The study involves advanced methods and data analysis, which are all applied in a sound and appropriate way. The results and the model are interesting and important for developmental biology and morphogenesis. Even without much background knowledge of biomechanics, this manuscript is exciting. The illustrations are of high quality and helpful. A weakness of the paper is that it depends on many interpretations of data where a proof of the mechanism is desirable. The authors propose a model which is certainly a significant scientific advance, but the conclusions should be toned down to acknowledge that many assumptions were necessary for building the model.

Specific major concerns essential to be addressed to support the conclusions

Can you better explain interpretations such as "we interpret that Slug decreases epidermal stiffness while Sostdc1 increases epidermal stiffness, and DKK1 and Gremlin decrease dermal stiffness"?

The section "Scale to feather conversion: softening scale epidermis enables feather follicle formation" contains some claims that are not well supported. What is the mechanism by which Spry2 softens the epidermis?

The molecular and cellular basis for "stiffness" is unclear. Please either explain it better or state that it is not fully understood. For example, it is not conclusive to state "the stiffness of epidermal cells primarily increases due to elastic cell stretching within the epidermal layers" (no reference). "Snail is known for downregulating epidermal cell junctions (Cano, Perez-Moreno et al. 2000); hence, it is reasonable to observe the overall reduction in epidermal stiffness when it is expressed" - sounds rather speculative.

Minor concerns that should be addressed

The Methods section requires more details to make the study reproducible. Under "Atomic force microscopy", more information about the microscope should be provided.

H2B-mOrange and mCherry-H2B labeling should be explained when they are first mentioned in the text.

The word "evolution" in the title is questionable.

The use of the English language needs minor improvements (present tense versus past tense).

Referee #3:

The manuscript by Harn, Chuong and colleagues aims to understand the chemo-mechanical regulation of cell behaviors that leads to morphogenesis of feathers on the avian skin. The study combines gene misexpression with live imaging, mechanical testing, and mathematical modeling. While the study aims to address important conceptual questions in a system well-suited for this purpose, there are several fundamental issues that need to be addressed, which would involve reworking the manuscript and its content significantly. The most pressing issues are detailed below. Generally, the manuscript may benefit from focusing on a particular phase of the feather bud process and digging in deeper, rather than the current format which feels dense and difficult to follow, but also superficial in many regards.

- Data are presented without indicating the number of replicates.
- No statistical tests are performed on quantitative data.
- The Methods are not described in sufficient detail to assess the quality of the work.
- There are no scale bars on any images to indicate spatial scale.
- Throughout the manuscript, correlation is conflated with causation. In fact, there are too many examples of this to list out. Some examples include page 5 ("vertical movement of DC cells...is guided, at least partially, by ...softened epidermis"), page 6 ("we interpret that Slug decreases epidermal stiffness while Sostdc1 increases epidermal stiffness, and DKK1 and Gremlin decrease dermal stiffness (Fig 1I, SI Fig 4B)."), page 7 ("We observed that as the feather bud continues to elongate, it stretches the

epidermal cells"), page 8("Together these results suggest that follicle formation is achieved via stretching of the bud/ interbud boundary - indicated by a stiff epidermal wall, YAP expression and a stretched cell morphology - that triggers the initial epithelial invagination."), and several more examples.

-Along the lines of correlation vs. causation, morphological data is frequently over/misinterpreted as mechanical data. For example larger cell areas on the bud are described as indicating stretching of these cells by outgrowth of dermis. But stretching is a physical process and cannot be determined from images alone, as expansion of cells pushing the neighboring tissue would look the same as stretching of the cells by the neighboring tissue. Force measurements (laser ablation, microdissection, or other methods) are needed to make any statement on forces and their effects.

-The distinction between true results and their interpretation is overly blurred, with schematic figures appearing in place of data (Fig. 1C, L for example). Many statements in the Results should be moved to the Discussion, as they are not a direct description of results, but instead an extrapolation of those results, which are often times overreaching (frequently through equating correlation to causation). A partial list of examples include page 6 ("By overlaying these concentric zone maps and their known properties, we interpret that Slug decreases epidermal stiffness while Sostdc1 increases epidermal stiffness, and DKK1 and Gremlin decrease dermal stiffness (Fig 1I, SI Fig 4B). This interpretation is based on previous findings that Dkk1 and Gremlin weaken dermal condensation (Chen et al., 2015), leading to decreased dermal stiffness. Together, after the bud protrusion induced-stretching, one anticipates a feedforward connection between these molecular signaling events and the physical morphogenetic processes. These responses prelude the mechanical environments for epidermal invaginations that will take place in the next stage at E9."), page 6 ("These results suggest that epidermal stiffness helps confine the bud shape, and the softened epidermis may allow the dermal cells to migrate laterally, leading to the expansion of bud width (Fig 1L)."), page 7 ("From a mechanical perspective, it is also likely that migrating cells with a certain degree of stiffness tend to invade softer regions rather than stiffer ones."), page 7 ("The moderately stiff DC (0.72 {plus minus} 0.07 kPa) could also serve as the physical barrier that divides distal bud dermal cells (0.32 {plus minus} 0.03 kPa, to become pulp) from the basal dermal cells..."), page 8 ("These results could imply that, initiated by feather bud elongation induced-stretching, YAP, Wnt, MMP, and others work in concert to soften the dermis around the DC and facilitate epidermal tongue invaginations (Fig 2D, I)."), and many more.

-Many data need to be quantified, as representative images - particularly without indicating replicates - are not compelling on their own.

-Time lapse data are presented as a single representative arrow plot with sparsely placed arrowheads, yet these data are overinterpreted in the text, and many claims need to be tested by directly quantifying velocity fields. For, example, spatial variation in velocity cannot be readily interpreted from the data, and strain rate fields should instead be quantified. Spreading behaviors are described, but should be directly quantified as divergence of the velocity field. Disorder is described but should be quantified using one of many measures of coherence.

-Measurements of stiffness by AFM do not appear to take into account geometric effects. Indentation testing of the same material in a flat vs curved geometry would produce differing effects, with apparent stiffness higher when indenting against the radius of curvature. However, this geometric effect is not considered, and results may therefore be incorrect.

-Many aspects of the mechanics are not well described or misleading. For example, phase transitions are conflated with viscoelasticity (page 3); changes in speed or direction do not imply more fluid like behavior, as a solid can accelerate, decelerate and change directions (page 4).

-Stiffness changes are achieved by misexpression of genes such as Spry2 and Snail, which will have many effects on cell behavior that extend beyond changes in tissue stiffness. Controls are lacking, however, to isolate effects as a result of stiffness changes as opposed to other cell behaviors.

-SI Fig 2D is missing.

-Unclear whether RCAS infections are restricted to epidermal vs dermal compartments, or spread throughout both. If the latter, this would need to be addressed, and interpretation of results adjusted accordingly.

-Many claims in the manuscript seem unsupported by data. For example, the claim that mechanical factors are claimed to have a role in dermal cell fate specification (page 9) or the statement "When putative DPs reach a compaction threshold, they secrete more Tgfb β that trigger the bending of the epithelial tongue toward DP, In so doing, other molecules are induced in different topological positions in this newly set up proximal follicle, positioning themselves for future cyclic renewal"

-SI Fig. 2A is referred to as SI Fig. 2B erroneously in the text (page 4)

Response letter to reviewers

We sincerely thank all three reviewers for their constructive comments. The comments are insightful and help us improve the manuscript. Here we have a general response followed by a point-by-point reply to the reviewers' comments.

A key question raised by the reviewers is about causal-relationship amongst the candidate molecule, cellular behaviour and tissue mechanics. Our major thesis centers around the cross talks between biomechanical signals and biochemical signals, and how these interactions lead to changes in biophysical properties, collective cell behavior and phenotypes during follicle formation.

In this paper and revision, we have developed the following methods to monitor mechanical changes during the formation of feather follicle:

- AFM (tissue rigidity)
- QMorF (force estimation based on cellular morphology)
- Cellular flow dynamics (4D movie)
- Laser ablation (cell junction retraction)

For feather follicle structural and molecular changes, we have done the following:

- Analyze the process of follicle architecture building in every topological transformation step (protruding, invagination, compaction, etc.) we can measure
- Study the expression of morphogens (Wnt3a, TGF beta, Sprouty2, etc.), force related molecules (Snai1, YAP, Vinculin, F-actin etc.) and cell-cell/cell-matrix interaction molecules (MMP, etc.).

We characterized the above aspects in detail during normal developmental processes, which enabled us to develop the three-phase model for follicle formation. For molecular perturbation, we selected multiple molecules—including Sprouty, Snai1, Wnt3a, TGF- β inhibitor, and actin polymerization stabilizers—based on their specific expression patterns during the three phases of follicle formation. Observing how different perturbations affected follicle structures provided novel insights into the progression of morphogenetic

processes. However, we acknowledge that in our initial submission, comprehensive biophysical analyses were not conducted for every molecular perturbation, leading to the reviewers' observation that some links to biophysical changes were correlative.

To address this concern in the revised manuscript, we focused on multiple representative molecules—Snai1, Wnt3a, Sprouty2, and YAP—by performing functional perturbations via RCAS transfection, which produced the phenotypes reported in the manuscript. We then applied our established methods to assess mechanical changes, including AFM measurements and cellular flow dynamics. Additionally, we introduced a laser ablation assay to evaluate cell membrane retraction speed, providing an indirect measure of cellular forces despite limitations in laser tissue penetration. We also used Jasplakinolide to stabilize actin filaments, demonstrating that morphogenesis requires dynamic mechanical contrasts in cellular forces, and that these mechanical dynamics are not merely by-products of morphogenesis.

For molecules not experimentally tested, we explicitly state their roles as correlative and discuss their potential functions in the Discussion section.

In summary, the revised manuscript thoroughly addresses the reviewers' comments through the addition of new results and substantial revisions. These efforts provide a comprehensive response and establish a new conceptual framework for understanding the topological transformation from planar skin to three-dimensional follicles, as well as the essential biophysical–biochemical crosstalk driving this morphological transition. Beyond advancing our understanding of skin and follicle biology, these findings reveal general principles of tissue topological remodeling, with implications for organogenesis, regeneration, and bioengineering.

Please see below for our point-by-point responses to each reviewer comment, where our responses are in green.

Point to point response to reviewer

Referee #1 (Report for Author)

This manuscript addresses the question of how the feather follicle arises from a planar embryonic skin composed of epithelium and mesenchyme. It takes multiple distinct experimental approaches to investigate the role of cell signalling, cell movement and tissue mechanics to aim at an integrated understanding of tissue morphogenesis, from 2D to 3D. Addressing morphogenesis in this integrated way is a major challenge and the feather system a great model to do this. In my view the approach is novel, commendable and important in this breadth, but more is needed to tighten the connections between some elements, and the methods and experimental details need to be made clearer so that they can be assessed rigorously.

The numbers of experimental replicates and statistical significance of results are not clear for many experiments and the methods section is very short and incomplete.

RE: We appreciate the comments. We now provide experimental replicates, statistical analyses, and significantly expand the description of methods. Please refer to the revised methods section.

It is unclear to me how "cell migration analysis" was done with Imaris. If individual cells were tracked, how many cells and how long the average tracks were, or if Imaris used a particle image velocimetry method to trace cell displacements is not specified here. The imaging itself should be explained more clearly, with references for the transgenic lines used and microscopy methods and post-processing explained more clearly (for example, if drift correction was applied).

RE: We now provide the details for each analyzed video: number of cells analyzed, average track over time, image velocimetry, cell displacement, in addition to detail explanation of post-processing methodology, drift correction, etc. An excerpt of the video analysis methodology is shown below:

“Imaris (Version 9.2. Oxford Instruments) was used for cell migration analysis. Drift correction was applied using the "Correct Drift" function in Imaris, based on the reference time point and structural alignment across time-lapse data. Background subtraction was performed using a rolling ball algorithm (radius = 10-12 μm) to enhance signal-to-noise ratio.

Approximately 20,000 to 25,000 cells were tracked and analysed per sample in this study. Tracked particles were detected using the Spot function in Imaris based on the nuclear RCAS-mOrange signal, and all analyses are based on particle trajectories. We extracted 3D displacement vectors and positions (X, Y, Z) for each tracked particle (e.g., cell nucleus or centroid) over time, from which the magnitude and direction of displacement could be computed.

Cells in different regions of the feather bud were manually selected and grouped by different colours according to their respective positions. From there, the migration behaviour of each group (e.g., dermal cells, invaginating epidermal cells) could be analysed independently.”

Please refer to Methods for the full description.

It is not clear how the tissue was prepared for the AFM for sections, or whole mount, or how epidermis and dermis were separated. I can't find the statistical significance of reported differences in stiffness or what the error bars drawn represent (e.g. for SI Fig 2B), or what number of samples were assessed for Fig 5 D and J. These need to be explained.

RE: Thank you for the opportunity to clarify these issues. We now provide detail methods on sample preparation, wholemount, AFM and its measurements, including statistical analysis, error bar representation, number of samples involved, etc. in METHODS – Sample preparation for AFM measurement. Please see below for the excerpt:

“Direct measurement (unprocessed sample). The skin explant was harvested and cultured on the culture-insert mesh and incubated with medium at normal culture condition for at least 2h until stably attached. Just prior to measurement, the mesh was cut out using a surgical blade and placed on a glass coverslip designated for JPK temperature control system.

Separating dermis/epidermis. The skin explant was harvested and placed in calcium-magnesium-free solution (2x CMF) solution with 0.25% EDTA for 15 to 25 min on ice, in which the epidermis and dermis could be separated mechanically under the dissecting microscope. The separated tissues were laid flat on a glass coverslip and loaded onto the AFM.

Longitudinal section. The skin explant was sliced longitudinally using a surgical blade. The sliced tissue was laid on its side onto the culture-insert mesh, and phase-contrast view of the stereomicroscope was used to visualize the epidermal-dermal interface. The mesh is then cut-out and transferred to a glass coverslip and loaded onto the AFM, where the cantilever can contact the cross-section of the feather bud structures.”

The viral transductions should include some evidence of where the viral infection is, and the use of lentivirus (Figure 3A) explained in the methods and more clearly in the text. How did the lentivirus infect SMA positive cells, or did the lentivirus carry a SMA reporter, or were many cells infected and only the SMA positive ones analysed after staining?

RE: Thank you for the opportunity to clarify these issues. The lentivirus carrying a SMA promoter was used to label cells expressing SMA, and the injection also included an ubiquitous backbone carrying an RFP tag at 1:1 ratio. Most of the cells would carry the RFP signal while only the SMA+ cells would show green. No staining was needed afterwards. We have included the details and the references in this revised manuscript, Methods – Viral production and transfection. Below is the excerpt:

“Lentivirus was made in reference to Harn et al. (Harn et al., 2021). Plasmid for the reporter smooth muscle actin was purchased from Addgene (acta2_GFP_pA_pDestTolpA2, (catalog # 78684). FURW (flap-Ub promoter-RFP-WRE) plasmids (catalog #14883) were made by replacing the EGFP vector with that of RFP. To co-transfect the embryo, the 2 viruses were mixed at 1:1 ratio and injected into the chicken embryo somite at E3.”

These issues I am sure can be clarified by the authors.

Experimentally, the most interesting and important observation is the lack of feather follicle formation upon overexpression of Snail (or Wnt3a). This is the data in Figure 2J-K and the image is very convincing. This is the point where a clear understanding of what happens in this case would anchor the whole paper and argument, as the feather buds are present but the follicle is not, and the cause is clearly experimental expression of Snail. The text on p.8 states that "we overexpressed Snail and observed lowered follicle wall stiffness, widened buds with a broader base, and failed follicle invagination (Fig 2J)" but I can see only the image of the widened buds and failed invagination. The issue of whether the follicle wall is less stiff, as stated, is key as this would link mechanics to the upstream, causative expression of Snail, but I can not see the stiffness measurements for this presented in the paper.

If these could be included and shown to be robustly significant, then this would strengthen the paper, together with a demonstration that the epidermis is expressing Snail (or at least that it is expressing RCAS proteins and thus infected).

RE: Thanks for your suggestion. To characterize the biophysical changes of these perturbed phenotypes, we now use AFM to measure the mechanical changes in the RCAS-Snai1 and RCAS-Wnt3a overexpressing feather buds in protrusion and invagination stages, respectively. The results show that both Snai1 and Wnt3a both significantly reduced the stiffness of epidermis and a more “dispersed” pattern of dermal cells. The results of RCAS-Snai1 is shown in Fig 1J-L and SI Fig 5.

For the Wnt3a expression experiment the text on p.8 states that Wnt3a overexpression causes "dispersed dermal cell flows (Fig 2K)" but it is not clear if these cell movements were actually observed directly or if Fig 2K is a summary of inferred cell movements that were not directly observed. This should be made clear.

RE: Thank you for the opportunity to clarify. We labeled and tracked cell migration patterns in RCAS-Wnt3a overexpressing E9 feather buds. Our analyses show that dermal cells adopt a dispersed pattern, as evidenced by their movement and the increased average distance from neighbouring cells.

The legend for the supplementary videos should be more informative and include the significance of all the coloured balls (is each one a specific cell, tracked throughout?) and what the colours signify (e.g. the blue balls/cells in Video S3).

RE: We now provide information on how cells are labeled and tracked in Methods. We also add how the coloured balls are labelled and tracked for all SI Videos legends. Please see below for the excerpt:

“Supplementary Video 1. Feather bud protrusion. Cell tracking video of E7 chicken explant culture for 18h reveals cell migration pattern changed from horizontal to vertical during bud protrusion. Each colour ball represents 1 cell particle tracked using Imaris. The cell tracks (dragon tail) represent the migration pattern of the cell particle throughout the video. The colour gradient of the tracks indicates the timing of the tracks.”

Supplementary video 2. Feather follicle invagination. Cell tracking video of E10+24h explant showing different cellular flows. Each colour ball represents 1 cell particle tracked using Imaris. The colour gradient of the tracks (dragon tail) is limited to the last 3h.

Supplementary video 3. Dermal papillae formation. Cell tracking video of E17+24h flight feather to reveal differential cellular flows: The particles are post-edited labelled in different colours according to their position and cell types. (green) DP forming cells migrate downwards. (red) Epidermal tongue cells migrate downwards while distal feather epidermal cells migrate upwards. (yellow) Feather elongating dermal cells migrate distally. Only the last 3h of the track (dragon tail) were shown.

Supplementary video 4. Scutate scale formation. Cell tracking video of E10+24h quail scale showing limited dermal cell migration during scale formation. Epidermal cells are labelled blue while dermal cells are purple. The entire duration of the cell tracks were shown in colour gradient.

Supplementary video 5. Scale to feather conversion. Cell tracking video of a newly induced feather bud on top of the scutate scale. The tracked particle cells are labelled by different colours according to their cell type and location. Purple: scale epidermal cells. Yellow: converted-feather epidermal cells. Blue: scale dermal cells. Black: converted-feather dermal cells. Total duration: 16h, only the last 3h of the track (dragon tail) were shown.”

Referee #2 (Report for Author)

General summary and opinion about the principal significance of the study

The authors determine cellular flows, tissue mechanics and morphogenetic signals to build a model for feather follicle formation. The model includes epidermal invagination around the dermal condensate and the formation of the dermal papilla. The study involves advanced methods and data analysis, which are all applied in a sound and appropriate way. The results and the model are interesting and important for developmental biology and morphogenesis. Even without much background knowledge of biomechanics, this manuscript is exciting. The illustrations are of high quality and helpful. A weakness of the paper is that it depends on many interpretations of data where a proof of the mechanism is desirable. The authors propose a model which is certainly a significant scientific advance, but the conclusions should be toned down to acknowledge that many assumptions were necessary for building the model.

Specific major concerns essential to be addressed to support the conclusions

Can you better explain interpretations such as "we interpret that Slug decreases epidermal stiffness while Sostdc1 increases epidermal stiffness, and DKK1 and Gremlin decrease dermal stiffness"?

RE: We measured changes in tissue stiffness and cell flow behaviours in Snai1-overexpressing feather buds. The results demonstrate epidermal softening accompanied by dispersed migratory patterns of dermal cells. These results are included in Fig 1J-L, SI Fig 5.

Our interpretation regarding the regulation of tissue stiffness by Sostdc, Dkk1, and Gremlin were based on observed morphological changes. Given the time constraints, it was not feasible to conduct detailed functional experiments for all these molecules. Therefore, we have moved most of these findings to the Supplementary Figures and removed speculative statements from the main text.

The section "Scale to feather conversion: softening scale epidermis enables feather follicle formation" contains some claims that are not well supported. What is the mechanism by which Spry2 softens the epidermis?

RE: Literature on *Drosophila* trachea formation has shown that spry2 genes encode proteins that act primarily as negative regulators of receptor tyrosine kinase (RTK) signaling

pathways, such as those involving Fibroblast Growth Factors (FGFs) or Epidermal Growth Factor (EGF). By inhibiting Ras-MAPK pathway, proper branching pattern forms. In feather morphogenesis, earlier we have shown that Spry over-expression leads to increase of barb ridge formation (Yue Z.... Chuong CM. *Deve Biol.* 2012. <https://doi.org/10.1016/j.ydbio.2012.09.004>). These works focus more on phenotypic changes and the explanations were based on biochemical signaling (e.g., MAPK).

However, the effect of Sprouty2 overexpression on cellular and tissue biophysical properties have not been studied directly. Our data now provide supporting evidence. In addition to biophysical measurements (AFM and cell tracking), we examined downstream events that can influence tissue rigidity, including actin cytoskeleton organization and focal adhesion assembly. Our IHC analyses show that focal adhesion markers (FAK and vinculin) and cytoskeleton assembly (revealed by Phalloidin) were downregulated during Sprouty2-induced scale-to-feather conversion, further supporting the notion that Sprouty2 can modulate tissue mechanics in this process. These results are now included in new Fig 5L.

The molecular and cellular basis for "stiffness" is unclear. Please either explain it better or state that it is not fully understood. For example, it is not conclusive to state "the stiffness of epidermal cells primarily increases due to elastic cell stretching within the epidermal layers" (no reference). "Snail is known for downregulating epidermal cell junctions (Cano, Perez-Moreno et al. 2000); hence, it is reasonable to observe the overall reduction in epidermal stiffness when it is expressed" - sounds rather speculative.

RE: We have revised these statements and complemented them with direct mechanical measurements to minimize speculation. Specifically, we performed laser ablation assay on cell junctions of protruding bud, interbud and invaginating bud neck (Fig 1F, 2H). We also performed AFM stiffness measurement on the molecularly perturbed feather buds (Fig 1L, 1N, 1O). These experimental findings provide better evidence to support our hypothesis that these chemo-mechanical coupling molecules play a role in modulating the stiffness of the tissues.

Minor concerns that should be addressed

The Methods section requires more details to make the study reproducible. Under "Atomic force microscopy", more information about the microscope should be provided.

RE: We now provide detailed description of "Atomic force microscopy, data analysis and stiffness heatmap" and "Sample preparation for AFM measurement" in Methods section. Please also see below for the excerpt:

"Atomic force microscopy, data analysis and stiffness heatmap. *The AFM (NanoWizard 4a/CellHesion, JPK, Berlin, Germany) combined with a stereomicroscope was set up in contact mode indentation to measure tissues in PBS. The stereomicroscope is aligned with cantilever and sample so the location of indentation can be precisely observed. SAA-SPH-5UM AFM cantilevers (Bruker, MA, USA) of cylindrical tip with 5 μm end radius and spring constant between 0.1-0.2 N/m. The spring constants of all cantilevers were calibrated via thermal noise method with correction factor in liquid (Butt & Jaschke, 1995; Hutter & Bechhoefer, 1993) prior to each measurement. A force series identified a maximum indentation force of 5 nN to show the most consistent results on test samples. A constant rate of 1 $\mu\text{m}/\text{s}$ was used for the entire approach and retract sequence. Force-distance curves were collected and post-processed using the JPK package software (Data Processing, 6.3.11). The force curves were analyzed using the Hertz model with a spherical indentation (Lin et al, 2009; Tripathy & Berger, 2009).*

Data analysis and heatmap generation were done in reference to Harn et al (Harn et al., 2021). At least 5 indentation points were taken for each region of interest, and at least 3 biological replicates were measured for each condition. The data from the same biological locations (e.g. dermal condensate, epidermal invagination site, epidermal protrusion) were averaged and used to plot the eventual stiffness heatmap.

The interpolation of tissue stiffness was performed by using 3-D meshgrid function of MATLAB (R2015b, Natick, MA, USA). After obtaining a Young's modulus (z) at a specific spatial location (x, y) in the wound, a 3-dimensional matrix was defined. When the positions

and stiffness of all the measured spots were identified, we could interpolate the stiffness of the positions in between to average the stiffness of the nearest parameters using 3-D meshgrid function, by defining (x, y) as meshgrid and (z) as griddata. In the end, the heatmap was generated by defining the representative color of stiffness.

Sample preparation for AFM measurement.

Direct measurement (unprocessed sample). The skin explant was harvested and cultured on the culture-insert mesh and incubated with medium at normal culture condition for at least 2h until stably attached. Just prior to measurement, the mesh was cut out using a surgical blade and placed on a glass coverslip designated for JPK temperature control system.”

Please see full description in the new Method section.

H2B-mOrange and mCherry-H2B labeling should be explained when they are first mentioned in the text.

RE: Thank you for pointing out this. We now provide explanations in the main text and methods. Below is the excerpt from the main text and methods:

“To characterize cell flow dynamics of the developing skin, we recorded 4D (x,y,z,t) confocal videos using RCAS-H2B-mOrange labelled chicken skin explants. RCAS-H2B-mOrange virus were injected into the somite of the chicken embryo and amniote sac to transfect and label the nucleus of dermal and epidermal cells, respectively at E3.”

“RCAS injection is performed on E3 embryo to transfect the epidermis, dermis or both, the virus is injected into the amniote sac, dermis or both.

RCAS-H2B-mOrange is a gift from the Bronner lab (Tang et al, 2019).”

The word "evolution" in the title is questionable.

We have removed evolution and replace it with “formation” from the title. The revised title now reads: “Novel Tissue Mechanics-Guided Cellular Flows Enable the Formation of Feather Follicles”.

The use of the English language needs minor improvements (present tense versus past tense).

The manuscript has been thoroughly reviewed for language and clarify. Thank you.

Referee #3 (Report for Author)

The manuscript by Harn, Chuong and colleagues aims to understand the chemo-mechanical regulation of cell behaviors that leads to morphogenesis of feathers on the avian skin. The study combines gene misexpression with live imaging, mechanical testing, and mathematical modeling. While the study aims to address important conceptual questions in a system well-suited for this purpose, there are several fundamental issues that need to be addressed, which would involve reworking the manuscript and its content significantly. The most pressing issues are detailed below. Generally, the manuscript may benefit from focusing on a particular phase of the feather bud process and digging in deeper, rather than the current format which feels dense and difficult to follow, but also superficial in many regards.

RE: Thank you for raising this important point. Our intention is to present the making of the feather follicle as a continuous process, with three highlighted stages that connect sequentially until a stable follicle configuration is established. We have invested substantial effort to strengthen the evidence in each of these stages, and we believe the revised manuscript now presents a more coherent and complete story of the topological transformation from a 2D epithelial plane to a 3D follicle architecture. In addition, we have done our best to perform key experiments to illustrate principles involved in different stages of feather bud development.

-Data are presented without indicating the number of replicates.

- No statistical tests are performed on quantitative data.
- The Methods are not described in sufficient detail to assess the quality of the work.
- There are no scale bars on any images to indicate spatial scale.

RE: Thank you for your comments about these neglected points. We have now included the sample size, and details of statistical analyses in Methods section. We also add scale bars in all the figures now.

- Throughout the manuscript, correlation is conflated with causation. In fact, there are too many examples of this to list out. Some examples include:

page 5 ("vertical movement of DC cells...is guided, at least partially, by ...softened epidermis"),

RE: Thank you for pointing out the gaps that required improvement. In this revision, we have addressed several of them by performing new experiments. Specifically, we show that overexpression of Wnt3a and Snai1 are accompanied by epidermal softening and a dispersed migratory pattern of dermal cells. These results are now included in Figs. 1J–L and SI Fig. 5 (Snai1 for bud protrusion), as well as Figs. 2M–O and SI Fig. 8 (Wnt3a for invagination).

In addition, we used an actin polymerization stabilizer Jasplakinolide (A Holzinger, Methos Mol Biol. doi: 10.1007/978-1-60761-376-3_4.) to demonstrate the importance of properly regulated tissue mechanics during feather follicle development. Jasp-treatment predicably enhanced F-actin polymerization and vinculin expression in E8+2 skin explants, increased cellular forces across bud and interbud regions (indicated by laser ablation assay). As a result, the Jasp-Tx feather buds were abnormally shaped, and early dermal condensates failed to develop properly. The results are included in SI Fig 6A-D.

page 6 (:we interpret that Slug decreases epidermal stiffness while Sostdc1 increases epidermal stiffness, and DKK1 and Gremlin decrease dermal stiffness (Fig 1I, SI Fig 4B)."),

RE: We have measured the changes in epidermal and dermal stiffness in Snai1 and Wnt3a overexpressing feather buds (Fig 1J-L, SI Fig 5, Fig 2M-P, SI Fig 8). For molecules not directly evaluated (e.g., Socdc1, DKK1, Gremlin,), they are mostly moved to supplementary data. The description involved is toned down to avoid inappropriate interpretation.

page 7 ("We observed that as the feather bud continues to elongate, it stretches the epidermal cells"),

Thank you for the opportunity to clarify. We now performed new laser ablation assay to investigate the tension in the epidermal cell junctions, in protruding feather buds, interbud regions, and invaginating neck region (Fig 1F, SI Fig 4A, Fig 2H, SI Fig 7A-B). The results show significant differences in change in traction distance between protruding bud, invaginating bud neck region to interbud region.

page 8("Together these results suggest that follicle formation is achieved via stretching of the bud/ interbud boundary - indicated by a stiff epidermal wall, YAP expression and a stretched cell morphology - that triggers the initial epithelial invagination."), and several more examples.

RE: YAP expression pattern is indeed unique and informative (Fig. 2I). YAP has widely been used as a molecule used to imply mechanical activation of cells. In this revision, we included new data using RCAS dominant negative YAP, which demonstrate that YAP is required for proper feather formation, and that its downstream effector, MMP2, is negatively affected when YAP activity is inhibited (Fig. 2K).

-Along the lines of correlation vs. causation, morphological data is frequently over/misinterpreted as mechanical data. For example larger cell areas on the bud are described as indicating stretching of these cells by outgrowth of dermis. But stretching is a physical process and cannot be determined from images alone, as expansion of cells pushing the neighboring tissue would look the same as stretching of the cells by the neighboring tissue. Force measurements (laser ablation, microdissection, or other methods) are needed to make any statement on forces and their effects.

RE: We performed laser ablation assay on the epidermal cells on the protruding bud, and on invaginating epidermal cells at the bud neck, and compare their retraction behavior with that of interbud epidermal cells. These results are presented in Fig 1F, SI Fig 4A, Fig 2H, SI Fig 7A-B.

-The distinction between true results and their interpretation is overly blurred, with schematic figures appearing in place of data (Fig. 1C, L for example). Many statements in the Results should be moved to the Discussion, as they are not a direct description of results, but instead an extrapolation of those results, which are often times overreaching (frequently through equating correlation to causation). A partial list of examples include:

page 6 ("By overlaying these concentric zone maps and their known properties, we interpret that Slug decreases epidermal stiffness while Sostdc1 increases epidermal stiffness, and DKK1 and Gremlin decrease dermal stiffness (Fig 1I, SI Fig 4B). This interpretation is based on previous findings that Dkk1 and Gremlin weaken dermal condensation (Chen et al., 2015), leading to decreased dermal stiffness. Together, after the bud protrusion induced-stretching, one anticipates a feedforward connection between these molecular signaling events and the physical morphogenetic processes. These responses prelude the mechanical environments for epidermal invaginations that will take place in the next stage at E9."),

RE: We have removed this interpretation from the manuscript.

page 6 ("These results suggest that epidermal stiffness helps confine the bud shape, and the softened epidermis may allow the dermal cells to migrate laterally, leading to the expansion of bud width (Fig 1L)."),

RE: We have now provided supporting evidence demonstrating that epidermal softening, either via *Sprty2* or *Snai1* overexpressing, led to dispersed dermal cell migration pattern and widened bud width.

page 7 ("From a mechanical perspective, it is also likely that migrating cells with a certain degree of stiffness tend to invade softer regions rather than stiffer ones."),

RE: We have removed this statement from the manuscript.

page 7 ("The moderately stiff DC (0.72 ± 0.07 kPa) could also serve as the physical barrier that divides distal bud dermal cells (0.32 ± 0.03 kPa, to become pulp) from the basal dermal cells..."),

RE: We have revised this statement to the following:

"The DC (0.72 ± 0.07 kPa) is also stiffer than the distal dermal cells (0.32 ± 0.03 kPa, forming pulp) and basal dermal cells (0.44 ± 0.05 kPa, forming DP)."

page 8 ("These results could imply that, initiated by feather bud elongation induced-stretching, YAP, Wnt, MMP, and others work in concert to soften the dermis around the DC and facilitate epidermal tongue invaginations (Fig 2D, I)."), and many more.

RE: We have provided new results on cell stretching, involvement on YAP and MMP, and revised the description as following:

“These findings suggest that bud elongation-induced stretching activates YAP, MMPs, and other effectors, which in concert stiffen the epidermis and soften the dermis around the DC, thereby facilitating downward extension of the epidermal tongue (Fig. 2L).

-Many data need to be quantified, as representative images - particularly without indicating replicates - are not compelling on their own.

RE: We have addressed issues on sample size, quantifications and statistical analyses in the figure legends, and provided the p-values where applicable.

-Time lapse data are presented as a single representative arrow plot with sparsely placed arrowheads, yet these data are overinterpreted in the text, and many claims need to be tested by directly quantifying velocity fields. For, example, spatial variation in velocity cannot be readily interpreted from the data, and strain rate fields should instead be quantified. Spreading behaviors are described, but should be directly quantified as divergence of the velocity field. Disorder is described but should be quantified using one of many measures of coherence.

RE: Thank you for pointing out these deficits. We now provide Imaris analysis, numbers on cells aggregating/spreading and strain rate fields as mentioned. Please see below for the excerpt:

“Live imaging and cell Tracking Analysis. Live xyzt imaging was done with a confocal microscope (Stellaris 5, Leica) taking 50 um thick z-stacks of 10 min intervals for 18-48h using a 10x air lens. Laser power, gain, and exposure times were kept at a minimal and constant level to avoid photobleaching. The sample was incubated at 37 oC, 5% CO2 and 90% humidity chamber (Oko-Lab).

Imaris (Version 9.2. Oxford Instruments) was used for cell migration analysis. Drift correction was applied using the "Correct Drift" function in Imaris, based on the reference time point and structural alignment across time-lapse data. Background subtraction was

performed using a rolling ball algorithm (radius = 10-12 μm) to enhance signal-to-noise ratio.

Approximately 20,000 to 25,000 cells were tracked and analysed per sample in this study. Tracked particles were detected using the Spot function in Imaris based on the nuclear RCAS-mOrange signal, and all analyses are based on particle trajectories. We extracted 3D displacement vectors and positions (X, Y, Z) for each tracked particle (e.g., cell nucleus or centroid) over time, from which the magnitude and direction of displacement could be computed...”

Please refer to Methods for the full description.

-Measurements of stiffness by AFM do not appear to take into account geometric effects. Indentation testing of the same material in a flat vs curved geometry would produce differing effects, with apparent stiffness higher when indenting against the radius of curvature. However, this geometric effect is not considered, and results may therefore be incorrect.

RE: The relative curvature of the feather bud (radius $\sim 150 \mu\text{m}$ at earliest stage E8) vs AFM cantilever tip (SAA-SPH-5UM, radius = $5 \mu\text{m}$) is 30x larger and its effect is minimal to the interpretation of the results. We have added this reference in METHODS: *Influence of cantilever tip geometry and contact model on AFM elasticity measurement of cells* <https://onlinelibrary.wiley.com/doi/full/10.1002/jmr.3018>

“The AFM (NanoWizard 4a/CellHesion, JPK, Berlin, Germany) combined with a stereomicroscope was set up in contact mode indentation to measure tissues in PBS. The stereomicroscope is aligned with cantilever and sample so the location of indentation can be precisely observed. SAA-SPH-5UM AFM cantilevers (Bruker, MA, USA) of cylindrical tip with $5 \mu\text{m}$ end radius and spring constant between 0.1-0.2 N/m. The spring constants of all cantilevers were calibrated via thermal noise method with correction factor in liquid (Butt & Jaschke, 1995; Hutter & Bechhoefer, 1993) prior to each measurement. A force series identified a maximum indentation force of 5 nN to show the most consistent results on test samples. A constant rate of $1 \mu\text{m/s}$ was used for the entire approach and retract sequence.

Force-distance curves were collected and post-processed using the JPK package software (Data Processing, 6.3.11). The force curves were analyzed using the Hertz model with a spherical indentation (Lin et al, 2009; Tripathy & Berger, 2009)."

-Many aspects of the mechanics are not well described or misleading. For example, phase transitions are conflated with viscoelasticity (page 3); changes in speed or direction do not imply more fluid like behavior, as a solid can accelerate, decelerate and change directions (page 4).

RE: Thank you for pointing out these. We have revised the description to the following:

"Sculpting biological structures requires dynamic changes in stiffness contrasts (Lenne & Trivedi, 2022), reflected in changes in cellular movements and mechanical properties. Local adjustments in tissue stiffness and fluidity, in turn, generate differences that allow cells to collectively migrate or "flow," shaping the developing organ."

We have also removed the "fluid-like" behaviour statement from the manuscript.

-Stiffness changes are achieved by misexpression of genes such as Sprty2 and Snail, which will have many effects on cell behavior that extend beyond changes in tissue stiffness.

RE: To address this point, we extended our analyses beyond stiffness mapping. Specifically, we have performed time-lapsed recording and cell tracking analyses on Wnt3a (Fig 2N) and Snai1 (Fig 1L) overexpressing mutants, which revealed alterations in cell dynamics in addition to tissue stiffness.

Furthermore, Sprty2 overexpression in scales not only resulted in epidermal softening and conversion of scales to feathers, but was also associated with

downregulation of FAK, Vinculin, and actin filament expression (Fig. 5L). These findings collectively demonstrate that the effects of gene misexpression extend well beyond stiffness modulation to include broader changes in cell behavior and cytoskeletal organization.

Controls are lacking, however, to isolate effects as a result of stiffness changes as opposed to other cell behaviors.

RE: Appropriate controls were included in our analyses. Measurements on RCAS-controls were always performed in parallel with untreated samples, and there is no significant difference from that of untreated samples. There's no morphological differences in RCAS-controls samples either. Thus, the RCAS system itself does not introduce detectable artifacts.

Additionally, we used Jasp-Tx to demonstrate the importance of spatially-regulated mechanical dynamics and demonstrate the importance of spatially regulated mechanical dynamics for proper morphogenesis (SI Fig 6).

-SI Fig 2D is missing.

RE: New SI Figure arrangements are revised.

-Unclear whether RCAS infections are restricted to epidermal vs dermal compartments, or spread throughout both. If the latter, this would need to be addressed, and interpretation of results adjusted accordingly.

RE: We could control the location (dermis only, epidermis only, or both) of RCAS transfection by means of injections. We could inject the virus into dermis only, amniote sac only (to infect epidermis only), or both. In the representative figures below, we demonstrate to the reviewer that we could control and confirm the location of virus by performing RCAS staining on the transfected samples, depending on the method of injection. This is a method we have performed regularly in our laboratory and have demonstrated its efficacy over the years. These are added to MEHODS now.

-Many claims in the manuscript seem unsupported by data. For example, the claim that mechanical factors are claimed to have a role in dermal cell fate specification (page 9) or the statement "When putative DPs reach a compaction threshold, they secrete more Tgf β that trigger the bending of the epithelial tongue toward DP, In so doing, other molecules are induced in different topological positions in this newly set up proximal follicle, positioning themselves for future cyclic renewal"

RE: Thank you for pointing out the speculation. This statement has been removed and the description now reads:

"Together, these results indicate that follicle base closure is achieved through mechanically and chemically interconnected events that guide cellular flows, tissue folding, and fate specification. Mechanical and molecular cues act in concert to induce

shape changes. This process also establishes molecular patterning in distinct topological zones of the proximal follicle, positioning tissues for future cyclic renewal (Wu et al., 2021b)."

-SI Fig. 2A is referred to as SI Fig. 2B erroneously in the text (page 4)

RE: Thank you for pointing out our mistake. The figures and SI figures have been significantly revised and this mistake has been corrected.

Dear Ming,

Thank you for submitting the revised version of your manuscript to The EMBO Journal. The study has now been seen by two of the original referees, who appreciate the revisions, but also find that several of their initial points were not sufficiently addressed or clarified. In particular, they indicate that further clarifications need to be provided on the experimental design and statistical analysis. Additionally, they indicate that a more accurate description of the causality of the results and the background knowledge is needed.

Therefore, I would like to invite you to address the remaining referee comments in a final revision round. Due to the nature of the comments, the revision might need an additional re-assessment round by the reviewers.

Furthermore, during our standard image analysis, we detected potential aberrations in the figure set, and we would like to clarify these issues. We kindly invite you to check the composition of the figures below:

*Possible reuse between Figure 1E and Appendix Fig 3E - Not listed in figure legends.

*Possible reuse between Figure 1F and Appendix Fig 4A - Not listed in figure legends.

*Possible reuse between Figure 2E,F&G and Appendix Fig 3C,D&E - Not listed in figure legends.

*Possible reuse between Figure 2H and Appendix Fig 7A - Not listed in figure legends.

*Possible reuse between Figure 3A and Appendix Fig 9A - Not listed in figure legends.

*During image review, we noted a possible reuse between Figure 5E of your manuscript and Figure 2G of another publication (Liu T.-Y., Chuong C.-M., J. Dev. Biol. 2023; 11(3):30; <https://doi.org/10.3390/jdb11030030>). This earlier publication includes partial author overlap (Tzu-Yu Liu, Cheng-Ming Chuong).

If the reuse is intentional and scientifically warranted, this must be explicitly referenced in both the figure legend and the main text of the manuscript. In addition, a statement confirming approval of figure reuse must be included in the manuscript (for example, in the acknowledgements or figure legend, depending on context).

Please clarify whether this reuse is intentional and, if so, update the revised manuscript accordingly to ensure proper citation and transparency.

Furthermore, please send us all corresponding source data. If you make changes to the figure set, please include a point-by-point describing what you have changed and why. If purposeful re-use of the image has occurred, please state this clearly in the figure legend.

Image source data should be provided as one file per figure that contains the original, uncropped and unprocessed scans of all or key gels/microscopy images used in the figure. The file(s) should be labelled with the appropriate figure/panel number, and should display molecular weight markers; further annotation may be useful but is not essential. Source data files will be published online with the article as supplementary "Source Data."

Finally, there are several formatting aspects that need to be addressed in the revised version:

1. Please upload the figures as individual, high resolution figure files. Please rename the supplementary figures into Figure EV1 - EV9 and also upload them as individual figure files.

2. We are missing the ORCID iD for the corresponding author (Cheng Ming Chuong). In order to link the ORCID iD to the account in our manuscript tracking system, the author in question has to do the following:

- Click the 'Modify Profile' link at the bottom of your homepage in our system.

- On the next page you will see a box halfway down the page titled ORCID*. Below this box is red text reading 'To Register/Link to ORCID, click here'. Please follow that link: you will be taken to ORCID where you can log in to your account (or create an account if you don't have one)

- You will then be asked to authorise the access to your ORCID information. Once you have approved the linking, you will be brought back to our manuscript system.

Unfortunately, we cannot do this linking on the author's behalf for security reasons.

3. We require a "Data Availability" section at the end of "Methods" section. As far as I can see, no data deposition in external databases is needed for this paper. If I am correct, then please state in this section: This study includes no data deposited in external repositories. Further information can be found at

<https://www.embopress.org/page/journal/14602075/authorguide#dataavailability>

4. CRedit has replaced the traditional author contributions section because it offers a systematic, machine-readable author contributions format that allows for more effective research assessment. Please remove the Authors Contributions from the manuscript and use the free text boxes beneath each contributing author's name in our online submission system to add specific details on the author's contribution. More information is available in our guide to authors.

5. Please add a "Disclosure and competing interests statement" section after "Acknowledgments" (further information on this section: <https://www.embopress.org/page/journal/14602075/authorguide#conflictsofinterest>).

6. Please rename the videos into Movie EV1-EV5 and update the callouts accordingly. The legends should be removed from the manuscript text file and zipped with each movie file. Further information is available here:

<https://www.embopress.org/page/journal/14602075/authorguide#expandedview>. Please remove the file "Description for

supplementary videos".

7. Please rename the file "Supplementary information of model in detail" to "Appendix" and add a title page with a brief table of contents to the file, making sure to include page numbers. Please replace the heading "Long Version for Supplemental Information" with "Appendix Supplementary Methods".

8. Please upload sirce data as one folder per figure.

9. Please cite the main figures and EV figures in sequential order, i.e. all panels of Figure 2 should be cited after Figure 1, etc.

10. Please rename the "Summary" section "Abstract" and replace the "Main text" heading with "Introduction".

11. Our data editors have flagged the following issues in figure legends that need correcting:

- Please indicate the statistical test used for data analysis in the legend of figure 1F.
- Please provide information on the number and nature of replicates in the legend of figure 1F.
- Please define the error bars in the legend of figure 1F.
- Please define the yellow arrows in the legend of figure 5E.

12. Papers published in The EMBO Journal are accompanied online by a 'Synopsis' to enhance discoverability of the manuscript. It consists of A) a short (1-2 sentences) summary of the findings and their significance, B) 3-4 bullet points highlighting key results and C) a synopsis image that is 550x300-600 pixels large (width x height, jpeg or png format). You can either show a model or key data in the synopsis image. Please note that the image size is rather small and that text needs to be readable at the final size. Please send us this information together with the revised manuscript.

We generally allow three months as standard revision time. Should you foresee a problem in meeting this deadline, please let us know in advance to discuss an extension.

As a matter of policy, competing manuscripts published during this period will not negatively impact on our assessment of the conceptual advance presented by your study. However, please contact me as soon as possible upon publication of any related work to discuss the appropriate course of action.

When preparing your letter of response to the referees' comments, please bear in mind that this will form part of the Review Process File and will therefore be available online to the community. For more details on our Transparent Editorial Process, please visit our website: <https://www.embopress.org/page/journal/14602075/authorguide#transparentprocess>. Please also see the attached document with instructions for guidelines on preparation of the revised manuscript.

Please feel free to contact me if have any questions regarding this final revision. Thank you again for giving us the chance to consider your manuscript for The EMBO Journal. I look forward to receiving the final version.

With best wishes,

Ieva

We realize that it is difficult to revise to a specific deadline. In the interest of protecting the conceptual advance provided by the work, we recommend a revision within 3 months (17th Feb 2026). Please discuss the revision progress ahead of this time with the editor if you require more time to complete the revisions.

Referee #1:

The submission and accompanying response make the work easier to assess.

I have several points related to the data and arguments that the authors should address:

1. How accurate is the automated cell tracking that was done? Can the authors validate this in some way? Or is it more like particle image velocimetry (I do not mean that this was done, just that low accuracy tracking may approximate this process)? The accuracy would be important to know and indicate to the reader, though even low accuracy of tracking might well be suitable for the tissue level behaviours under study.

2. How much active cell movement is actually going on during these processes?

'Flow' is not defined - is this all movement of cells that takes place, whether active migration of individual cells or passive movement driven by expansion processes taking place elsewhere, such as distant proliferation or ECM expansion? If active migration is observed and is what the model the authors present intends then it would be good to indicate this more clearly in the text and title.

3. What is the role of cell proliferation in these processes? Can the videos identify dividing cells that split into two, and where this is happening in the tissues? It would be good to show an example of this in an annotated video in the supplements. As presented, cell proliferation seems not to be a driving part of the process but the authors should specify their view of the relationship between cell movements and the extent and locations of cell divisions.

The manuscript retains assertions, errors and very tenuous links that are not warranted and the authors need to find and correct these. An example is the assertion that FZD8, a recognised WNT receptor, is a TGF-beta receptor (page 8 "Tgf-β2 is expressed in the basal dermis (Fig. 5E), but its receptor FZD8 is not strongly expressed."). The reference given earlier (Spanjer et al., 2016) does not show this, instead it shows that the effect of Tgf-B2 on human and mouse lung cells (not the prostate as stated on page 9 "In the prostate gland, Tgf-β signaling is partly regulated by the Wnt receptor Fzd8 (Spanjer et al, 2016)" is partly mediated through FZD8 as a target gene, which acts as a receptor for WNTs.

The legend for Figure 3H states that it shows stages "E14, E11, E17 and newborn flight feathers" but E11 is not shown in the panels, only the other three stages.

The legend for SI Figure 6D states that it shows "FAK, vinculin and phalloidin expression" but it does not show FAK detection at all. Also, phalloidin is not 'expressed' as it is a detection reagent, so this should instead be 'Actin detection' or similar.

The authors must remove all of these errors as they reflect badly on them and the journal (and reviewers) if this were to be published.

Referee #3:

The authors have been highly responsive to reviewer critiques, and the manuscript is significantly improved in clarity and completeness. The addition of laser ablation to confirm stretching rather than reliance on cell morphology, for example, demonstrates a significant improvement in the manuscript. However, some key concerns remain only partially addressed. These relate most pressingly to data presentation and statistical rigor of statements made throughout the manuscript. Details are below.

1. Data quantitation and statistics remain only partially addressed. When the same control is being compared to more than one experimental group, a t-test is not appropriate, but instead a one-way ANOVA is needed. When effect of more than one independent variable is considered (e.g. effects of stage and location, effects of treatment at different stages, comparison of bud vs. peri-bud in whole and different tissue compartments), a two way ANOVA with appropriate post-hoc tests are needed.

2. The exact number of biological replicates must be indicated for each experiment, ideally in the figure legend or figure itself. The revised manuscript includes statements in the Methods that broadly state for a given assays that, for example, more than five measurements were averaged across more than three biological replicates. This is not sufficient, and precise replicate numbers need to be added to each piece of quantitative data.

3. AFM data in SI Fig. 2B has no reported p-values or significance symbols, but includes statements in the text (page 5) establishing differences in stiffness between tissues and regions. These statements must be supported by statistical tests.

4. In-text statements on AFM results, including controls (page 5) and Snai1 and Sprty2 o/e phenotypes are also lacking replicate numbers and p values.

5. RCAS-controls are not included for comparison to RCAS-Snai1, RCAS-Sprty2, RCAS-Wnt3a cell trajectories or stiffness maps. Each experiment must have internal controls for comparison, but these are generally not shown.

6. Each qualitative statement in the Results should include "(n=/_)" indicating the number of biological replicates in which the statement held true out of the total number examined.

7. Correlation/causation and presentation/interpretation of results continue to be blurred. Again taking AFM measurements on page 5 as an example, "These findings suggest that the vertical movement of DC cells...is guided, at least partially, by this locally softened epidermis." This statement is misleading: findings suggest that vertical movement is *correlated with* locally softened epidermis. Or alternatively, that vertical movements *may be* guided by softening of the epidermis. Care needs to be taken throughout the manuscript to present the data as it is, recognizing that over-interpretation or conflating correlation with causation introduces bias.

8. While additional information has been added to methods on the AFM assay, it does not address the original concern raised regarding geometric effects, which were not related to effects of tip geometry but of the sample being tested. Indentation of a curved surface will produce different stiffness measurements than a flat surface of the same material. More generally, indentation measurements of material properties will vary as a function of material curvature, with higher curvatures producing

artificially higher modulus measurements. Typically, this is addressed by mechanics arguments (either analytical or via finite element simulations) indicating that for a particular indenter size and shape, measured properties will not differ when the substrate being investigated has different curvatures. Given how dramatically the bud curvature changes and how it differs from interbud regions, this would seem to be an important issue to address.

9. Cell shape and orientation data obtained from QMorF are insightful but are not in-and-of-themselves an indication of forces or force fields. Laser ablation experiments verify some of the interpretations made from QMorF data, but language throughout the Results continues to conflate morphological information with force/mechanical processes, which is misleading.

10. Method details on MMP-FRET are not provided.

Response letter to the editor and reviewers

We thank the Editor and Reviewers for their careful evaluation of our manuscript and for the constructive and insightful comments. We have revised the manuscript extensively and to the best of our ability in response to all points raised, with the aim of improving appropriate statistical analysis, data presentation and interpretation. Below, we provide a detailed, point-by-point response **in blue** outlining how each comment has been addressed, and we believe that these revisions have significantly strengthened the manuscript. The changes made to the manuscript is **highlighted in yellow** for clarity, and a clean manuscript file and another containing most of the changes in yellow highlights are included in our submission package.

We have also modified the title to “Novel Tissue Mechanics-Guided Cellular Flows Drive the Formation of Feather Follicles.”

Referee #1:

The submission and accompanying response make the work easier to assess.

I have several points related to the data and arguments that the authors should address:

How accurate is the automated cell tracking that was done? Can the authors validate this in some way? Or is it more like particle image velocimetry (I do not mean that this was done, just that low accuracy tracking may approximate this process)? The accuracy would be important to know and indicate to the reader, though even low accuracy of tracking might well be suitable for the tissue level behaviours under study.

RE: We used Imaris' automated cell tracking framework, which links individual nuclei based on centroid positions rather than PIV-based flow estimation. The tracking accuracy is defined as correct object-to-object linking between consecutive frames without identity switching. In other words, it is based on linking designated objects, rather than pixel-level localization accuracy.

Imaris' Autoregressive Motion algorithm predicts object displacement based on motion in the immediately preceding time point, thereby limiting unrealistic jumps and erroneous track linking.

Using bud formation video as an example. The diameter of a dermal fibroblast is approximately 8 μm , and the estimated diameter of the automated assigned particle was 14.8 μm , while the maximum allowed inter-frame displacement was set to 8.35 μm , ensuring that candidate links exceeding half of a nuclear diameter were rejected. Gap filling was disabled to avoid artificial trajectory interpolation.

We also manually inspected trajectories were across representative time points and cell densities to confirm continuous motion without frame-to-frame jumping or identity switching, and we only had to correct image drifting. This indicates that automated tracking was reliable under the chosen parameters.

The same methods were used for other stages of feather bud formation, and the parameters are provided as follows:

Bud formation

Estimated Diameter = 14.8 μm
Algorithm Name = Autoregressive Motion
MaxDistance = 8.35 μm
MaxGapSize = 3
Fill Gap Enable = false

Invagination

Estimated Diameter = 22.7 μm
Algorithm Name = Autoregressive Motion
MaxDistance = 19.3 μm
MaxGapSize = 3
Fill Gap Enable = false

DP formation

Estimated Diameter = 22.7 μm
Algorithm Name = Autoregressive Motion
MaxDistance = 15.7 μm
MaxGapSize = 3
Fill Gap Enable = false

Scutate Scale

Estimated Diameter = 9.10 μm
Algorithm Name = Autoregressive Motion
MaxDistance = 6.15 μm
MaxGapSize = 3
Fill Gap Enable = false

The below description has been added to “Live imaging and cell tracking analysis” of methods:

“Imaris uses object-based tracking of individual nuclear centroids. Nuclei were tracked using the Autoregressive Motion algorithm, which models object displacement based on motion in the immediately preceding time point. Each nucleus is designated as a particle for tracking. In bud formation, invaginations, DP and scutate scale videos, the mean particle diameter was estimated to be 14.8 μm , 22.7 μm , 22.7 μm and 9.10 μm , while the maximum allowed inter-frame displacement was set to 8.35 μm , 19.3 μm , 15.7 μm and 6.15 μm , respectively. These parameters minimize erroneous linking between neighboring nuclei. Maximum gap size allowed between frames were set a 3, and gap filling was disabled to avoid artificial interpolation of trajectories. Tracking results were manually inspected to confirm continuity of motion and absence of identity switching.”

2. How much active cell movement is actually going on during these processes?

'Flow' is not defined - is this all movement of cells that takes place, whether active migration of individual cells or passive movement driven by expansion processes taking place elsewhere, such as distant proliferation or ECM expansion? **If active migration is observed and is what the model the authors present intends then it would be good to indicate this more clearly in the text and title.**

In this study, we define “flow” as follows, which has been added into introduction:

"Here, we define 'cellular flows' as the collective, spatially organized movement of cell populations during morphogenesis, encompassing both active migration (cell-autonomous, cytoskeleton-driven translocation) and passive displacement (movement resulting from mechanical forces, tissue deformation)."

We believe both active and passive cell movements are involved, and they are stages/context dependent. For example, during bud protrusion, we contend that the coordinated dynamics of both active migration of the dermal cells and passive stretching of the epidermal cells are required for successful protrusion. On the other hand, the movement within the feather bud and scutate scale is also "passively" guided by the epidermal follicle/scale walls.

The active movement of dermal cells is supported by:

Blebbistatin (Myosin II inhibitor) treatment: Led to "sac-like bud morphology" because dermal cells failed to migrate vertically (SI Fig 6E).

ADH-1 (N-cadherin inhibitor) treatment: Resulted in shorter buds, implying adhesion is needed for the traction forces required for active migration (SI Fig 6F).

Accordingly, we have made the following changes in results and discussion:

"The results show that ablated membrane at the protruding bud retracted significantly faster than that of the interbud region (Fig 1F), **implying epidermal cells at the bud tip experience greater passive stretching force from the actively protruding dermal cells than those at the interbud.**"

"This led to a sac-like bud morphology. This may be due to the failure of dermal cells to **actively** migrate vertically upward and create reciprocal forces at the epidermis (Fig 1M, SI Fig 4C)."

"Together, these results show feather bud protrusion is achieved through a sequence of spatial biomechanical-biochemical interactions and characterized by dermal cells to **actively migrate** vertically and upward, while keeping the interbud skin in place."

"Stage 1: Patterning. Dermal condensation increases stiffness at its center. These condensed cells must displace—either downward into the dermis (as in hair follicles) or upward (as in feathers). **In feathers, the placodal epidermis softens and deforms, as shown by AFM measurements and QMorF analysis, allowing dermal cells to actively move vertically upward and push out the feather bud.**"

"Stage 2: Invagination. As the bud elongates, the ring-shaped basal epithelial cells are **passively** stretched, evidenced by changes in cell shape and YAP and MMP expression. This stretching drives the surrounding epithelium, organized in a cylindrical wall, to invaginate into the dermis."

The role of cell proliferation is discussed in next reviewer's comment.

3. What is the role of cell proliferation in these processes? Can the videos identify dividing cells that split into two, and where this is happening in the tissues? It would be good to show an example of this in an annotated video in the supplements. As presented, cell proliferation seems not to be a driving part of the process but the authors should specify their view of the relationship between **cell movements and the extent and locations of cell divisions**.

RE: Proliferation of the dermal cells within the feather bud is observed but the number is not significant compared to the overall increase in cell number.

We quantified the change in number of nuclei in our E8+18h videos, and saw an average of 695 cells (n=3), yet only identified on average 48 cell division counts. This suggests that cell division within the feather bud accounted for less than 7% of the overall cell number, and should not play a major role in driving cell migration.

Interestingly, the location of proliferation occurs mostly at the basal posterior end of the feather bud (circle),

which corresponds to previous reports (Chen CF et al., *Annu Rev Anim Biosci*, 2017). Others have also reported that “dermal or mesenchymal condensate... its mode of formation is through cell recruitment rather than local proliferation” (Riddell and Headon, *Developmental Biology*, 2025), and “Directional Cell Migration, but Not Proliferation, Drives Hair Placode Morphogenesis” (Ahtiainen L et al., *Developmental Cell*, 2014). On the other hand, proliferation of the epidermal cells does contribute to the elongation and invagination (E9-12), as reported (Chen CF et al., *Annu Rev Anim Biosci*, 2017).

We added the following text to results:

“To evaluate whether cell proliferation contributes significantly to feather bud protrusion, we quantitatively analyzed dermal cell division during the E8+18 h time window. We found that the location of proliferation occurs mostly at the basal posterior end of the feather bud (circle), and detected less than 7% of total cells showing cell division activity (SI Fig 2D-F, averaged 48 events in 695 nuclei, n=3), indicating that local dermal proliferation plays a minor role. This is consistent with prior reports that proliferation is spatially restricted to the basal posterior region of the protruding bud, and dermal condensate formation is driven predominantly by directed cell migration rather than local cell division (Riddell and Headon, *Developmental Biology*, 2025; Ahtiainen L et al., *Developmental Cell*, 2014.”

The manuscript retains assertions, errors and very tenuous links that are not warranted and the authors need to find and correct these. An example is the assertion that FZD8, a recognised WNT receptor, is a TGF-beta receptor (page 8 “Tgf-β2 is expressed in the basal dermis (Fig. 5E), but its receptor FZD8 is not strongly expressed.”). The reference given earlier (Spanjer et al., 2016) does not show this, instead it shows that the effect of Tgf-B2 on human and mouse lung cells (not the prostate as stated on page 9 “In the prostate gland, Tgf-β signaling is partly regulated by the Wnt receptor Fzd8 (Spanjer et al, 2016)” is partly mediated through FZD8 as a

target gene, which acts as a receptor for WNTs.

RE: Corrected:

Page 9:

"In the prostate gland, Tgf- β signaling is partly regulated by the Wnt receptor Fzd8 (Spanjer et al, 2016)."

Is changed to:

"In the prostate gland and lung fibroblasts, the WNT receptor Fzd8 physically interacts with TGF- β receptors to modulate TGF- β -induced profibrotic signaling (Spanjer et al., 2016)."

Page 11:

"Tgf- β 2 is expressed in the basal dermis (Fig. 5E), but its receptor FZD8 is not strongly expressed."

Is changed to:

"Tgf- β 2 is expressed in the basal dermis (Fig. 5E), but FZD8, a WNT receptor that can modulate TGF- β signaling through receptor complex formation (Spanjer et al., 2016), is not strongly expressed."

The legend for Figure 3H states that it shows stages "E14, E11, E17 and newborn flight feathers" but E11 is not shown in the panels, only the other three stages.

RE: Corrected. The legend is changed to:

"FZD8, TNC, CTNNB1, NOG, SOSTDC1, Tgf- β 2, YAP (longitudinal section) and SMA expression in E14, E17 and newborn (NB) flight feathers and particularly around dermal papillae region."

The legend for SI Figure 6D states that it shows "FAK, vinculin and phalloidin expression" but it does not show FAK detection at all. Also, phalloidin is not 'expressed' as it is a detection reagent, so this should instead be 'Actin detection' or similar.

RE: Corrected. The legend is changed to:

"Representative results of vinculin and phalloidin (F-actin) detection in E8 \pm 2d control and Jasp-Tx quail feather buds. Vinculin is a focal adhesion protein, and phalloidin labels F-actin. Scale bar 100 μ m."

The authors must remove all of these errors as they reflect badly on them and the journal (and reviewers) if this were to be published.

Referee #3:

The authors have been highly responsive to reviewer critiques, and the manuscript is significantly improved in clarity and completeness. The addition of laser ablation to confirm stretching rather than reliance on cell morphology, for example, demonstrates a significant improvement in the manuscript. However, some key concerns remain only partially addressed. These relate most pressing to data presentation and statistical rigor of statements made throughout the manuscript. Details are below.

1. Data quantitation and statistics remain only partially addressed. When the same control is being compared to more than one experimental group, a t-test is not appropriate, but instead a **one-way ANOVA is needed**. When effect of more than one independent variable is considered (e.g. effects of stage and location, effects of treatment at different stages, comparison of bud vs. peri-bud in whole and different tissue compartments), **a two way ANOVA with appropriate post-hoc tests are needed**.

RE: We appreciate the critical and constructive comments. One-way or two-way ANOVA with appropriate post-hoc tests have been performed and updated to figure panels as follows:

Fig 1F:

2-way ANOVA, $F(1, 4) = 9.448$, * $P = 0.0372$, $N=3/3$

SI Fig 2B:

2-way ANOVA

Whole, bud vs peribud: $F(1, 8) = 304.7$ ****, <0.0001

Dermis, bud vs peribud: $F(1, 8) = 35.79$ ***, $P = 0.0003$

Epidermis, bud vs peribud: $F(1, 8) = 37.66$ **, $P = 0.0031$

Dermis bud vs epi bud, $F(1, 8) = 11.47$, ** $p = 0.0095$

Dermis peribud vs epidermis peribud: $F(1, 8) = 45.76$, *** $p = 0.0001$

$N = 9/5$

SI Fig 6C:

2-way ANOVA

Ctl vs Jasp bud protrusion: $F(1.177, 4.707) = 12.15$, * $P = 0.0179$

Ctl vs Jasp interbud: $F(2.021, 8.084) = 7.817$, * $P = 0.0128$

Ctl bud vs interbud: $F(1, 4) = 9.448$, * $P = 0.0372$

$N = 3/3$

SI Fig 6E:

T-test, $p < 0.001$ for all 3 comparisons.

$N=10/3$

SI Fig 7B:

2-way ANOVA: Bud neck vs interbud, $F(1.422, 5.687) = 7.419$, * $P = 0.0308$

$N = 3/3$

SI Fig 9B:

One way ANOVA post hoc Tukey's. $F = 5.974$, 0 vs 36h, $p = 0.0223$

$N = 4/4$

2. The exact number of biological replicates must be indicated for each experiment, ideally in the figure legend or figure itself. The revised manuscript includes statements in the Methods that broadly state for a given assays that, for example, more than five measurements were averaged across more than three biological replicates. This is not sufficient, and precise replicate numbers need to be added to each piece of quantitative data.

RE: All the measurements and biological replicates were added to figure legends and methods.

3. AFM data in SI Fig. 2B has no reported p-values or significance symbols, but includes statements in the text (page 5) establishing differences in stiffness between tissues and regions. These statements must be supported by statistical tests.

RE: Statistical analyses have been performed for stiffness measurements taken from 9 feather buds across 5 embryo replication (N = 9/5) and indicated in SI Fig 2B: 2-way ANOVA, Whole, bud vs peribud: $F(1, 8) = 304.7$ ****, $P < 0.0001$. Dermis, bud vs peribud: $F(1, 8) = 35.79$ ***, $P = 0.0003$. Epidermis, bud vs peribud: $F(1, 8) = 37.66$ **, $P = 0.0031$. Dermis bud vs epi bud, $F(1, 8) = 11.47$, ** $p = 0.0095$. Dermis peribud vs epidermis peribud: $F(1, 8) = 45.76$, *** $p = 0.0001$. N= 9/5

4. In-text statements on **AFM results**, including controls (page 5) and **Snai1** and **Sprty2 o/e** phenotypes are also lacking replicate numbers and p values.

RE: We have added p values and replicates for control, Snai1 and Sprty2 o/e expression as follows:

For normal development:

“The stiffness heatmaps demonstrate that the whole skin feather bud is more rigid (0.273 ± 0.031 kPa) than the surrounding peri-bud region (0.187 ± 0.019 kPa, Fig 1D, left, SI Fig 2B, $P < 0.001$, $n=9/5$). We then separate the epidermis and dermis to examine their contributions to tissue rigidity separately. The dermis shows a similar stiffness difference, as well as an overall decrease in stiffness (bud: 0.239 ± 0.022 kPa, inter-bud: 0.178 ± 0.019 kPa, SI Fig 2B, $P < 0.001$). Interestingly, the peeled epidermis shows an opposite stiffness map. The bud region is softer (0.189 ± 0.017 kPa), while the inter-bud region is stiffer (0.209 ± 0.021 kPa) than that of the dermis (Fig 2E, SI Fig 2B, $F(1, 8) = 37.66$, $P = 0.0031$ $n=9/5$).”

E E8.5 RCAS-Snai1

For Snai1, a new graph (SI Fig 5E) has been added to show the difference and statistical differences.

“AFM stiffness map also showed that RCAS-Snai1 overexpressed feather bud is composed of a significantly softer epidermis layer (491 ± 34 Pa) than the dermis (680 ± 42 Pa, SI Fig 5E, $P = 0.0007$, $N=3/3$).”

For Sprty2, a new graph (SI Fig 5G) has been added to show the difference and statistical differences between control and Sprty2-o/e.

G Epidermal Stiffness

“By overexpressing Sprty2, we observed a visibly widened distal end and a shorter bud length in feather buds (Fig 1M), and a decrease in epidermal stiffness (from 1325 ± 180 to 934 ± 102 Pa, Fig 1N, SI Fig 5F-G, $P = 0.0109$, $N=3/3$).”

5. RCAS-controls are not included for comparison to RCAS-Snai1, RCAS-Sprty2, RCAS-Wnt3a **cell trajectories** or **stiffness maps**.

Each experiment must have internal controls for comparison, but these are generally not shown.

RE: We have now included E8.5 RCAS-Ctl stiffness heatmap (SI Fig 5F. N=3/3) to be compared to *Snai1* and *Sprty2* o/e, and E9.5 RCAS-Ctl stiffness heatmap (SI Fig 8C. N=3/3) to be compared to *Wnt3a* o/e. For cell tracking, we included E9+18h RCAS-Ctl (SI Fig 5A) to be compared to *Snai1* and *Wnt3a* o/e. Additionally, we also created a graph comparing the stiffness of epidermis and dermal condensate of E9.5 *Wnt3a*-o/e bud (SI Fig. 8D: T-test, $P = 0.0048$. N=3/3, biological replicates).

F E8.5 RCAS-Ctl Stiffness Heatmap

6. Each qualitative statement in the Results should include "(n= _/_)"

"(n= _/_) indicating the number of biological replicates in which the statement held true out of the total number examined.

RE: (n= _/_) and nature of replicates have been added in results and figure legends.

7. Correlation/causation and presentation/interpretation of results continue to be blurred. Again taking AFM measurements on page 5 as an example, "These findings suggest that the vertical movement of DC cells...is guided, at least partially, by this locally softened epidermis." This statement is misleading: findings suggest that vertical movement is *correlated with* locally softened epidermis. Or alternatively, that vertical movements *may be* guided by softening of the epidermis. Care needs to be taken throughout the manuscript to present the data as it is, recognizing that over-interpretation or conflating correlation with causation introduces bias.

RE: Corrected. In page 5:
 "These findings suggest that the vertical movement of DC cells...is guided, at least partially, by this locally softened epidermis."
 Is changed to:
 "These findings demonstrate a spatial correlation between locally softened epidermis and vertical DC movement. Functional experiments with *Snai1* overexpression (Fig 1J-L), which further softens epidermis and alters dermal flow patterns, support this mechanical guidance mechanism."
 Various other parts of the manuscript were also revised.

8. While additional information has been added to methods on the AFM assay, it does not address the original concern raised regarding geometric effects, which were not related to effects of tip geometry but of the sample being tested. Indentation of a curved surface will produce different stiffness measurements than a flat surface of the same material. More generally, indentation measurements of material properties will vary as a

function of material curvature, with higher curvatures producing artificially higher modulus measurements. Typically, this is addressed by mechanics arguments (either analytical or via finite element simulations) indicating that for a particular indenter size and shape, measured properties will not differ when the substrate being investigated has different curvatures. Given how dramatically the bud curvature changes and how it differs from interbud regions, this would seem to be an important issue to address.

RE: Tissue surface curvature can affect apparent stiffness measurements, with higher curvatures potentially producing artificially elevated Young's modulus values. To address this concern, we approached from 2 angles: 1) tip radius to sample curvature radius ratio, and 2) measurement on inclined surfaces.

We first measured the curvature of the E8 feather bud using its longitudinal H&E image, and calculated the average bud tips radius (R_{sample}) to be $\sim 50\text{-}80\ \mu\text{m}$. In comparison to the AFM tip radius ($R_{\text{tip}} = 5\ \mu\text{m}$) used in this study, it is $\sim 10\text{-}16\times$ smaller than the smallest measured curvature (bud tips, $R_{\text{sample}} \approx 50\text{-}80\ \mu\text{m}$), hence the $R_{\text{tip}} \ll R_{\text{sample}}$ (ratio < 0.1). We also maintained maximum indentation depths of $< 1\ \mu\text{m}$ (controlled by 5 nN force limit), ensuring indent depth \ll both R_{tip} and R_{sample} , minimizing geometric artifacts.

Additionally, if tissue curvature alone were responsible for elevated modulus values, one would expect the most curved regions (bud tips) to consistently appear stiffer. Instead, we observe stage-dependent and tissue-specific patterns, including locally softened epidermis overlying curved bud tips during early protrusion. This is therefore inconsistent with a purely geometric artifact and instead reflects changes in tissue mechanics.

Lastly, a recent finite element-based and analytical studies explicitly addressing AFM indentation on curved biological substrates indicate that when the sample is measured at an inclination of 34 degrees, the measured Young's modulus is underestimated by 24%. However, between 20 and 34 degrees of incline, the experimental observations show less underestimation (Nassim Ahmine et al., Scientific Reports, 2024). We also measured the tissue angle of E8 bud vs adjacent interbud region, and it averaged to be less than 34 degrees (~ 33.4 degrees).

For invagination and DP formation analyses (Fig 2D, 3F), the tissues lie flat longitudinally, eliminating curvature concerns.

The following text is added into methods under "Accounting for surface curvature effects in AFM measurements"

We measured the curvature of the E8 feather bud using its longitudinal H&E image, and calculated the ratio of average bud tips radius (R_{sample} , $\sim 50\text{-}80\ \mu\text{m}$, $n=5$) to AFM tip radius ($R_{\text{tip}} = 5\ \mu\text{m}$), and ensure the ratio is < 0.1 . We also maintained maximum indentation depths of $< 1\ \mu\text{m}$ (controlled by 5 nN force limit), ensuring indent depth is much less than both R_{tip} and R_{sample} , minimizing geometric artifacts.

We also measured the tissue angle of E8 bud vs adjacent interbud region, and it averaged to be less than 34 degrees (~ 33.4 degrees, $n=5$), which would minimize the underestimated effects of using a spherical tip on a tilted biological substrate (Nassim Ahmine et al., Scientific Reports, 2024)

9. Cell shape and orientation data obtained from QMorF are insightful but are not in-and-of-themselves an indication of forces or force fields. Laser ablation experiments verify some of the interpretations made from QMorF data, but language throughout the

Results continues to conflate morphological information with force/mechanical processes, which is misleading.

RE: We have modified the results and figure legends according to the suggestions as follows:

In Results:

To evaluate the cell shape changes associated with early chemo-mechanical remodeling at the bud base, we performed QMorF analysis of regional cellular deformation (Fig. 2E–G).

These findings suggest that cell shape anisotropy detected by QMorF correlates with stored mechanical forces (Fig 2H, SI Fig 7A-B).

In Fig 1E and 2F:

1E. QMorF analysis showing the spatial patterns of cell deformation in epidermal cells in terms of cell area and aspect ratio during feather bud protrusion.

2F. QMorF analysis of panel E showing the spatial patterns of cell deformation in epidermal cells during feather follicle invagination. Arrows indicate directions of cell deformation, which are consistent with principal stress directions.

"To validate that QMorF-detected cell shape changes reflect underlying mechanical forces, we performed laser ablation at regions showing high cell aspect ratio (stretched cells) and low aspect ratio (compressed cells). Ablation-induced recoil velocities were 2.8-fold higher in stretched regions ($7.21 \pm 0.89 \mu\text{m}$ at 100s post-ablation) compared to compressed regions ($2.58 \pm 0.45 \mu\text{m}$; $p=0.002$, t-test), confirming that cell shape anisotropy detected by QMorF correlates with stored mechanical forces (Fig 2H, SI Fig 7A-B)."

10. Method details on MMP-FRET are not provided.

RE: Method details of MMP-FRET have been added to "FRET-Biosensor, imaging and analysis" of Methods section as follows:

FRET-Biosensor, imaging and analysis. MMP14-FRET biosensor (Chung et al, 2015) is a kind gift from Yingxiao Wang's lab (previous in UCSD and now in USC). The lentivirus-backboned (FUGW, plasmid #14883, Addgene) MMP14-FRET were cloned by replacing the EGFP segment with the MMP14-FRET vector. The biosensor consists of CFP and YFP separated by an MMP14-cleavable peptide linker. MMP14 protease activity cleaves the linker, separating the fluorophores and reducing FRET efficiency. The virus was generated as described in Harn et al. (Harn et al., 2021). The virus was injected into the chicken somite at E3. The explant was harvested on the desired embryonic day and harvested on the culture insert for imaging.

Imaging protocol is adopted from Chung e al (Chung et al., 2015). Briefly, the Leica Stellaris 5 confocal microscope with 10× air objective was used, with an incubation chamber set at 37°C. Laser excitation was set at 453 nm laser for CFP excitation, and emission channels were set at 475-500 nm for CFP (donor) and 545-570 nm for YFP (acceptor). Laser power was set at 5% to minimize photobleaching. Z-stacks were set at 50 μm depth, 5 μm step size.

FRET ratio was calculated as:

$$\text{FRET ratio} = I_{\text{YFP}} / (I_{\text{CFP}} + I_{\text{YFP}})$$

where I_{CFP} and I_{YFP} are background-subtracted intensities in respective channels. Background was measured from regions outside tissue and subtracted from all images.

FRET ratio values were normalized to control (apteric) regions, which show minimal MMP activity (SI Fig 7C, FRET ratio = 0.1-0.5).

Statistical analysis:

FRET ratio heatmaps represent average of 3 feather buds from 3 biological replicates (n=3/3). Regions with FRET ratio >1.2 (>4-fold above apteric control) were classified as MMP-positive. Statistical comparisons used t-test comparing bud base and apteric regions.

Dear Dr. Chuong,

Thank you for submitting the revised version of your manuscript. I sincerely apologise for the protracted assessment of your revised manuscript due to delays in reviewer comment submission and the unusually high number of submissions that we receive at the moment.

I have received input from two of the original reviewers, who now find that their main concerns have been sufficiently addressed, with reviewer #1 now requesting only minor textual changes.

Therefore, I am now happy to accept your manuscript for publication in principle, pending correction of the appropriate references as highlighted by reviewer #1. Furthermore, some editorial textual changes will need to be included in the final version, which I will share shortly in a follow-up letter.

Please let me know if you have any questions at this point. Please use the link below to upload the revised files once you have received the full outline of the editorial requests.

Thank you again for giving us the chance to consider your manuscript for The EMBO Journal. I look forward to working with you on finalising the manuscript for publication.

With best wishes,

Ieva

We realize that it is difficult to revise to a specific deadline. In the interest of protecting the conceptual advance provided by the work, we recommend a revision within 3 months (3rd Jun 2026). Please discuss the revision progress ahead of this time with the editor if you require more time to complete the revisions.

Referee #1:

The authors have responded appropriately to my points and suggestions.

Some points that are not core to the message of the paper:

1. The authors add the text and references "dermal condensate formation is driven predominantly by directed cell migration rather than local cell division (Riddell and Headon, *Developmental Biology*, 2025; Ahtiainen L et al., *Developmental Cell*, 2014." but the Ahtiainen et al paper from 2014 is about the placode, not the dermal condensate.

2. Page 9. They have corrected, or altered, the text to "In the prostate gland and lung fibroblasts, the WNT receptor Fzd8 physically interacts with TGF- β receptors to modulate TGF- β -induced profibrotic signaling (Spanjer et al., 2016)." but I have gone back to Spanjer et al 2016 today and can not find references to prostate (including a CTRL+F search for 'prost') and I can not see in that paper where they demonstrate a physical interaction between Fzd8 protein and TGF-b receptors. The paper reports TGF-b induction of Fzd8 and Wnt5b gene expression, and then Wnt5b protein binds to and forms a complex with Fzd8 for non-canonical Wnt signalling.

Referee #3:

My major concerns have been addressed.

Response letter to the editor and reviewer

We appreciate the comments from the editor and reviewer. We have revised the manuscript accordingly. Please see the response below in blue.

Referee #1

1. The authors add the text and references "dermal condensate formation is driven predominantly by directed cell migration rather than local cell division (Riddell and Headon, *Developmental Biology*, 2025; Ahtiainen L et al., *Developmental Cell*, 2014." but the Ahtiainen et al paper from 2014 is about the placode, not the dermal condensate.

RE: The reference Ahtiainen et al. has been removed from the manuscript.

2. Page 9. They have corrected, or altered, the text to "In the prostate gland and lung fibroblasts, the WNT receptor Fzd8 physically interacts with TGF- β receptors to modulate TGF- β -induced profibrotic signaling (Spanjer et al., 2016)." but I have gone back to Spanjer et al 2016 today and can not find references to prostate (including a CTRL+F search for 'prost') and I can not see in that paper where they demonstrate a physical interaction between Fzd8 protein and TGF-b receptors. The paper reports TGF-b induction of Fzd8 and Wnt5b gene expression, and then Wnt5b protein binds to and forms a complex with Fzd8 for non-canonical Wnt signalling.

RE: We appreciate the reviewer's attention to this point, and have corrected the text accordingly as follows:

"In lung fibroblasts, Tgf- β signaling induces expression of Fzd8 and its ligand Wnt5b, which together activate non-canonical Wnt signaling (Spanjer et al, 2016). We found Tgf- β 2 expressed in dermal cells at the follicle base beginning at E14 and persisting through E17 (Fig. 3H), coinciding with high SMA in the DP and Fzd8 at the epidermal tip. These observations raise the possibility that DP cells-secreted Tgf- β 2 may act as a chemotactic cue for epidermal invagination, consistent with Fzd8 expression at the epidermal tip."

Editorial:

During our standard source data check, we noticed unusually clustered numerical repetitions in the source data for figures 1D (E8 epidermis only tab) and 2F-G (please check rows 41 (neck row 3), 185 and 742 (neck row 4) and 152 (neck row 5)). I have attached the corresponding files with the detected duplications labelled in colour. Please take a look and correct if needed. A brief explanation would be very helpful - I appreciate that these duplications can also occur due to specific measurement or calculation methods used.

RE: We have carefully checked the data of Fig. 2F-G and Fig. 1D again. In Fig. 2F-G, the repeated values arise from the ellipse-fitting geometry (cases where $\varphi = 0^\circ$ or 90° , or where $a = b$) and are expected outcomes of the analysis. The raw data themselves are correct and do not require any correction.

As for Fig. 1D, the repeated values observed in the original dataset are the attributes of smoothing, baseline adjustments and rounding during the earlier analysis procedure (back in 2014) of measurements on the curved E8 epidermis around the feather bud. We have re-analyzed the data points in question and additional force curves in the surrounding region using our current analysis pipeline with adjusted smoothing and baseline, and the new raw data is submitted.

Please see below for the extended clarifications.

Clarification on repeated numerical values in the source data for Fig. 2F–G

The highlighted entries correspond to rows 41 (neck row 3), 185 and 742 (neck row 4), and 152 (neck row 5). These rows come from different worksheets within the same raw data Excel file. The repeated values arise from expected outcomes of the ellipse-fitting procedure used to extract cell shape parameters and therefore do not require correction.

Each segmented cell is fitted with an ellipse defined by semi-axes (a , b), orientation angle (φ), and center position (X_0 , Y_0). The coordinates (X_0_in , Y_0_in) represent the same center after a rotation step used only for visual inspection to align the ellipse with the image axes. For rows 41 (neck row 3) and 185 and 742 (neck row 4), the fitted orientation angle φ is either 0° or 90° , meaning that the ellipse is already aligned with the image axes. As a result, the rotation does not change the center coordinates ($X_0_in = X_0$, $Y_0_in = Y_0$), producing identical values in those columns.

For row 152 (neck row 5), the fitted semi-axes are equal ($a = b$), corresponding to a nearly circular cell shape; consequently, the derived long and short axes ($2a$ and $2b$) are identical.

These repetitions therefore reflect expected geometric outcomes of the fitting procedure rather than duplicated measurements, and no correction to the source data is required.

Clarification on Fig. 1D “Epidermis only”

Because the E8 epidermis forms a curved epithelial sheet around the feather bud region and does not attach to the substrate well during measurement compared to dermis or whole skin, the raw AFM force curves contain substantial noise and curvature-related artifacts. In order to obtain the force–distance (indentation) slope for fitting, the curves were smoothed and adjusted for baseline prior to model fitting. The slope of the adjusted curve was then used as input for the modified Hertz model to calculate Young’s modulus.

Additionally, these data were originally collected and analyzed in October 2014. At that time, the overall analytical approach was conceptually the same to our current workflow but implemented in a more manual manner, including manual determination of the baseline and slope values. Most importantly, the values were rounded to three digits during calculations.

When relatively high smoothing and baseline adjustment are applied, multiple curves can yield very similar slope values after processing. The slope values were rounded again, and subsequently used as inputs for the Hertz model, which further increased the likelihood of repeated or closely clustered Young’s modulus values in the resulting dataset (some of which were also input manually to 3 digits only).

In this round of revision, we have reanalyzed the data points in question using our current analysis pipeline. We also included the force curves that previously required stronger smoothing and baseline adjustments, particularly in the highly curved feather bud regions.

As you may see from the newly submitted raw data, although some of the recalculated values are different from the original data at individual level, they remain within a similar range, and do not alter the overall stiffness distribution on tissue level.

Given that the updated data does not lead to an observable change in the resulting stiffness heatmap, we have retained Fig. 1D as originally presented. Furthermore, since

the reanalysis of these force curves still required some degree of smoothing (Gaussian smoothing width = 3.00) and baseline adjustments prior to fitting, we have added the following statement to the Methods section:

“For force curves obtained from the epidermis, an additional Gaussian smoothing step (smoothing width = 3.00) and baseline adjustments (Offset + Tilt) were implemented prior to fitting.”

Figure legend for Figure 4 seems to lack the description of the panel 4D. Panels 4F-G are mentioned before panel 4E, while the panels should be called out in an alphabetical order. Please correct.

RE: Thank you for pointing it out. The proper description and adjustments were made to the figure legend for Figure 4.

Before we forward your manuscript to our publishers, we would like to propose some edits in the manuscript abstract and synopsis – the latter to adjust to the journal style and to shorten the text. I have also written a short blurb that will accompany the title of your manuscript in our table of contents. Please take a look at the proposed text changes in the attached text file and let me know if any corrections are needed.

RE: We are satisfied with the text changes and no corrections are needed.

In Figure 6 and the synopsis image, there are a few typos: “Widen feather bud” (should be “widened”, in Fig. 6); “local softening of epidermis” (should be “softening”); “DP chemotatic gradient” (should be “chemotactic”); “Follicle formation” (should be “formation”).

RE: We have corrected the typos in Figure 6 and synopsis image, accordingly.

Dear Ming and Hans,

Thank you for addressing the final editorial requests. I am now pleased to inform you that your manuscript has been accepted for publication in the EMBO Journal. Congratulations with a nice study!

You may qualify for financial assistance for your publication charges - either via a Springer Nature fully open access agreement or an EMBO initiative. Check your eligibility: <https://link.springer.com/journal/44318/how-to-publish-with-us>

If you have any questions, please do not hesitate to contact the Editorial Office. Thank you for this insightful contribution to The EMBO Journal!

With best wishes,

Ieva

Please note that it is The EMBO Journal policy for the transcript of the editorial process (containing referee reports and your response letters) to be published as an online supplement to each paper. If you should prefer removal of any referee-only figures included in the point-by-point response(s), e.g. because they may still be used for future publication or because they have been reproduced from published work by others, please do let us know immediately via response email.

More information is available here: <https://link.springer.com/partners/embo-press/editorial-policies#Peer%20review>